# Pareto Invariant Risk Minimization: Towards Mitigating The Optimization Dilemma in Out-of-Distribution Generalization

**Yongqiang Chen**[1][*]**, Kaiwen Zhou**[1]**, Yatao Bian**[2]**, Binghui Xie**[1]**, Bingzhe Wu**[2]
[1]The Chinese University of Hong Kong [2]Tencent AI Lab [3]Hong Kong Baptist University
`{yqchen,kwzhou,bhxie21,hyang,klma,jcheng}@cse.cuhk.edu.hk`
**Yonggang Zhang**[3]**, Han Yang**[1]**, Kaili Ma**[1]**, Peilin Zhao**[2]**, Bo Han**[3]**, James Cheng**[1]
`{yatao.bian,wubingzheagent}@gmail.com masonzhao@tencent.com`
`{csygzhang,bhanml}@comp.hkbu.edu.hk`

## Abstract

Recently, there has been a growing surge of interest in enabling machine learning systems to generalize well to Out-of-Distribution (OOD) data. Most efforts are devoted to advancing *optimization objectives* that regularize models to capture the underlying invariance; however, there often are compromises in the *optimization process* of these OOD objectives: i) Many OOD objectives have to be relaxed as penalty terms of Empirical Risk Minimization (ERM) for the ease of optimization, while the relaxed forms can weaken the robustness of the original objective; ii) The penalty terms also require careful tuning of the penalty weights due to the intrinsic conflicts between ERM and OOD objectives. Consequently, these compromises could *easily lead to suboptimal performance* of either the ERM or OOD objective. To address these issues, we introduce a multi-objective optimization (MOO) perspective to understand the OOD optimization process, and propose a new optimization scheme called **PA**reto **I**nvariant **R**isk Minimization (`PAIR`). `PAIR` improves the robustness of OOD objectives by cooperatively optimizing with other OOD objectives, thereby bridging the gaps caused by the relaxations. Then `PAIR` approaches a Pareto optimal solution that trades off the ERM and OOD objectives properly. Extensive experiments on challenging benchmarks, WILDS, show that `PAIR` alleviates the compromises and yields top OOD performances.[1]

## 1 Introduction

The interplay between optimization and generalization is crucial to the success of deep learning (Zhang et al., 2017; Arora et al., 2019; Allen-Zhu et al., 2019; Jacot et al., 2021; Allen-Zhu & Li, 2021). Guided by empirical risk minimization (ERM) (Vapnik, 1991), simple optimization algorithms can find uneventful descent paths in the non-convex loss landscape of deep neural networks (Sagun et al., 2018). However, when distribution shifts are present, the optimization is usually biased by spurious signals such that the learned models can fail dramatically in *Out-of-Distribution* (OOD) data (Beery et al., 2018; DeGrave et al., 2021; Geirhos et al., 2020). Therefore, overcoming the OOD generalization challenge has drawn much attention recently. Most efforts are devoted to proposing better *optimization objectives* (Rojas-Carulla et al., 2018; Koyama & Yamaguchi, 2020; Parascandolo et al., 2021; Krueger et al., 2021; Creager et al., 2021; Liu et al., 2021; Pezeshki et al., 2021; Ahuja et al., 2021a; Wald et al., 2021; Shi et al., 2022; Rame et al., 2021; Chen et al., 2022b) that regularize the gradient signals produced by ERM, while it has been long neglected that the interplay between optimization and generalization under distribution shifts has already changed its nature.

In fact, the *optimization process* of the OOD objectives turns out to be substantially more challenging than ERM. There are often compromises when applying the OOD objectives in practice. Due to the optimization difficulty, many OOD objectives have to be relaxed as penalty terms of ERM in

---

[*]Work done during an internship at Tencent AI Lab.
[1]Code is available at `https://github.com/LFhase/PAIR`.

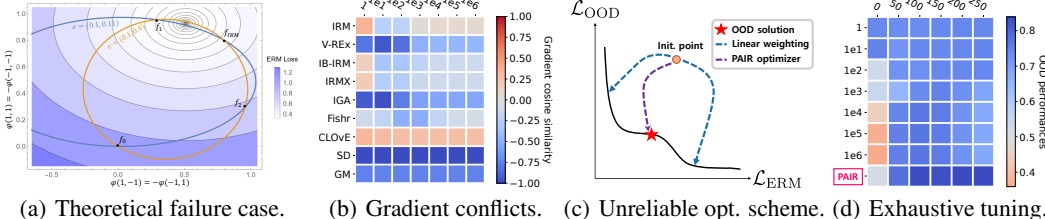

(a) Theoretical failure case.  (b) Gradient conflicts.  (c) Unreliable opt. scheme. (d) Exhaustive tuning.

Figure 1: Optimization issues in OOD algorithms. (a) OOD objectives such as IRM usually require several relaxations for the ease of optimization, which however introduces huge gaps. The ellipsoids denote solutions that satisfy the invariance constraints of practical IRM variant IRMv1. When optimized with ERM, IRMv1 prefers $f_1$ instead of $f_{IRM}$ (The predictor produced by IRM). (b) The gradient conflicts between ERM and OOD objectives generally exist for different objectives at different penalty weights ($x$-axis). (c) The typically used linear weighting scheme to combine ERM and OOD objectives requires careful tuning of the weights to approach the solution. However, the scheme cannot reach any solutions in the non-convex part of the Pareto front. In contrast, PAIR finds an adaptive descent direction under gradient conflicts that leads to the desired solution. (d) Due to the optimization dilemma, the best OOD performance (e.g., IRMv1 w.r.t. a modified COLOREDMNIST from Sec. 5) usually requires exhaustive tuning of hyperparameters ($y$-axis: penalty weights; $x$-axis: pretraining epochs), while PAIR robustly yields top performances by resolving the compromises.

practice (Arjovsky et al., 2019; Koyama & Yamaguchi, 2020; Krueger et al., 2021; Pezeshki et al., 2021; Ahuja et al., 2021a; Rame et al., 2021), but the relaxed formulations can behave very differently from the original objective (Kamath et al., 2021) (Fig. 1(a)). Moreover, due to the generally existing gradient conflicts between ERM and OOD objectives (Fig. 1(b)), trade-offs among ERM and OOD performance during the optimization are often needed. Sagawa* et al. (2020); Zhai et al. (2022) suggest that ERM performance usually needs to be sacrificed for better OOD generalization. On the other hand, it usually requires careful tuning of the OOD penalty hyperparameters (Zhang et al., 2022a) (Fig. 1(d)), which however either weakens the power of OOD objectives or makes them too strong that prevents models from capturing all desirable patterns. Consequently, using the traditional optimization wisdom to train and select models can *easily lead to suboptimal performance* of either ERM or OOD objectives. Most OOD objectives remain struggling with distribution shifts or even underperform ERM (Gulrajani & Lopez-Paz, 2021; Koh et al., 2021). This phenomenon calls for a better understanding of the optimization in OOD generalization, and raises a challenging question:

*How can one obtain a desired OOD solution under the conflicts of ERM and OOD objectives?*

To answer this question, we take a multi-objective optimization (MOO) perspective of the OOD optimization. Specifically, using the representative OOD objective IRM (Arjovsky et al., 2019) as an example, we find that the failures in OOD optimization can be attributed to two issues. The first one is the compromised robustness of OOD objectives due to the relaxation in the practical variants. In fact, it can even eliminate the desired invariant solution from the Pareto front w.r.t. the ERM and the OOD penalty (Fig. 1(a)). Therefore, merely optimizing the ERM and the relaxed OOD penalty can hardly approach the desired solution. On the other hand, when the Pareto front contains the desired solution, as shown in Fig. 1(c), using the traditional linear weighting scheme that linearly reweights the ERM and OOD objectives, cannot reach the solution if it lies in the non-convex part of the front (Boyd & Vandenberghe, 2014). Even when the OOD solution is reachable (i.e., lies in the convex part), it still requires careful tuning of the OOD penalty weights to approach the solution, as shown in Fig. 1(d).

To address these issues, we propose a new optimization scheme for OOD generalization, called **PA**reto **I**nvariant **R**isk Minimization (PAIR), which includes a new optimizer (PAIR-o) and a new model selection criteria (PAIR-s). Owing to the MOO formulation, PAIR-o allows for cooperative optimization with other OOD objectives to improve the robustness of practical OOD objectives. Despite the huge gaps between IRMv1 and IRM, we show that incorporating VREx (Krueger et al., 2021) into IRMv1 provably recovers the causal invariance (Arjovsky et al., 2019) for some group of problem instances (Sec. 3.2). When given robust OOD objectives, PAIR-o finds a descent path with adaptive penalty weights, which leads to a Pareto optimal solution that trades off ERM and OOD performance properly (Sec. 4). In addition, the MOO analysis also motivates PAIR-s, which facilitates the OOD model selection by considering the trade-offs between ERM and OOD objectives.

We conducted extensive experiments on challenging OOD benchmarks. Empirical results show that PAIR-o successfully alleviates the objective conflicts and empowers IRMv1 to achieve high perfor-

mance in 6 datasets from WILDS (Koh et al., 2021). PAIR-s effectively improves the performance of selected OOD models up to $10\%$ across 3 datasets from DOMAINBED (Gulrajani & Lopez-Paz, 2021), demonstrating the significance of considering the ERM and OOD trade-offs in optimization.

## 2 BACKGROUND AND RELATED WORK

We first briefly introduce the background of our work (more details are given in Appendix B.1.

**Problem setup.** The problem of OOD generalization typically considers a supervised learning setting based on the data $\mathcal{D} = \{\mathcal{D}^e\}_{e \in \mathcal{E}_{\text{all}}}$ collected from multiple causally related environments $\mathcal{E}_{\text{all}}$, where a subset of samples $\mathcal{D}^e = \{X_i^e, Y_i^e\}$ from a single environment $e \in \mathcal{E}_{\text{all}}$ are drawn independently from an identical distribution $\mathbb{P}^e$ (Peters et al., 2016). Given the data from training environments $\{\mathcal{D}^e\}_{e \in \mathcal{E}_{\text{tr}}}$, the goal of OOD generalization is to find a predictor $f : \mathcal{X} \to \mathcal{Y}$ that generalizes well to all (unseen) environments, i.e., to minimize $\max_{e \in \mathcal{E}_{\text{all}}} \mathcal{L}_e(f)$, where $\mathcal{L}_e$ is the empirical risk under environment $e$. The predictor $f = w \circ \varphi$ is usually composed of a featurizer $\varphi : \mathcal{X} \to \mathcal{Z}$ that learns to extract useful features, and a classifier $w : \mathcal{Z} \to \mathcal{Y}$ that makes predictions from the extracted features.

**Existing solutions to OOD generalization.** There exists a rich literature aiming to overcome the OOD generalization challenge, which usually appear as *additional regularizations* of ERM (Vapnik, 1991). Ganin et al. (2016); Sun & Saenko (2016); Li et al. (2018); Dou et al. (2019) regularize the learned features to be **domain-invariant**. Namkoong & Duchi (2016); Hu et al. (2018); Sagawa* et al. (2020) regularize the models to be **robust to mild distributional perturbations** of the training distributions, and Zhang et al. (2022c); Liu et al. (2021); Zhang et al. (2022b); Yao et al. (2022) improve the robustness with additional assumptions. Recently there is increasing interest in adopting the causality theory (Pearl, 2009; Schölkopf et al., 2021) and introducing the **causal invariance** to representation learning (Peters et al., 2016; Arjovsky et al., 2019; Creager et al., 2021; Parascandolo et al., 2021; Wald et al., 2021; Ahuja et al., 2021a). They require $\varphi$ to learn causally invariant representations such that a predictor $w$ acting on $\varphi$ minimizes the risks of all the environments simultaneously. This work focuses on resolving the optimization issue in learning the causal invariance. In addition, Koyama & Yamaguchi (2020); Krueger et al. (2021); Shi et al. (2022); Rame et al. (2021) implement the invariance by encouraging **agreements** at various levels across environments. However, they mostly focus on developing better objectives while neglecting the optimization process of the objectives.

**Optimization dilemma in OOD generalization.** Along with the development of OOD methods, the OOD optimization dilemma is gradually perceived in the literature. Gulrajani & Lopez-Paz (2021) find it hard to select a proper model in OOD generalization given ERM performance at different environments. Sagawa* et al. (2020); Zhai et al. (2022) find the ERM performance needs to be sacrificed for satisfactory OOD performance. Some initial trials are proposed. Lv et al. (2021) use the guidance of the data from similar distributions with the test environment in MOO to resolve gradient conflicts and achieve better performance in domain adaption. Zhang et al. (2022a) propose to construct diverse initializations for stabilizing OOD performance under the dilemma. However, why there exists such a dilemma in OOD generalization and whether we can resolve it remain elusive.

**Multi-Objective Optimization (MOO).** MOO considers solving $m$ objectives w.r.t. $\{\mathcal{L}_i\}_{i=1}^m$ losses, i.e., $\min_\theta \boldsymbol{L}(\theta) = (\mathcal{L}_1(\theta), ..., \mathcal{L}_m(\theta))^T$ (Kaisa, 1999). A solution $\theta$ dominates another $\bar{\theta}$, i.e., $\boldsymbol{L}(\theta) \preceq \boldsymbol{L}(\bar{\theta})$, if $\mathcal{L}_i(\theta) \leq \mathcal{L}_i(\bar{\theta})$ for all $i$ and $\boldsymbol{L}(\theta) \neq \boldsymbol{L}(\bar{\theta})$. A solution $\theta^*$ is called **Pareto optimal** if no other $\theta$ dominates $\theta^*$. The set of Pareto optimal solutions is called Pareto set ($\mathcal{P}$) and its image is called **Pareto front**. In practice, it is usual that one cannot find a global optimal solution for all objectives, hence Pareto optimal solutions are of particular value. Although MOO has been widely applied to improving multi-task learning (Sener & Koltun, 2018), it remains underexplored on how to model and mitigate objective conflicts in OOD generalization from the MOO perspective.

## 3 OPTIMIZATION CHALLENGES IN IRM AND ITS EFFECTIVE FIX

This work focus on one of the most representative OOD objectives in learning the causal invariance–IRM, to show how we can understand and mitigate the optimization dilemma through the MOO lens.

### 3.1 DRAWBACKS OF IRM IN PRACTICE

We first introduce the basics of IRM and the drawbacks of its practical variants, and leave theoretical details in Appendix C.1. Specifically, the IRM framework approaches OOD generalization by finding

an invariant representation $\varphi$, such that there exists a classifier acting on $\varphi$ that is simultaneously optimal in $\mathcal{E}_{\text{tr}}$. Hence, IRM leads to a challenging bi-level optimization problem as

$$\min_{w,\varphi} \sum_{e \in \mathcal{E}_{\text{tr}}} \mathcal{L}_e(w \circ \varphi), \text{s.t. } w \in \arg\min_{\bar{w}:\mathcal{Z}\to\mathcal{Y}} \mathcal{L}_e(\bar{w} \circ \varphi), \ \forall e \in \mathcal{E}_{\text{tr}}. \tag{1}$$

Given the training environments $\mathcal{E}_{\text{tr}}$, and functional spaces $\mathcal{W}$ for $w$ and $\Phi$ for $\varphi$, predictors $f = w \circ \varphi$ satisfying the constraint in Eq. 1 are called invariant predictors, denoted as $\mathcal{I}(\mathcal{E}_{\text{tr}})$. When solving for invariant predictors, characterizing $\mathcal{I}(\mathcal{E}_{\text{tr}})$ is particularly difficult in practice, hence it is natural to restrict $\mathcal{W}$ to be the space of linear functions on $\mathcal{Z} = \mathbb{R}^d$ (Jacot et al., 2021). Furthermore, Arjovsky et al. (2019) argue that linear classifiers actually do not provide additional representation power than *scalar* classifiers, i.e., $d = 1, \mathcal{W} = \mathcal{S} = \mathbb{R}^1$. The scalar restriction elicits a practical variant $\text{IRM}_\mathcal{S}$ as

$$\min_{\varphi} \sum_{e \in \mathcal{E}_{\text{tr}}} \mathcal{L}_e(\varphi), \text{s.t. } \nabla_{w|w=1}\mathcal{L}_e(w \cdot \varphi) = 0, \ \forall e \in \mathcal{E}_{\text{tr}}. \tag{2}$$

Since Eq. 2 remains a constrained programming. Arjovsky et al. (2019) further introduce a soften-constrained variant, called IRMv1, as the following

$$\min_{\varphi} \sum_{e \in \mathcal{E}_{\text{tr}}} \mathcal{L}_e(\varphi) + \lambda |\nabla_{w|w=1}\mathcal{L}_e(w \cdot \varphi)|^2. \tag{3}$$

**Theoretical failure of practical IRM variants.** Although the practical variants seem promising, the relaxations introduce huge gaps between IRM and the practical variants, so that both $\text{IRM}_\mathcal{S}$ and IRMv1 can fail to capture the invariance (Kamath et al., 2021). The failure case is illustrated by the two-bit environment with $\alpha_e, \beta_e \in [0, 1]$. Each environment $\mathcal{D}_e = \{X^e, Y^e\}$ is generated following

$$Y^e := \text{Rad}(0.5), \ X^e := (X_1^e, X_2^e), \ X_1^e := Y^e \cdot \text{Rad}(\alpha_e), \ X_2^e := Y^e \cdot \text{Rad}(\beta_e), \tag{4}$$

where $\text{Rad}(\sigma)$ is a random variable taking value $-1$ with probability $\sigma$ and $+1$ with probability $1 - \sigma$. Each environment is denoted as $\mathcal{E}_\alpha = \{(\alpha, \beta_e) : 0 < \beta_e < 1\}$ where $X_1^e$ is the invariant feature as $\alpha$ is fixed for different environment $e$, and $X_2^e$ is the spurious feature as $\beta_e$ varies across different $e$.

Let $\mathcal{I}_\mathcal{S}(\mathcal{E}_{\text{tr}})$ denote the set of invariant predictors elicited by the relaxed constraint in $\text{IRM}_\mathcal{S}$. It follows that $\mathcal{I}(\mathcal{E}_{\text{tr}}) \subseteq \mathcal{I}_\mathcal{S}(\mathcal{E}_{\text{tr}})$. Consequently, there exist some undesired predictors but considered "invariant" by $\text{IRM}_\mathcal{S}$ and IRMv1. For example, in $\mathcal{E}_{\text{tr}} = \{(0.1, 0.11), (0.1, 0.4)\}$, the solutions satisfying the constraint in $\text{IRM}_\mathcal{S}$ are those intersected points in Fig. 1(a) (The ellipsoids are the constraints). Although $f_1, f_{\text{IRM}} \in \mathcal{I}_\mathcal{S}(\mathcal{E}_{\text{tr}})$, both $\text{IRM}_\mathcal{S}$ and IRMv1 prefer $f_1$ instead of $f_{\text{IRM}}$ (the predictor produced by IRM), as $f_1$ has the smallest ERM loss. In fact, Kamath et al. (2021) show that the failure can happen in a wide range of environments even given *infinite* amount of environments and samples, demonstrating the huge gap between the practical and the original IRM variants.

**Empirical drawback of practical IRM variants.** In addition, the optimization of IRMv1 introduces more challenges due to the conflicts between the IRMv1 penalty and ERM objective. As shown in Fig. 1(d), it often requires significant efforts to tune the hyperparameters such as pretraining epochs and penalty weights $\lambda$ in Eq. 3. Otherwise, the IRMv1 penalty could be either too weak to enforce the invariance as required by IRM, or too strong that prevents ERM from learning all desirable patterns.

## 3.2 Pareto Optimization for IRM

As shown that both $\text{IRM}_\mathcal{S}$ and IRMv1 fail to properly trade off between ERM and IRM objectives, we switch to a new perspective, i.e., the lens of MOO, to understand the failures of IRM in practice.

**Understanding the IRM failures through the MOO perspective.** To begin with, it is natural to reformulate the practical IRM problem (Eq. 3) as an MOO problem:

$$\min_{\varphi}(\mathcal{L}_{\text{ERM}}, \mathcal{L}_{\text{IRM}})^T, \tag{5}$$

where $\mathcal{L}_{\text{ERM}} = \frac{1}{|\mathcal{E}_{\text{tr}}|} \sum_{e \in \mathcal{E}_{\text{tr}}} \mathcal{L}_e$ denotes the ERM loss, and $\mathcal{L}_{\text{IRM}} = \sum_e |\nabla_{w|w=1}\mathcal{L}_e(w \cdot \varphi)|^2$ denotes the practical IRMv1 loss. To understand the behaviors of solutions to Eq. 5, We visualize the Pareto front w.r.t. $\{\mathcal{L}_e\}_{e \in \mathcal{E}_{\text{tr}}}$ using the previous failure case in Fig. 1(a).

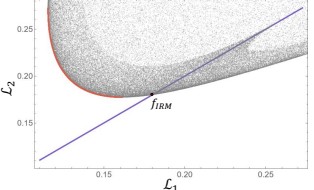

Figure 2: Pareto front of ERM losses w.r.t. environments.

Let $\mathcal{P}(\mathcal{L}_1(\theta), ..., \mathcal{L}_m(\theta))$ denote the set of Pareto optimal solutions w.r.t. $(\mathcal{L}_1(\theta), ..., \mathcal{L}_m(\theta))$. As shown in Fig. 2, at first, we can find that $f_{\text{IRM}} \notin \mathcal{P}(\mathcal{L}_1, \mathcal{L}_2)$. In other words, solving any

environment-reweighted ERM losses cannot obtain $f_{\text{IRM}}$. Moreover, together with Fig. 1(a), the failure remains even combined with the $\text{IRM}_{\mathcal{S}}$ or IRMv1, i.e., $f_{\text{IRM}} \notin \mathcal{P}(\mathcal{L}_1, \mathcal{L}_2, \mathcal{L}_{\text{IRM}})$, hence $f_{\text{IRM}} \notin \mathcal{P}(\mathcal{L}_{\text{ERM}}, \mathcal{L}_{\text{IRM}})$, as $f_{\text{IRM}}$ is dominated by $f_1$. Therefore, no matter how we carefully control the optimization process, we cannot obtain $f_{\text{IRM}}$ by merely minimizing the objectives in Eq. 5. This is essentially because of the weakened OOD robustness of $\text{IRM}_{\mathcal{S}}$ and IRMv1 caused by the relaxations. Thus, choosing robust objectives for optimization is of great importance to OOD generalization. The ideal objectives should at least constitute a Pareto front that contains the desired OOD solution.

**Improving OOD robustness of practical IRM variants.** In pursuit of proper optimization objectives, we resort to the OOD extrapolation explanation of IRM (Bottou et al., 2019). A solution that is simultaneously optimal to all training environments (i.e., satisfying the original IRM constraints) is also a stationary point of ERM loss w.r.t. some OOD distribution:

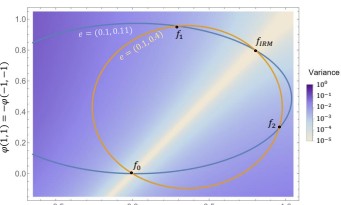

Figure 3: Variance distribution.

$$\partial \mathcal{L}_t / \partial f_{\text{IRM}} = \mathbf{0}, \ \mathcal{L}_t \in \{\textstyle\sum_{e \in \mathcal{E}_{\text{tr}}} \lambda_e \mathcal{L}_e | \textstyle\sum_{e \in \mathcal{E}_{\text{tr}}} \lambda_e = 1\}, \quad (6)$$

where $\mathcal{L}_t$ is the ERM loss under the OOD distribution. Different from Distributionally Robust Optimization approaches (Namkoong & Duchi, 2016), Eq. 6 allows for some negative $\lambda_e$ and hence its solutions are expected to extrapolate better (Bottou et al., 2019).

The previous failure case implies that both $\text{IRM}_{\mathcal{S}}$ and IRMv1 fail in the extrapolation due to the relaxations, nevertheless, we can introduce additional objectives to directly improve the OOD extrapolation power of the practical IRM variants. To this end, we introduce the REx objective to IRMv1, which is derived by directly minimizing the worst case ERM loss under all OOD distributions up to a certain distance from the training distributions (Krueger et al., 2021). More formally, REx minimizes the worst case $\mathcal{L}_t$ under an additional constraint of $\{\lambda_e\}_{e \in \mathcal{E}_{\text{tr}}} \geq -\beta$ in Eq. 6. For the ease of optimization, they also propose an alternative objective as $\mathcal{L}_{\text{VREx}} := \text{var}(\{\mathcal{L}_e\}_{e \in \mathcal{E}_{\text{tr}}})$. In Fig. 3, we plot the distribution of $\mathcal{L}_{\text{VREx}}$ in the the failure case of Fig. 1(a). It can be found that, $f_{\text{IRM}}$ lies in the low variance region. Similarly, in Fig. 2, the zero variance solutions (shown as the purple line at middle) points out the underlying $f_{\text{IRM}}$ beyond the Pareto front. Therefore, incorporating $\mathcal{L}_{\text{VREx}}$ in Eq. 5 can relocate $f_{\text{IRM}}$ into the Pareto front, which implies the desirable objectives as the following

$$(\text{IRMX}) \qquad\qquad \min_{\varphi}(\mathcal{L}_{\text{ERM}}, \mathcal{L}_{\text{IRM}}, \mathcal{L}_{\text{VREx}})^T. \qquad\qquad (7)$$

By resolving a large class of failure cases of $\text{IRM}_{\mathcal{S}}$ and IRMv1 (Kamath et al., 2021), solutions to Eq. 7 are more powerful than those to $\text{IRM}_{\mathcal{S}}$ and IRMv1 in OOD extrapolation. In fact, we have

**Proposition 1.** *(Informal) Under Setting A (Kamath et al. (2021)), for all $\alpha \in (0,1)$, let $\mathcal{E} := \{(\alpha, \beta_e) : \beta_e \in (0,1)\}$ be any instance of the two-bit environment (Eq. 4), $\mathcal{I}_X$ denote the invariant predictors produced by Eq. 7, it holds that $\mathcal{I}_X(\mathcal{E}) = \mathcal{I}(\mathcal{E})$.*[2]

The formal description and proof of Proposition 1 are given in Appendix E.1. Proposition 1 implies that Eq. 7 are the ideal objectives for optimization. However, Eq. 7 can even add up the difficulty of OOD penalty tunning. It introduces one more penalty to the overall objective that makes the Pareto front more complicated for the linear weighting scheme to find the desired solution.

**Pareto optimization for IRMX.** Ideally, the set of Pareto optimal solutions is small such that each $f \in \mathcal{P}(\mathcal{L}_{\text{ERM}}, \mathcal{L}_{\text{IRM}}, \mathcal{L}_{\text{VREx}})$ satisfies the invariance constraints of IRMv1 and VREx, i.e., $\mathcal{L}_{\text{IRM}} = 0$ and $\mathcal{L}_{\text{VREx}} = 0$, and with a minimal $\mathcal{L}_{\text{ERM}}$, thereby eliciting the desired OOD solutions. However, the ideal constraints might be too strong to be achieved when there are noises among invariant features and labels (Duchin et al., 2020; Ahuja et al., 2021b), which will future enlarge the set of Pareto optimal solutions. Therefore, it is natural to relax the constraints as $\mathcal{L}_{\text{IRM}} \leq \epsilon_{\text{IRM}}$ and $\mathcal{L}_{\text{VREx}} \leq \epsilon_{\text{VREx}}$. When $\epsilon_{\text{IRM}} \to 0, \epsilon_{\text{VREx}} \to 0$, it recovers the ideal invariance. To obtain a desired solution under these circumstances, the optimization process is expected to meet the following two necessities:

(i). The additional objective in IRMX can make the Pareto front more complicated such that the desired solutions are more likely to appear in the non-convex part, which are however not reachable by the linear weighting scheme (Boyd & Vandenberghe, 2014). Therefore, the optimizer needs to be able to reach any Pareto optimal solutions in the front, e.g., MGDA algorithms (Désidéri, 2012).[3]

---

[2]Readers might be interested in the necessities of keeping IRMv1 in the objectives. Proposition 1 considers only the ideal case, we additionally provide more empirical reasons in Appendix C.2; Our results can also be extended to multi-class following typical machine learning theory practice.

[3]We leave more sophisticated Pareto front exploration methods (Zhang & Golovin, 2020; Ma et al., 2020) to future investigation.

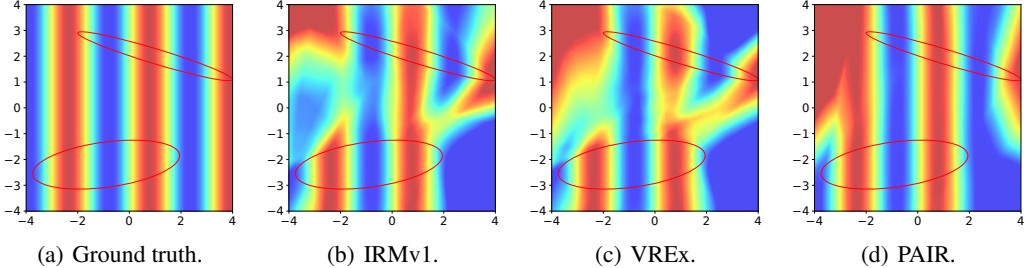

|  (a) Ground truth. | (b) IRMv1. | (c) VREx. | (d) PAIR. |

Figure 4: Recovery of causal invariance. The causal invariance (Definition. 3.1) requires the model predictions to be independent of the spurious features within the overlapped invariant features. In this example, intuitively it requires the colored belts to be perpendicular to $x$-axis within $[-2, 2]$. It can be found that PAIR succeeds out of IRMv1 and VREx in recovering the causal invariance.

(ii). When both $\epsilon_{\text{IRM}}, \epsilon_{\text{VREx}} > 0$, there can be multiple Pareto optimal solutions while there are few desired OOD solutions. Hence a preference of ERM and OOD objectives is usually needed. As the optimality of each OOD objective usually appears as a necessary condition for satisfactory OOD performance, the preferences for OOD objectives are expected to be higher.

Given the two requirements, we leverage a preference-aware MOO solver to solve IRMX for the desired Pareto optimal solution (Mahapatra & Rajan, 2020). We summarize the overall solution as **PA**reto **I**nvariant **R**isk Minimization (PAIR). When assigning a high preference to $\mathcal{L}_{\text{IRM}}$ and $\mathcal{L}_{\text{VREx}}$ in IRMX (Eq. 7), PAIR approaches a Pareto optimal solution that minimizes the OOD losses while not sacrificing the ERM performance too much, and has good OOD performance, shown as in Table. 1.

### 3.3 RECOVERY OF CAUSAL INVARIANCE

To better understand how PAIR bridges the gaps between the practical and original IRM objectives, we examine to what extent PAIR can recover the causal invariance specified by Arjovsky et al. (2019) in a more difficult case. More formally, the causal invariance is defined as follows.

**Definition 3.1.** *(Causal Invariance) Given a predictor $f := w \circ \varphi$, the representation produced by the featurizer $\varphi$ is invariant over $\mathcal{E}_{all}$ if and only if for all $e_1, e_2 \in \mathcal{E}_{all}$, it holds that*

$$\mathbb{E}_{\mathcal{D}_{e_1}}[Y|\varphi(X) = z] = \mathbb{E}_{\mathcal{D}_{e_2}}[Y|\varphi(X) = z],$$

*for all $z \in \mathcal{Z}_\varphi^{e_1} \cap \mathcal{Z}_\varphi^{e_2}$, where $\mathcal{Z}_\varphi^e := \{\varphi(X)|(X, Y) \in supp(\mathcal{D}_e)\}$.*

Following Definition 3.1, we construct a regression problem. As shown in Fig. 4, $Y = \sin(X_1) + 1$ is solely determined by $X_1$, i.e., the values of the $x$-axis, while $X_2$ is the values of $y$-axis and does not influence the values of $Y$. Different colors indicate different values of $Y$. In this problem, the invariant representation $\varphi$ should only take $X_1$ and discard $X_2$. We sampled two training environments as denoted by the ellipsoids colored in red, among which the overlapped region of the invariant features $X_1$ is $[-2, 2]$. Hence the prediction produced by the invariant predictor following Definition 3.1 is expected to be independent of $X_2$. In other words, the plotted belts need to be perpendicular to the $x$-axis within the overlapped invariant features $[-2, 2]$. More details can be found in Appendix C.3.

We plot predictions with the best MSE losses of IRMv1 and VREx in Fig. 4(b) and Fig. 4(c), respectively. Although both IRMv1 and VREx fail to achieve the causal invariance as expected, perhaps surprisingly, PAIR almost recovers the causal invariance, as shown in Fig. 4(d).

## 4 PARETO INVARIANT RISK MINIMIZATION

The success of PAIR in empowering unrobust IRMv1 to achieve the causal invariance of IRM demonstrates the significance of considering the trade-offs between ERM and OOD objectives in the optimization. In the next, we will summarize our findings and elaborate PAIR in more details.

### 4.1 METHODOLOGY OUTCOMES

**Key takeaways from the IRM example.** To summarize, the failures of OOD optimization can be attributed to: i) Using unrobust objectives for optimization; ii) Using unreliable scheme to approach the desired solution. Nevertheless, we can improve the robustness of the OOD objectives by

introducing additional guidance such that the desired solution is relocated in the Pareto front w.r.t. the new objectives. After obtaining robust objectives to optimize, we then leverage a preference-aware MOO solver to find the Pareto optimal solutions that maximally satisfy the invariance constraints by assigning the OOD objective a higher preference while being aware of retaining ERM performance.

More formally, let $f_{\mathrm{ood}}$ be the desired OOD solution and $\mathcal{F}$ be the functional class of $f_{\mathrm{ood}}$, a group of OOD objectives $\boldsymbol{L}_{\mathrm{ood}} = \{\mathcal{L}_{\mathrm{ood}}^i\}_{i=1}^m$ are robust if their composite objective $\boldsymbol{L}_{\mathrm{ood}}$ satisfies that

$$\boldsymbol{L}_{\mathrm{ood}}(f_{\mathrm{ood}}) \preceq \boldsymbol{L}_{\mathrm{ood}}(f), \forall f \neq f_{\mathrm{ood}} \in \mathcal{F}, \tag{8}$$

When given a robust OOD objective $\boldsymbol{L}_{\mathrm{ood}}$, our target is to solve the following MOO problem

$$\min_f (\mathcal{L}_{\mathrm{ERM}}, \boldsymbol{L}_{\mathrm{ood}})^T, \tag{9}$$

where $\boldsymbol{L}_{\mathrm{ood}}$ corresponds to an $\epsilon_{\mathrm{ood}}$-relaxed invariance constraint as $\boldsymbol{L}_{\mathrm{ood}}(f_{\mathrm{ood}}) = \epsilon_{\mathrm{ood}} \preceq \boldsymbol{L}_{\mathrm{ood}}(f), \forall f \neq f_{\mathrm{ood}} \in \mathcal{F}$. Denote the $\epsilon_{\mathrm{inv}}$ as empirical loss of using the underlying invariant features to predict labels, then the optimal values of the desired OOD solution w.r.t. Eq. 9 are $(\epsilon_{\mathrm{inv}}, \epsilon_{\mathrm{ood}})^T = (\mathcal{L}_{\mathrm{ERM}}(f_{\mathrm{ood}}), \boldsymbol{L}_{\mathrm{ood}}(f_{\mathrm{ood}}))^T$, which corresponds to an ideal preference (or OOD preference) for the objectives, that is $\boldsymbol{p}_{\mathrm{ood}} = (\epsilon_{\mathrm{inv}}^{-1}, \epsilon_{\mathrm{ood}}^{-1})^T$. The optimal solutions of Eq. 9 that satisfy the exact Pareto optimality, i.e., $\boldsymbol{p}_{\mathrm{ood}\,i}\mathcal{L}_i = \boldsymbol{p}_{\mathrm{ood}\,j}\mathcal{L}_j, \forall \mathcal{L}_i, \mathcal{L}_j \in \boldsymbol{L}$, are expected to recover $f_{\mathrm{ood}}$ in Eq. 8.

**`PAIR-o` as an optimizer for OOD generalization.** To find a desired Pareto optimal solution specified by $\boldsymbol{p}_{\mathrm{ood}}$, we adopt a 2-stage optimization scheme, which consists of two phases, i.e., the "descent" and the "balance" phase, following the common practice (Gulrajani & Lopez-Paz, 2021).

In the "descent" phase, we train the model with the ERM loss such that it approaches the Pareto front by merely minimizing $\mathcal{L}_{\mathrm{ERM}}$ first. Then, in the "balance" phase, we adjust the solution to maximally satisfy the exact Pareto optimality specified by $\boldsymbol{p}_{\mathrm{ood}}$. We adopt the off-the-shelf preference-aware MOO solver EPO (Mahapatra & Rajan, 2020) to find the desired Pareto optimal solutions with the given $\boldsymbol{p}_{\mathrm{ood}}$. Specifically, at each step, $\boldsymbol{p}_{\mathrm{ood}}$ implies a descent direction $\boldsymbol{g}_b$ that maximally increase the satisfaction to the exact Pareto optimality. Then, we will find an objective weight vector to reweight both the ERM and OOD objectives (thus their gradients), such that the reweighted descent direction $\boldsymbol{g}_{\mathrm{dsc}}$ has a maximum angle with $\boldsymbol{g}_b$. Meanwhile, to avoid divergence, $\boldsymbol{g}_{\mathrm{dsc}}$ also needs to guarantee that it has a positive angle with the objective that diverges from the preferred direction most. We provide detailed descriptions and theoretical discussions of the algorithm in Appendix D.1.

**`PAIR-s` for OOD model selection.** Model selection in OOD generalization is known to be challenging, as the validation data used to evaluate the model performance is no longer necessarily identically distributed to the test data (Gulrajani & Lopez-Paz, 2021). The IRM example also implies that the traditional model selection methods that merely depends on the validation performance, i.e., the ERM performance, can easily compromise OOD performance due to the conflicts with ERM objective, especially when the validation set has a large gap between the test set (cf. CMNIST in Table 3).

When given no additional assumption, we posit that the OOD loss values can serve as a proxy for OOD performance, which essentially corresponds to the *underlying prior* assumed in the OOD methods. It naturally resembles `PAIR` optimization therefore motivates `PAIR-s`. `PAIR-s` jointly considers and trades off the ERM and OOD performance in model selection, and select models that maximally satisfy the exact Pareto optimality. We leave more details and discussions in Appendix D.2.

## 4.2 THEORETICAL DISCUSSIONS AND PRACTICAL CONSIDERATIONS

Essentially both `PAIR-o` and `PAIR-s` aim to solve Eq. 9 up to the exact Pareto optimality. However, in practice, the ideal preference is usually unknown and the exact Pareto optimality could be too strict to achieve . Therefore, we develop an $\epsilon$-approximated formulation of Eq. 9, i.e.,$|\boldsymbol{p}_{\mathrm{ood}\,i}\mathcal{L}_i - \boldsymbol{p}_{\mathrm{ood}\,j}\mathcal{L}_j| \leq \epsilon, \forall \mathcal{L}_i, \mathcal{L}_j \in \boldsymbol{L}$, which might be of independent interest. Built upon the relaxed variant, we analyze the OOD performance of `PAIR` in terms of sample complexity, given the empirical risk and imprecise OOD preference, and prove the following Theorem in Appendix E.2.

**Theorem 4.1.** *(Informal) For $\gamma \in (0,1)$ and any $\epsilon, \delta > 0$, if $\mathcal{F}$ is a finite hypothesis class, both ERM and OOD losses are bounded above, let $I_{PAIR}$ be the index of all losses, $p_{\max} := \max_{i \in I_{PAIR}} p_i$ and $L_{\max} := \max_{i \in I_{PAIR}} L_i$, if the number of training samples $|D| \geq (32L_{\max}^2 p_{\max}^2/\delta^2) \log[2(m+1)|\mathcal{F}|/\gamma]$, then with probability at least $1 - \gamma$, `PAIR-o` and `PAIR-s` yield an $\epsilon$-approximated solution of $f_{ood}$.*

**Practical Considerations.** Theorem 4.1 establishes the theoretical guarantee of `PAIR-o` and `PAIR-s` given only an imprecise OOD preference. Empirically, we find that assigning a large

enough preference to the OOD objectives is generally sufficient for `PAIR-o` to find a desired OOD solution. For example, in most experiments `PAIR-o` yields a satisfactory OOD solution with a relative preference of $(1, 1e10, 1e12)$ for ERM, IRMv1, and VREx. For `PAIR-s`, we can estimate the empirical upper bounds of $(\epsilon_{inv}, \epsilon_{ood})$ from the running history and adjust OOD preference to be slightly larger. We provide a detailed discussion on the preference choice in practice in Appendix D.3.

Besides, the requirement of whole network gradients in `PAIR-o` can be a bottleneck when deployed to models that have a prohibitively large number of parameters (Sener & Koltun, 2018). To this end, we can use only the gradients of classifier $w$ to solve for the objective weights, or freeze the featurizer after the "descent" phase to further reduce the resource requirement (Zhang et al., 2022a). We discuss more practical options and how `PAIR` can be applied to other OOD methods in Appendix D.4.

## 5 EXPERIMENTS

We conduct extensive experiments on COLOREDMNIST, WILDS and DOMAINBED to verify the effectiveness of `PAIR-o` and `PAIR-s` in finding a better OOD solution under objective conflicts.

**Proof of concept on COLOREDMNIST.** In Table 1, we compare `PAIR-o` implemented with IRMX to other strong baselines on COLOREDMNIST (CMNIST) and the failure case variant (Kamath et al., 2021) (CMNIST-m). We follow the evaluation setup as in IRM (Arjovsky et al., 2019) and report the results from 10 runs. We assign a relative preference $(1, 1e10, 1e12)$ to ERM, IRMv1 and VREx objectives, respectively. It can be found that `PAIR-o` significantly improves over IRMv1 across all environment settings, while IRMX using the linear weighting scheme performs worse than `PAIR-o`, confirming

Table 1: OOD Performance on COLOREDMNIST

| Method | CMNIST | CMNIST-m | Avg. |
|---|---|---|---|
| ERM | $17.1 \pm 0.9$ | $73.3 \pm 0.9$ | 45.2 |
| IRMv1 | $67.3 \pm 1.9$ | $76.8 \pm 3.2$ | 72.1 |
| V-REx | $68.6 \pm 0.7$ | $82.9 \pm 1.3$ | 75.8 |
| IRMX | $65.8 \pm 2.9$ | $81.6 \pm 2.0$ | 73.7 |
| **PAIR-o**$_f$ | $68.6 \pm 0.9$ | $\mathbf{83.7} \pm 1.2$ | 76.2 |
| **PAIR-o**$_\varphi$ | $68.6 \pm 0.8$ | $\mathbf{83.7} \pm 1.2$ | 76.2 |
| **PAIR-o**$_w$ | $\mathbf{69.2} \pm 0.7$ | $\mathbf{83.7} \pm 1.2$ | **76.5** |
| Oracle | $72.2 \pm 0.2$ | $86.5 \pm 0.3$ | 79.4 |
| Optimum | 75 | 90 | 82.5 |
| Chance | 50 | 50 | 50 |

the effectiveness of `PAIR-o`. Interestingly, using only the gradients of the classifier $w$ in `PAIR-o` can yield competitive performance as that uses $f$ or $\varphi$, while the former has better scalability. Therefore, we will use `PAIR-o`$_w$ in the following experiments. More details are given in Appendix F.1.

Table 2: OOD generalization performances on WILDS benchmark.

| | CAMELYON17 | CIVILCOMMENTS | FMOW | IWILDCAM | POVERTYMAP | RXRX1 | AVG. RANK(↓)[†] |
|---|---|---|---|---|---|---|---|
| | Avg. acc. (%) | Worst acc. (%) | Worst acc. (%) | Macro F1 | Worst Pearson r | Avg. acc. (%) | |
| ERM | 70.3 (±6.4) | 56.0 (±3.6) | 32.3 (±1.25) | 30.8 (±1.3) | 0.45 (±0.06) | 29.9 (±0.4) | 4.50 |
| CORAL | 59.5 (±7.7) | 65.6 (±1.3) | 31.7 (±1.24) | **32.7** (±0.2) | 0.44 (±0.07) | 28.4 (±0.3) | 5.50 |
| GroupDRO | 68.4 (±7.3) | 70.0 (±2.0) | 30.8 (±0.81) | 23.8 (±2.0) | 0.39 (±0.06) | 23.0 (±0.3) | 6.83 |
| IRMv1 | 64.2 (±8.1) | 66.3 (±2.1) | 30.0 (±1.37) | 15.1 (±4.9) | 0.43 (±0.07) | 8.2 (±0.8) | 7.67 |
| V-REx | 71.5 (±8.3) | 64.9 (±1.2) | 27.2 (±0.78) | 27.6 (±0.7) | 0.40 (±0.06) | 7.5 (±0.8) | 7.00 |
| Fish | 74.3 (±7.7) | 73.9 (±0.2) | 34.6 (±0.51) | 24.8 (±0.7) | 0.43 (±0.05) | 10.1 (±1.5) | 4.33 |
| LISA | **74.7** (±6.1) | 70.8 (±1.0) | 33.5 (±0.70) | 24.0 (±0.5) | **0.48** (±0.07) | **31.9** (±0.8) | 2.67 |
| IRMX | 67.0 (±6.6) | 74.3 (±0.8) | 33.7 (±0.78) | 26.6 (±0.9) | 0.45 (±0.04) | 28.7 (±0.2) | 4.00 |
| **PAIR-o** | 74.0 (±7.0) | **75.2** (±0.7) | **35.5** (±1.13) | 27.9 (±0.7) | 0.47 (±0.06) | 28.8 (±0.1) | **2.17** |

[†]Averaged rank is reported because of the dataset heterogeneity. A lower rank is better.

**Can `PAIR-o` effectively find better OOD solutions under realistic distribution shifts?** We evaluate `PAIR-o` implemented with IRMX on 6 challenging datasets from WILDS benchmark (Koh et al., 2021), and compare `PAIR-o` with other state-of-the-art OOD methods from different lines (Sec. 2), including CORAL (Sun & Saenko, 2016), GroupDRO (Sagawa* et al., 2020), IRM (Arjovsky et al., 2019), V-REx (Krueger et al., 2021), Fish (Shi et al., 2022) and an advanced importance-aware data augmentation method LISA (Yao et al., 2022). By default, we assign a relative preference $(1, 1e10, 1e12)$ to ERM, IRMv1 and VREx objectives, respectively, and restrict the search space of the preference. Our implementation and evaluation protocol follow the exact configuration as previous works (Koh et al., 2021; Shi et al., 2022; Yao et al., 2022). Details can be found in Appendix F.3.

Table 2 shows that `PAIR-o` substantially improves over IRMv1 as well as IRMX and yields top-ranking OOD performance among all state-of-the-art methods across different realistic distribution shifts, demonstrating the effectiveness and significance of resolving the optimization dilemma in OOD generalization. Besides, the advances of `PAIR` over IRMX also confirm the effectiveness of `PAIR-o` in finding a better trade-off between ERM and OOD objectives.

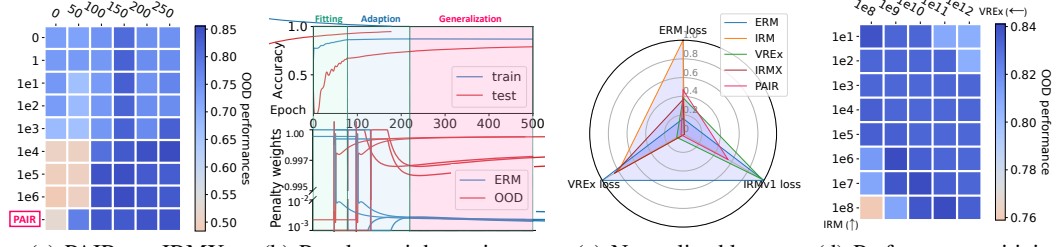

|                         |                         |                         |                            |
|-------------------------|-------------------------|-------------------------|----------------------------|
| (a) PAIR v.s. IRMX.     | (b) Penalty weights trajectory. | (c) Normalized losses. | (d) Preference sensitivity. |

Figure 5: (a) Each point is the *best* performed IRMX among corresponding pretraining epoch ($x$-axis), the IRMv1 penalty weights ($y$-axis) and *all* possible VREx penalty weights. Despite the substantial tunning efforts, IRMX performs no better than PAIR. That is because (b) PAIR can adaptively adjust the penalty weights during the optimization process, and leads to a (c) Pareto optimal solution. (d) The robustness of PAIR-o to different preference choices enables it adaptable to various scenarios.

**How can PAIR-o mitigate the objective conflicts?** We conduct ablation studies with the modified COLOREDMNIST (More details and results are given in Appendix F.2). First, as shown in Fig. 5(a), PAIR-o effectively finds a better solution than exhaustive tuning of penalty weights in IRMX. That is because PAIR can adaptively adjust the penalty weights (Fig. 5(b)), which leads to a Pareto optimal solution that has lower OOD losses while not compromising the ERM loss too much (Fig. 5(c)). The other reason is that, PAIR-o is generally robust to different choices of preference choices (Fig. 5(d)), which makes it adaptable to various scenarios, confirming our discussions in Sec. 4.2.

**Can PAIR-s effectively select better OOD solutions under realistic distribution shifts?** To verify the effectiveness of PAIR-s, we apply PAIR-s to multiple representative OOD methods as discussed in Sec. 2, and examine whether PAIR-s can improve the model selections under rigorous hyperparameters tunning (Gulrajani & Lopez-Paz, 2021) on COLOREDMNIST (Kamath et al., 2021), PACS (Li et al., 2017) and TERRAINCOGNITA (Beery et al., 2018). Intuitively, models selected merely based on ERM performance tend to have a high preference or better performance on environments that have a similar distribution of the corresponding validation set, which will lead to higher variance of performances at different environments or a lower worst environment performance. Hence we use training-domain validation accuracy for COLOREDMNIST and TERRAINCOGNITA, and test-domain validation accuracy for PACS to validate the existence of this issue under different scenarios (Teney et al., 2021). More details and results are provided in Appendix G.

Table 3: OOD generalization performances using DOMAINBED evaluation protocol.

|          | PAIR-s | COLOREDMNIST [†] |        |        |        | PACS [‡] |        |        |        |        | TERRAINCOGNITA [†] |        |        |        |        |
|----------|--------|-------|-------|-------|-------|------|------|------|------|-------|------|------|------|------|-------|
|          |        | +90%  | +80%  | 10%   | Δ wr. | A    | C    | P    | S    | Δ wr. | L100 | L38  | L43  | L46  | Δ wr. |
| ERM      |        | 71.0  | **73.4** | 10.0  |       | 87.2 | 79.5 | 95.5 | 76.9 |       | 46.7 | **41.8** | 57.4 | 39.7 |       |
| DANN     |        | 71.0  | **73.4** | 10.0  |       | 86.5 | 79.9 | 97.1 | 75.3 |       | 46.1 | 41.2 | 56.7 | 35.6 |       |
| DANN     | ✓      | 71.6  | 73.3  | 10.9  | +0.9  | 87.0 | 81.4 | 96.8 | 77.5 | +2.2  | 43.1 | 41.1 | 55.2 | 38.7 | +3.1  |
| GroupDRO |        | 72.6  | 73.1  | 9.9   |       | 87.7 | 82.1 | 98.0 | 79.6 |       | 48.4 | 40.3 | 57.9 | 40.0 |       |
| GroupDRO | ✓      | **72.7** | 73.2  | 13.0  | +3.1  | 86.7 | **83.2** | **97.8** | 81.4 | +1.8  | 48.4 | 40.3 | 57.9 | 40.0 | +0.0  |
| IRMv1    |        | 72.3  | 72.6  | 9.9   |       | 82.3 | 80.8 | 95.8 | 78.9 |       | 48.4 | 35.6 | 55.4 | 40.1 |       |
| IRMv1    | ✓      | 67.4  | 64.8  | **24.2** | +14.3 | 85.3 | 81.7 | 97.4 | 79.7 | +0.8  | 40.4 | 38.3 | 48.8 | 37.0 | +1.4  |
| Fishr    |        | 72.2  | 73.1  | 9.9   |       | **88.4** | 82.2 | 97.7 | 81.6 |       | 49.2 | 40.6 | 57.9 | 40.4 |       |
| Fishr    | ✓      | 69.1  | 70.9  | 22.6  | +12.7 | 87.4 | 82.6 | 97.5 | **82.2** | +0.6  | **51.0** | 40.7 | **58.2** | **40.8** | +0.3  |

[†]Using the training domain validation accuracy. [‡]Using the test domain validation accuracy.

Table 3 shows that there is a high variance in the performances at different environments of the models selected only based on the validation accuracy. In contrast, by jointly considering and trading off the ERM and OOD performances in model selection, PAIR-s substantially mitigates the variance by improving the worst environment performance of all methods under all setups up to $10\%$. It could serve as strong evidence for the importance of considering ERM and OOD trade-offs.

## 6 CONCLUSION

In this work, we provided a new understanding of optimization dilemma in OOD generalization from the MOO perspective, and attributed the failures of OOD optimization to the compromised robustness of relaxed OOD objectives and the unreliable optimization scheme. We highlighted the importance of trading off the ERM and OOD objectives and proposed a new optimizer PAIR-o and a new model selection criteria PAIR-s to mitigate the dilemma. We provided extensive theoretical and empirical evidence to show the necessity and significance of properly handling the ERM and OOD trade-offs.

## ACKNOWLEDGEMENTS

We thank the reviewers for their valuable comments. This work was supported by CUHK direct grant 4055146. YZ and BH were supported by the NSFC Young Scientists Fund No. 62006202, Guangdong Basic and Applied Basic Research Foundation No. 2022A1515011652, RGC Early Career Scheme No. 22200720, and Tencent AI Lab Rhino-Bird Gift Fund.

## ETHICS STATEMENT

Considering the wide applications and high sensitivity of deep neural networks to distribution shifts and spurious correlations, it is important to develop new methods that are able to generalize to OOD data, especially for some human-centered AI scenarios such as autopilot and social welfare. By understanding and mitigating the optimization dilemma in OOD generalization, our work could serve as an initiate step towards a new foundation of optimization for OOD generalization, with the hope for building more trustworthy and AI systems to facilitate broader AI applications and social benefits. Besides, this paper does not raise any ethical concerns. This study does not involve any human subjects, practices to data set releases, potentially harmful insights, methodologies and applications, potential conflicts of interest and sponsorship, discrimination/bias/fairness concerns, privacy and security issues, legal compliance, and research integrity issues.

## REPRODUCIBILITY STATEMENT

To ensure the reproducibility of our theoretical results, we provide detailed proofs for our propositions and theorems in Appendix E. To ensure the reproducibility of our methods and experimental results, we provide detailed description of the IRM case in Appendix C.1, the algorithms D, and the experimental setting in Appendix F, in addition to the main text. Besides, we will further provide a link to an anonymous repository that contains the source codes for reproducing the results in our paper during the discussion phase.

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

# Appendix of "Pareto Invariant Risk Minimization"

CONTENTS

# A  NOTATIONS

We first list the notations for key concepts in our paper.

Table 4: Notations

| | |
|---|---|
| $\mathcal{X} = \mathbb{R}^n$ | the input space |
| $\mathcal{Y} = \mathbb{R}$ | the label space |
| $\mathcal{Z} = \mathbb{R}^d$ | the latent space |
| $\varphi$ | the featurizer $\varphi : \mathcal{X} \to \mathcal{Z}$ learns a latent representation for each input example |
| $w$ | the classifier $w : \mathcal{Z} \to \mathcal{Y}$ |
| $f \in \mathcal{F}$ | the predictor $f = w \circ \varphi : \mathcal{X} \to \mathcal{Y}$ is composed of a featurizer and classifier when $w$ is linear, $f$ can be simply represented via dot product $w \cdot \varphi$ |
| $\mathcal{E}_{\text{all}}$ | the set of indices for all environments |
| $\mathcal{E}_{\text{tr}}$ | the subset of indices of training environments |
| $e$ | the index set of a specific environment |
| $\mathcal{D}^e, \mathcal{D}_e$ | the dataset from environment $e$, containing samples $\{X_i^e, Y_i^e\}$ considered as i.i.d. from $\mathbb{P}^e$ |
| $\mathcal{D}$ | the overall dataset containing data from all environments, $\mathcal{D} = \{\mathcal{D}^e\}_{e \in \mathcal{E}_{\text{all}}}$ |
| $\mathcal{I}(\mathcal{E})$ | the set of invariant predictors w.r.t. some OOD objectives (e.g., IRM) and environments $\mathcal{E}$ |
| $\mathcal{L}_e$ | the empirical risk calculated based on $\mathcal{D}^e$, e.g., square loss or logistic loss |
| $\boldsymbol{L}$ | the vector of losses $\{\mathcal{L}_i\}_{i=1}^m$ considered in $m$ objectives from a MOO problem, shared a set of parameters $\theta$ |
| $\mathcal{P}(\boldsymbol{L})$ | the set of Pareto optimal solutions w.r.t. the objectives $\boldsymbol{L}$ |
| $\boldsymbol{p}_{\text{ood}}$ | the vector of objective preference |
| $\boldsymbol{G} \in \mathbb{R}^{m \times d}$ | the matrix of gradients w.r.t. $m$ objectives $\boldsymbol{L}$ and parameters $\theta \in \mathbb{R}^d$ each objective $\mathcal{L}_i$ corresponds to a gradient vector $\boldsymbol{g} \in \mathbb{R}^d$ |
| $\mathcal{S}^{m+1}$ | the $m$-simplex corresponding to $m$ OOD objectives, $\{\beta \in \mathbb{R}_+^{m+1} \vert \sum_{i=1}^{m+1} \beta_i = 1\}$ |

# B  MORE DISCUSSIONS ON BACKGROUND AND FUTURE DIRECTIONS

## B.1  BACKGROUND AND RELATED WORK

In this section, we provide more details of the backgrounds and closely related works to ours, in complementary to Sec. 2.

**The problem of OOD generalization.** The problem of OOD generalization typically considers a supervised learning setting based on the data $\mathcal{D} = \{\mathcal{D}^e\}_{e \in \mathcal{E}_{\text{all}}}$ collected from multiple causally related environments $\mathcal{E}_{\text{all}}$, where a subset of samples $\mathcal{D}^e = \{X_i^e, Y_i^e\}$ from a single environment $e \in \mathcal{E}_{\text{all}}$ are drawn independently from an identical distribution $\mathbb{P}^e$ (Peters et al., 2016). Given the data from training environments $\{\mathcal{D}^e\}_{e \in \mathcal{E}_{\text{tr}}}$, the goal of OOD generalization is to find a predictor $f : \mathcal{X} \to \mathcal{Y}$ that generalizes well to all (unseen) environments, i.e., to minimize $\max_{e \in \mathcal{E}_{\text{all}}} \mathcal{L}_e(f)$, where $\mathcal{L}_e$ is the empirical risk (Vapnik, 1991) under environment $e$, $\mathcal{X}$ and $\mathcal{Y}$ are the input and labeling spaces, respectively. The predictor $f = w \circ \varphi$ is usually composed of a featurizer $\varphi : \mathcal{X} \to \mathcal{Z}$ that learns to extract useful features, and a classifier $w : \mathcal{Z} \to \mathcal{Y}$ that makes predictions from the extracted features. In practice, $\varphi$ is commonly implemented as a deep feature extractor, while $w$ is generically implemented as a simple dense linear classifier (Gulrajani & Lopez-Paz, 2021; Koh et al., 2021; Rame et al., 2021; Rosenfeld et al., 2022).

**Existing solutions to OOD generalization.** There exists a rich literature aiming to overcome the OOD generalization challenge, which usually appear as *additional regularizations* of ERM (Vapnik, 1991). The first line is the Domain Generalization works (Ganin et al., 2016; Sun & Saenko, 2016; Li

et al., 2018; Dou et al., 2019) that tries to regularize the learned features to be **domain-invariant**. However, Zhao et al. (2019) show that the domain invariant features solely are not sufficient for guaranteed good OOD generalization. We refer readers to Gulrajani & Lopez-Paz (2021) for more details of the literature about Domain Generalization. Moreover, Namkoong & Duchi (2016); Hu et al. (2018); Sagawa* et al. (2020) aim to regularize the models to be **robust to mild distributional perturbations** of the training distributions such that the models are expected to perform well in unseen test environments. Following the line of distributional robustness, Liu et al. (2021); Zhang et al. (2022b); Yao et al. (2022) further propose advanced strategies to improve the robustness by assuming that models trained with ERM have strong reliance to spurious features.

Recently there is increasing interest in adopt theory of causality (Pearl, 2009; Peters et al., 2017; Schölkopf et al., 2021) and introduce the **causal invariance** to the learned representations (Peters et al., 2016; Rojas-Carulla et al., 2018; Arjovsky et al., 2019). The causal invariance is inspired by the assumption of Independent Causal Mechanism (ICM) in causality (Peters et al., 2017). ICM assumes that conditional distribution of each variable given its causes (i.e., its mechanism) does not inform or influence the other conditional distributions (Pearl, 2009; Peters et al., 2017). Peters et al. (2016) introduce the concept of environments which are generated by different interventions on certain variables involved in the underlying data generation process of $(X, Y)$. Despite of the changes to the intervened variables, the conditional distribution of intervened variables (they usually are the direct parents of $Y$ in the underlying causal graph) and $Y$ is invariant. Therefore, the invariant relationship can be leveraged to predict $Y$ and generalize to different environments. We refer interested readers to Peters et al. (2016); Schölkopf et al. (2021); Ahuja et al. (2021a) for more details. Inspired by the causal invariance principle, Arjovsky et al. (2019) propose the framework of Invariant Risk Minimization (IRM) that allows the adoption of the causal invariance in neural networks. It further inspires plentiful invariant learning works (Parascandolo et al., 2021; Mahajan et al., 2021; Creager et al., 2021; Wald et al., 2021; Ahuja et al., 2021a; Chen et al., 2022b; Lin et al., 2022b). At the heart of these works is the intuition that: When a predictor $w$ acting on $\varphi$ minimizes the risks in all of the environments simultaneously, $\varphi$ is expected to discard the spurious signals while keeping the causally invariant signals. Additionally, there can be more definitions and implementations of the invariance (Koyama & Yamaguchi, 2020; Krueger et al., 2021; Shi et al., 2022; Rame et al., 2021) which further encourage **agreements** at various levels across different environments. We refer interested readers to Rame et al. (2021) for a detailed comparison and discussion. As shown that most of the existing approaches encounter the optimization dilemma when learning the causal invariance, this work mainly focus on resolving the optimization issue in learning the causal invariance defined by the framework of Invariant Risk Minimization (Arjovsky et al., 2019), which is different from the literature of IRM variants or other OOD objectives that focus on proposing better objectives to learn the causal invariance.

**Optimization Dilemma in OOD Algorithms.** Along with the developments of OOD methods, the optimization dilemma in OOD generalization is gradually perceived in the literature, and raises new puzzles to the community. In fact, several recent works also notice the optimization dilemma in OOD algorithms, specifically, the trade-off between discovering the statistical correlations (i.e., ERM) and preventing the usage of spurious correlations (e.g., IRM). Empirically, Gulrajani & Lopez-Paz (2021) observe that, with careful hyperparameter tuning and evaluation setting, many OOD algorithms cannot outperform ERM in domain generalization, demonstrating the difficulties of properly mitigating the trade-offs between OOD and ERM objectives in practice. Moreover, Sagawa* et al. (2020); Zhai et al. (2022) find that, regularization on ERM, or sacrificing ERM performance, is usually needed for achieving satisfactory OOD performance. A similar phenomenon has also been observed by Zhao et al. (2020); Xie et al. (2021); Sadeghi et al. (2022); Sener & Koltun (2022); Teney et al. (2022), which aligns with our findings through Pareto front as shown in Fig. 6(a) and Fig. 7(a). Besides, Lin et al. (2022a) find that IRM can easily overfit and learns unexpected features when applying IRM on large neural networks. Zhou et al. (2022) propose to alleviate this problem by imposing sparsity constraints. Orthogonal to Lin et al. (2022a); Zhou et al. (2022) that focuses on the optimization consequences, we focus on the optimization process of OOD objectives. In addition, Zhang et al. (2022a) find that, the performance of OOD algorithms largely relies on choosing proper pretraining epochs which aligns with our findings in Fig. 1(d), hence propose to construct a ready-to-use features for stable OOD generalization performance. Orthogonal to Zhang et al. (2022a), we focus on developing better optimization scheme for OOD algorithms, including choosing the proper objectives and the achievability of the invariant predictors. Besides, Lv et al. (2021) propose ParetoDA to leverage MOO to resolve the gradient conflicts amon the objectives

in Domain Adaption. ParetoDA uses the guidance of validation loss based on the data that has the identical distribution to test distribution, to trade-off the conflicts in domain adaption objectives. However, there can be multiple test domains, and the data that has identical distribution with the test domain is usually unavailable in OOD generalization. Therefore, ParetoDA is unsuitable for general OOD generalization methods. Despite the increasing literature that perceives the OOD optimization dilemma, it remains an open problem on why there exists such a dilemma, and how to effectively mitigate the conflicts of ERM and OOD objectives and obtain a OOD generalizable solution.

**Further implications by the OOD optimization dilemma.** In addition to preventing finding a proper OOD solution, the OOD optimization dilemma also raises significant challenges for model selection of OOD algorithms. Gulrajani & Lopez-Paz (2021) highlight this challenge with rigorous evaluation of OOD algorithms. Similar to PAIR-o, PAIR-s resolves the dilemma by leveraging the OOD loss values and explicitly considering the trade-offs of ERM and OOD performance. We present more details in Sec. G.1.

**Multi-Objective Optimization (MOO) and its applications in Multi-Task Learning.** MOO considers solving $m$ objectives, w.r.t. $\{\mathcal{L}_i\}_{i=1}^m$ losses, i.e., $\min_\theta \boldsymbol{L}(\theta) = (\mathcal{L}_1(\theta), ..., \mathcal{L}_m(\theta))^T$ (Kaisa, 1999). A solution $\theta$ dominates another $\bar{\theta}$, i.e., $\boldsymbol{L}(\theta) \preceq \boldsymbol{L}(\bar{\theta})$, if $\mathcal{L}_i(\theta) \leq \mathcal{L}_i(\bar{\theta})$ for all $i$ and $\boldsymbol{L}(\theta) \neq \boldsymbol{L}(\bar{\theta})$. A solution $\theta^*$ is called **Pareto optimal** if there exists no other solution that dominates $\theta^*$. The set of Pareto optimal solutions is called Pareto set, denoted as $\mathcal{P}$, and its image is called **Pareto front**. As it is usual that we cannot find a global optimal solution for all objectives in practice, hence Pareto optimal solutions are of particular value. The multiple-gradient descent algorithm (MGDA) is one of the commonly used approaches to efficiently find the Pareto optimal solutions (Désidéri, 2012) but limited to low-dimensional data. Sener & Koltun (2018) then resolve the issue and apply MGDA to high-dimensional multi-task learning scenarios, where the objective conflicts may degenerate the performance when using linear scalarization. As pure MGDA cannot find a Pareto optimal solution specified by certain objective preferences, Lin et al. (2019); Zhang & Golovin (2020); Ma et al. (2020) propose efficient methods to explore the Pareto set. Mahapatra & Rajan (2020) propose EPO to find the exact Pareto optimal solution with the specified objective preferences. Although MOO has gained success in mitigating the task conflicts in multi-task learning, it remains underexplored on whether and how we can leverage the MOO to model and resolve the ERM and OOD conflicts. Without a proper set of objectives and preference guidance, the existing MOO solvers are unable to obtain a desired solution for OOD generalization.

## B.2 LIMITATIONS AND FUTURE DIRECTIONS

Although PAIR is shown to effectively mitigate the objective conflicts and boost the OOD performance via better optimization and model selection, the performance gain sometimes can decrease given the limitations of PAIR. We believe future works can be built upon resolving the limitations of PAIR, as detailed below.

From the optimizer perspective, the improvements of PAIR-o can decrease on some datasets. We hypothesize it is because of the inevitable stochastic gradient bias in all MGDA MOO solvers (Liu & Vicente, 2021), and potentially large variance in estimating the IRMv1 penalties (e.g., RxRx1 where both IRMv1 and VREx are shown to perform poor ), as we discussed in Appendix D.4.2.

For PAIR-s, as discussed in Sec. 4 that PAIR-s can mitigate the drawbacks of selecting models using an unreliable validation set (has a large gap from the test domain), the improvements will be a bit smaller when the gaps narrow down (e.g., PACS using test domain validation accuracy). Besides, the estimation of satisfaction to Pareto optimality in PAIR-s can also be affected by the variances in estimating loss values in stochastic setting (e.g., TERRAINCOGNITA), as discussed in Appendix D.2.

Additionally, PAIR can also be applied to scenarios where gradient conflicts exist, such as the tradeoff between adversarial power and unnoticeability of the attacks (Chen et al., 2022a), as well as improving the quality of representations in contrastive learning (Ma et al., 2021).

## C MORE DETAILS ON IRM FAILURES AND FIX

In this section, we provide more details about the failure case of IRM and its effective fix from the perspective of MOO, in complementary to Sec. 3.

### C.1 MORE DETAIL ABOUT FAILURE CASE OF IRM

We follow Kamath et al. (2021) to discuss the failure case of IRM. Specifically, given the problem setup as in Sec. B.1, we are interested in the linear classification/regression following the setting. The loss values are measured as population loss in each environment.

**Setting A (identical to (Kamath et al. (2021))):** $\hat{\mathcal{Y}} = \mathbb{R}, \mathcal{Y} \subseteq \mathbb{R}$, $\ell$ is either the square loss $\ell_{\text{sq}}(\hat{y}, y) := \frac{1}{2}(\hat{y} - y)^2$, or the logistic loss $\ell_{\log}(\hat{y}, y) := \log\left(1 + \exp\left(-\hat{y}y\right)\right)$ when $\mathcal{Y} = \{-1, 1\}$ (binary classification).

IRM approaches the problem by finding an invariant representation $\varphi : \mathcal{X} \to \mathcal{Z}$, such that there exists a predictor $w : \mathcal{Z} \to \mathcal{Y}$ acting on $\varphi$ that is simultaneously optimal among $\mathcal{E}_{\text{all}}$. Hence, IRM leads to a challenging bi-level optimization problem (Arjovsky et al., 2019) as

$$\min_{w,\varphi} \sum_{e \in \mathcal{E}_{\text{tr}}} \mathcal{L}_e(w \circ \varphi), \tag{10}$$
$$\text{s.t. } w \in \arg\min_{\bar{w}:\mathcal{Z} \to \mathcal{Y}} \mathcal{L}_e(\bar{w} \circ \varphi), \ \forall e \in \mathcal{E}_{\text{tr}}.$$

Given the training environments $\mathcal{E}_{\text{tr}}$, and functional spaces $\mathcal{W}$ for $w$ and $\Phi$ for $\varphi$, predictors $w \circ \varphi$ satisfying the constraint are called invariant predictors, denoted as $\mathcal{I}(\mathcal{E}_{\text{tr}})$. When solving Eq. 10, characterizing $\mathcal{I}(\mathcal{E}_{\text{tr}})$ is particularly difficult in practice, given the access only to finite samples from a small subset of environments. It is natural to introduce a restriction that $\mathcal{W}$ is the space of linear functions on $\mathcal{Z} = \mathbb{R}^d$ (Jacot et al., 2021). Furthermore, Arjovsky et al. (2019) argue that linear predictors actually do not provide additional representation power than *scalar* predictors, i.e., $d = 1, \mathcal{W} = \mathcal{S} = \mathbb{R}^1$. The scalar restriction on $\mathcal{W}$ elicits a practical variant $\text{IRM}_\mathcal{S}$ as

$$\min_\varphi \sum_{e \in \mathcal{E}_{\text{tr}}} \mathcal{L}_e(\varphi), \text{s.t. } \nabla_{w|w=1} \mathcal{L}_e(w \cdot \varphi) = 0, \ \forall e \in \mathcal{E}_{\text{tr}}. \tag{11}$$

Let $\mathcal{I}_\mathcal{S}(\mathcal{E}_{\text{tr}})$ denote the set of invariant predictors elicited by the relaxed constraint in $\text{IRM}_\mathcal{S}$. It follows that $\mathcal{I}(\mathcal{E}_{\text{tr}}) \subseteq \mathcal{I}_\mathcal{S}(\mathcal{E}_{\text{tr}})$ (Kamath et al., 2021). Yet, Eq. 11 remains a constrained programming. Hence, Arjovsky et al. (2019) introduce a soft-constrained variant IRMv1 as

$$\min_\varphi \sum_{e \in \mathcal{E}_{\text{tr}}} \mathcal{L}_e(\varphi) + \lambda |\nabla_{w|w=1} \mathcal{L}_e(w \cdot \varphi)|^2. \tag{12}$$

**Theoretical Failure of Practical IRM Variants.** Although the practical variants seem promising, Kamath et al. (2021) show there exists huge gaps between the variants and the original IRM such that both $\text{IRM}_\mathcal{S}$ and IRMv1 can fail to capture the desired invariance, even being given the *population loss* and *infinite* amount of training environments. The failure case, called two-bit environment (Kamath et al., 2021), follows the setup of ColoredMNIST in IRM (Arjovsky et al., 2019), and defines environments with two parameters $\alpha_e, \beta_e \in [0, 1]$. Each $\mathcal{D}_e$ is defined as

$$Y := \text{Rad}(0.5), X_1 := Y \cdot \text{Rad}(\alpha_e), X_2 := Y \cdot \text{Rad}(\beta_e), \tag{13}$$

where $\text{Rad}(\sigma)$ is a random variable taking value $-1$ with probability $\sigma$ and $+1$ with probability $1 - \sigma$. We denote an environment $e$ with $(\alpha_e, \beta_e)$ for simplicity. The setup in IRM can be denoted as $\mathcal{E}_\alpha = \{(\alpha, \beta_e) : 0 < \beta_e < 1\}$ where $X_1$ is the invariant feature as $\alpha$ is fixed for different $e$.

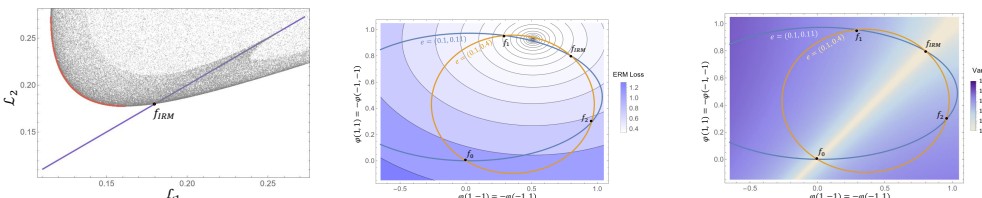

(a) Pareto Front under MSE loss.  (b) Failure case under MSE loss.  (c) Variance distribution under MSE loss.

Figure 6: Counterparts of Fig. 1(a), Fig. 3 and Fig. 2 implemented in MSE loss.

In the example given by Arjovsky et al. (2019), i.e., $\mathcal{E}_{\text{tr}} := \{(0.25, 0.1), (0.25, 0.2)\}$, $\text{IRM}_\mathcal{S}$ and IRMv1 are shown to be able to learn the invariant predictor $f_{\text{IRM}}$ as the original IRM despite of the relaxation. However, due to $\mathcal{I}(\mathcal{E}_{\text{tr}}) \subseteq \mathcal{I}_\mathcal{S}(\mathcal{E}_{\text{tr}})$, Kamath et al. (2021) show that the set of "invariant predictors" produced by $\text{IRM}_\mathcal{S}$ and IRMv1 is broader than our intuitive sense. For example, when given $\mathcal{E}_{\text{tr}} := \{(0.1, 0.11), (0.1, 0.4)\}$, the solutions satisfying the constraint in $\text{IRM}_\mathcal{S}$ are those intersected points in Fig. 1(a) (The ellipsoids are the constraints). Although $f_0, f_1, f_2, f_{\text{IRM}} \in \mathcal{I}_\mathcal{S}(\mathcal{E}_{\text{tr}})$, both $\text{IRM}_\mathcal{S}$ and IRMv1 prefer $f_1$ instead of $f_{\text{IRM}}$ (the predictor elicited by the original IRM), as $f_1$ has the smallest ERM loss. In fact, Kamath et al. (2021) prove that, the failure can happen in a wide range of environments with $\alpha < 0.1464$ and $\alpha > 0.8356$, even being given *infinite* number of additional environments, under MSE loss. It follows that $\mathcal{I}(\mathcal{E}_{\text{tr}}) \subsetneq \mathcal{I}_\mathcal{S}(\mathcal{E}_{\text{tr}})$. In other words, the relaxation in $\text{IRM}_\mathcal{S}$ and IRMv1 will introduce additional "invariant predictors" which however do not satisfy the original IRM constraint. Both $\text{IRM}_\mathcal{S}$ and IRMv1 will prefer those "invariant predictors" when they have lower ERM loss than $f_{\text{IRM}}$, demonstrating the significant theoretical gap between the practical variants and the original IRM.

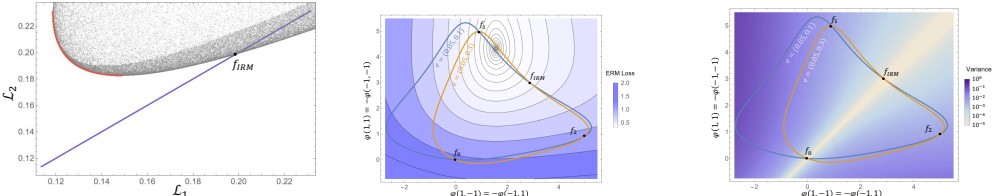

(a) Pareto Front under Logistic loss. (b) Failure case under Logistic loss. (c) Variance distribution under Logistic loss.

Figure 7: Counterparts of Fig. 1(a), Fig. 3 and Fig. 2 implemented in Logistic loss.

**More visualization results of the failure cases.** In the main paper, we visualize the Pareto front, ERM loss distribution, and the variance distribution of the failure case given MSE losses, given the environment setup of $\mathcal{E}_{\text{tr}} := \{(0.1, 0.11), (0.1, 0.4)\}$. We plot Fig. 1(a) and Fig. 3 based on the Mathematica code provided by Kamath et al. (2021), where we focus on the odd predictors due to the symmetry in two-bit environments, i.e., predictors satisfying $\varphi(1, -1) = -\varphi(-1, 1)$ and $\varphi(1, 1) = -\varphi(-1, -1)$. Since Fig. 1(a), Fig. 3 and Fig. 2 are implemented in MSE loss, for completing the discussion under Setting A (Kamath et al., 2021), we also give their logistic counterparts as in Fig. 7.

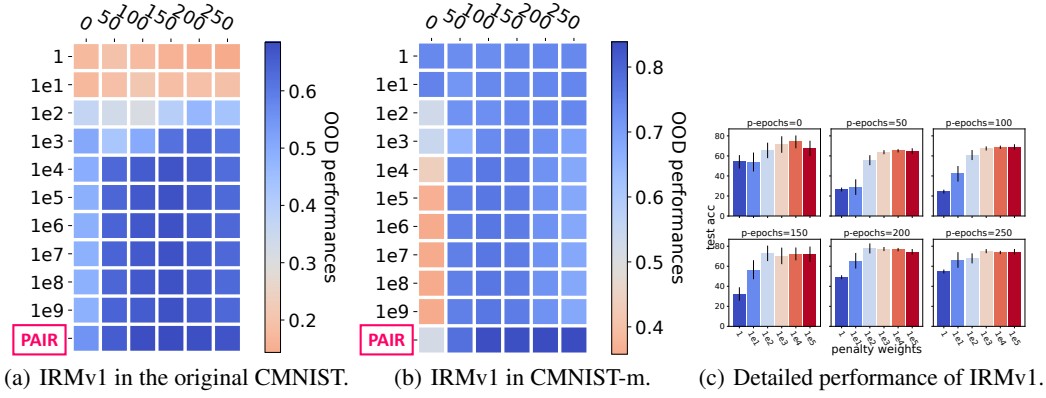

(a) IRMv1 in the original CMNIST. (b) IRMv1 in CMNIST-m. (c) Detailed performance of IRMv1.

Figure 8: Performances of IRMv1 in CMNIST and CMNIST-m under different hyperparameters.

**Practical Drawback of Practical IRM Variants.** In addition to the theoretical gap, the optimization of IRMv1 is also difficult due to the conflicts between the IRM penalty and ERM penalty in Eq. 12. It often requires significant efforts for choosing proper hyperparameters such as pretraining epochs and IRM penalty weights, i.e., $\lambda$. Otherwise, IRMv1 may not enforce the constraint in $\text{IRM}_\mathcal{S}$, hence will lead to unsatisfactory performance, as shown in Fig. 1(d). We argue that the gradient conflicts generally exist in OOD optimization for various objectives, in Fig. 1(b), we visualize the cosine similarity between the gradients produced by ERM and OOD objectives, which is averaged from 50

epochs after the pretraining. It can be found that, all of the OOD objectives (Arjovsky et al., 2019; Krueger et al., 2021; Ahuja et al., 2021a; Koyama & Yamaguchi, 2020; Rame et al., 2021; Wald et al., 2021; Pezeshki et al., 2021) tend to yield gradients that have a lower cosine similarity with those of ERM. The generally existed conflicts can further lead to suboptimal performances of these OOD objective in practice even with exhaustive parameter tunning.

In complementary to Fig. 1(d), we provide full results in Fig. 8, where we show the results of IRMv1 under different penalty weights ($y$-axis) and pretraining epochs ($x$-axis) on COLOREDMNIST (Arjovsky et al., 2019) (CMNIST) as well as the failure case (Kamath et al., 2021) (CMNIST-m), or $\mathcal{E}_{\mathrm{tr}} := \{(0.1, 0.2), (0.1, 0.25)\}$ described in two-bit environment. It can be found that the performances of IRMv1 are highly dependent on proper tuning of pretraining epochs and the penalty weights. The dependence grows stronger when IRMv1 is shown to be unrobust on CMNIST-m. We also provide a more detailed results of IRMv1 on CMNIST-m in Fig. 8(c), where the dependence can be clearly observed. In contrast, PAIR performs robustly well under different pretraining epochs, using a default preference $(1, 1e10, 1e12)$ to ERM, IRMv1 and VREx objectives, respectively. In Sec. 5, we provide more evidences to demonstrate the power of PAIR-o.

## C.2 DISCUSSIONS OF OBJECTIVES IN PAIR

In Sec. 3.2, we derive a group of ideal objectives for improving the robustness of IRMv1, shown as the following IRMX

$$\text{(IRMX)} \qquad \min_{\varphi}(\mathcal{L}_{\mathrm{ERM}}, \mathcal{L}_{\mathrm{IRM}}, \mathcal{L}_{\mathrm{VREx}})^T. \tag{14}$$

We prove in Proposition 2 that IRMX is able to solve a large number of failure cases of $\mathrm{IRM}_{\mathcal{S}}$ and IRMv1, and recovers the set of invariant predictors produced by the original IRM. However, motivated readers might be interested in the reasons for keeping IRMv1 in IRMX, since VREx solely could resolve the two-bit environment failure case.

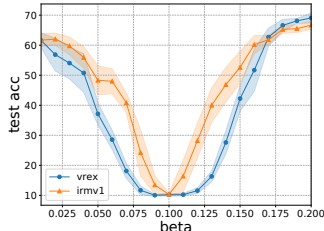

Figure 9: Drawbacks of V-REx in practice.

Theoretically, Proposition 2 requires also the invariant predictors produced by $\mathrm{IRM}_{\mathcal{S}}$, i.e., $\mathcal{I}_{\mathcal{S}}(\mathcal{E})$, to recover the invariant predictors yielded by IRM. Nevertheless, it considers only the ideal case. In the next, we elaborate on a detailed discussion from the empirical side.

**Drawbacks of Robust Minimization in Practice.** After showing REx (Krueger et al., 2021) can help avoiding the failure cases of $\mathrm{IRM}_{\mathcal{S}}$, a natural question is that, does $\mathcal{L}_{\mathrm{IRM}}$ remain necessary? We find the answer is "Yes". In Fig. 9, we use a modified example of $\mathcal{E}_{\mathrm{tr}} = \{(0.25, 0.1), (0.25, \beta)\}$ with ColoredMNIST (Arjovsky et al., 2019), where we change the variance between two environments through different $\beta$. It can be found that, as the variance between two environments getting closer, the performance of REx (Krueger et al., 2021) (denoted as vrex) drops more sharply than IRMv1 (denoted as irmv1). The main reason is that, as the variation of spurious signals in two environments tends to be smaller, the gradient signal of $\mathrm{var}(\{\mathcal{L}_e\}_{e \in \mathcal{E}_{\mathrm{tr}}})$ tends to vanish, while the signals from $\mathcal{L}_{\mathrm{IRM}}$ maintains. This issue can be more serious in stochastic gradient descent where the estimates of the variance of $\{\mathcal{L}_e\}_{e \in \mathcal{E}_{\mathrm{tr}}}$ in minibatches tend to be noisy, leading to weaker signals.

## C.3 MORE DETAILS ON THE EXTRAPOLATION EXAMPLE

In this section, we provide more details and results about the extrapolation example that examines the recovery of causal invariance, in complementary to Sec. 3.3.

We first restate the definition of causal invariance specified by Peters et al. (2016); Arjovsky et al. (2019); Kamath et al. (2021) as in Definition C.1.

**Definition C.1.** *(Causal Invariance) Given a predictor $f := w \circ \varphi$, the representation produced by the featurizer $\varphi$ is invariant over $\mathcal{E}_{all}$ if and only if for all $e_1, e_2 \in \mathcal{E}_{all}$, it holds that*

$$\mathbb{E}_{\mathcal{D}_{e_1}}[Y|\varphi(X) = z] = \mathbb{E}_{\mathcal{D}_{e_2}}[Y|\varphi(X) = z],$$

*for all $z \in \mathcal{Z}_{\varphi}^{e_1} \cap \mathcal{Z}_{\varphi}^{e_2}$, where $\mathcal{Z}_{\varphi}^e := \{\varphi(X)|(X, Y) \in supp(\mathcal{D}_e)\}$.*

Then, we construct a regression example from $\mathcal{X} : \mathbb{R}^2 \to \mathcal{Y} : \mathbb{R}$. The input $X$ is a two dimensional inputs, i.e., $X = (X_1, X_2)$. $X_1$ is designed to be the invariant feature, i.e., $Y = \sin(X_1) + 1$, while

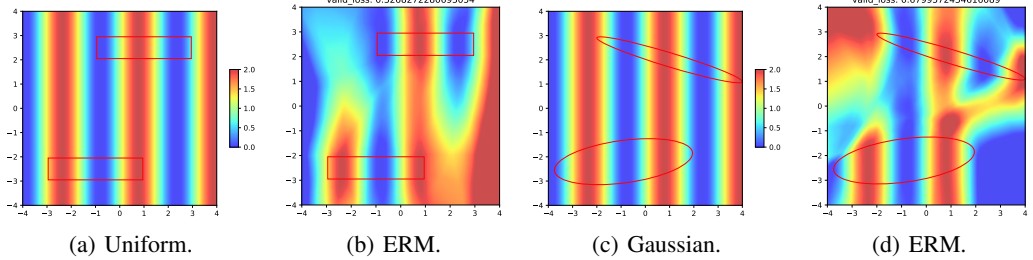

(a) Uniform.  (b) ERM.  (c) Gaussian.  (d) ERM.

Figure 10: Recovery of causal invariance via PAIR. (a), (c) We adopt two sampling methods where we sample the training data (mainly) from the regions marked in red, and evaluate the predictions across all region from $(-4, -4)$ to $(4, 4)$. The predictor following the invariance defined in IRM (Arjovsky et al., 2019) requires the predictions to be independent of spurious features within the overlapped invariant features. In this example, intuitively it requires the colored lines to be perpendicular to $x$-axis within $[-2, 2]$. (b) and (d) show the performances of ERM under two sampling methods, it can be found that ERM fail to recover the causal invariance and incurs a high MSE loss.

$X_2$ is designed to be the spurious feature that can be controlled to be spuriously correlated with label $Y$. The environments are synthesized according to different sampling methods.

Shown as in Fig. 10, we leverage two sampling methods: i) Uniform sampling and ii) Gaussian sampling, where the latter is more difficult than the former. For Uniform sampling, we uniformly sample the rectangle regions $\{(-3, -3), (-2, 1)\}$ as environment 1 and $\{(-1, 2), (3, 3)\}$ as environment 2, shown as the red regions marked in Fig. 10(a). For Gaussian sampling, we sample from two Gaussian distributions: the first one has the center as $(-0.9, -2.2)$ with the covariance matrix as $\{(0.9, 0.11), (0.11, 0.1)\}$; the second one has the center as $(1, 2)$ with the covariance matrix as $\{(1, -0.3), (-0.3, 0.1)\}$, shown as the red regions marked in Fig. 10(c).

Therefore, in these two examples, the invariant representation $\varphi$ should only take $X_1$ and discard the spurious features $X_2$ under the overlapped invariant features, i.e., $[-2, 2]$. As we use different colors to denote, the prediction produced by the invariant predictor following Definition C.1 is expected be independent of $X_2$. In other words, the plotted lines need to be *perpendicular* to the $x$-axis within the overlapped invariant features $[-2, 2]$.

We implement the predictor with a 3-layer linear perceptron that has a hidden dimension of 128. We use the MSE loss and Adam (Kingma & Ba, 2015) to optimize the neural network. We sample 2500 training data points from each environment and evaluate with 1000 data points uniformly sampled across all regions. For fair comparison, we train all algorithms 10000 epochs until converge. Following the common practice (Gulrajani & Lopez-Paz, 2021), we use a anneal iterations of the OOD penalties for all methods as 150. For IRMv1, VREx and IRMX, we search the penalty weights from $1e-4$ to $1e$ and find they generically perform well when with the penalty weights of $1e-2$ to $1e1$. While for PAIR, we search the relative preferences across 6 choices $(1, 1e4, 1e16), (1, 1e4, 1e12), (1, 1e6, 1e8), (1, 1e8, 1e4), (1, 1e4, 1e4), (1, 1e8, 1e8)$, and find $(1, 1e4, 1e12), (1, 1e8, 1e4), (1, 1e4, 1e4), (1, 1e8, 1e8)$ have lower validation losses.

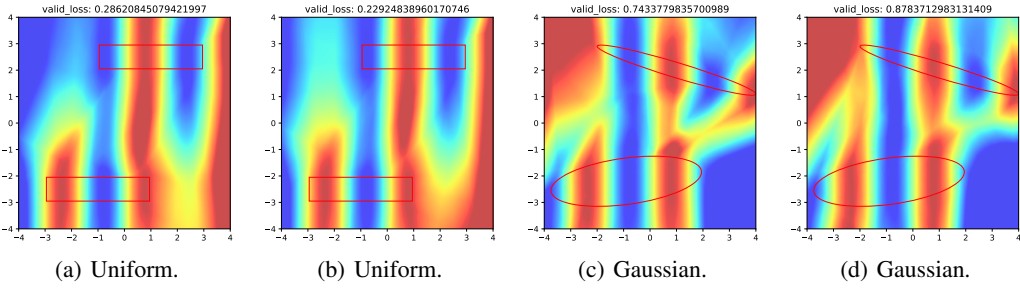

(a) Uniform.  (b) Uniform.  (c) Gaussian.  (d) Gaussian.

Figure 11: Recovery of causal invariance via IRMv1.

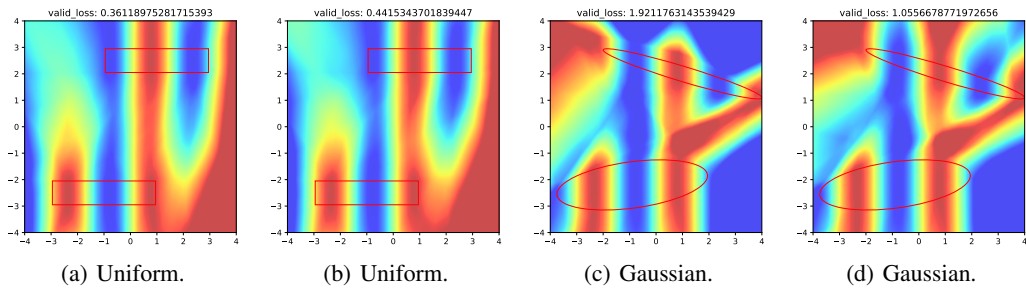

Figure 12: Recovery of causal invariance via VREx.

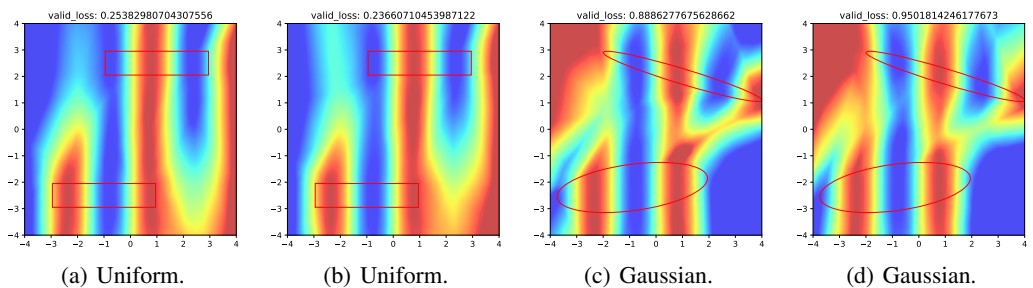

Figure 13: Recovery of causal invariance via IRMX.

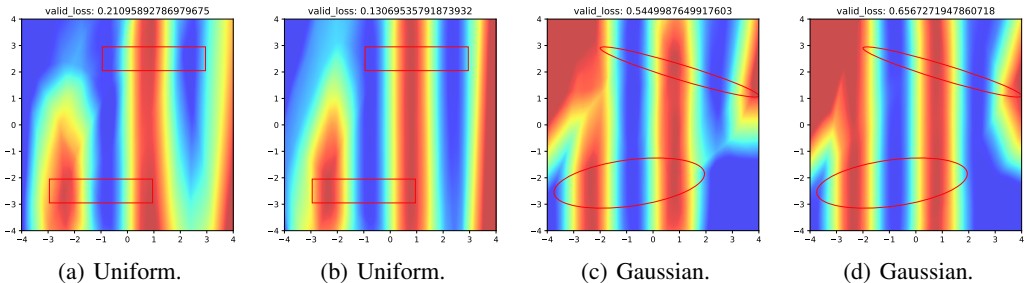

Figure 14: Recovery of causal invariance via PAIR.

We plot predictions with the best MSE losses of IRMv1, VREx, IRMX and `PAIR` in Fig. 11, Fig. 12, Fig. 13, and Fig. 14 respectively. We also plot the validation loss at the top of the image while *it does not necessarily indicate a better recovery of causal invariance*. It can be found that, when given the uniform sampled environments, the unrobust IRMv1, VREx and IRMX can recover part of the causal invariance, while when switching to the Gaussian sampled environments, they can fail dramatically as expected. In contrast, for both uniform sampling and Gaussian sampling, `PAIR` manage to recover the causal invariance almost perfectly. Perhaps even more surprisingly, `PAIR` achieve a lower extrapolation loss up to 0.06 and 0.32, which are essentially beyond the extrapolation requirement issued by the causal invariance. Hence we believe it is an interesting and promising future direction to probe the extrapolation ability within and beyond causal invariance.

## D  MORE DETAILS ON THE IMPLEMENTATIONS OF `PAIR`

In this section, we provide more details about the implementation of `PAIR` as a optimizer and a model selection criteria, in complementary to Sec. 4.1.

**Key takeaways from the IRM example.** Recall that the key takeaways from the failures of OOD optimization can be attributed to: i) using unrobust objectives for optimization; ii) using unreliable scheme to approach the desired solution. Nevertheless, we can improve the robustness of the OOD objectives by introducing additional guidance such that the desired solution can be relocated in the Pareto front w.r.t. to the new objectives. After obtaining robust objectives to optimize, we then leverage a preference-aware MOO solver to find the Pareto optimal solutions that maximally satisfy the invariance constraints by assigning the OOD objective a higher preference while being aware of retaining ERM performance.

More formally, let $f_{\text{ood}}$ be the desired OOD solution, a group of OOD objectives $\boldsymbol{L}_{\text{ood}} = \{\mathcal{L}_{\text{ood}}^i\}_{i=1}^m$ are robust if they satisfy that

$$\boldsymbol{L}_{\text{ood}}(f_{\text{ood}}) \preceq \boldsymbol{L}_{\text{ood}}(f), \forall f \neq f_{\text{ood}} \in \mathcal{F}, \tag{15}$$

where $\mathcal{F}$ denotes the functional class of possible predictors. When given a robust OOD objective $\boldsymbol{L}_{\text{ood}}$, our target is to solve the following MOO problem

$$\min_f (\mathcal{L}_{\text{ERM}}, \boldsymbol{L}_{\text{ood}})^T, \tag{16}$$

where $\boldsymbol{L}_{\text{ood}}$ corresponds to a $\epsilon_{\text{ood}}$-relaxed invariance constraint as $\boldsymbol{L}_{\text{ood}}(f_{\text{ood}}) = \epsilon_{\text{ood}} \preceq \boldsymbol{L}_{\text{ood}}(f), \forall f \neq f_{\text{ood}} \in \mathcal{F}$. Denote the $\epsilon_{\text{inv}}$ as empirical loss of using the underlying invariant features to predict labels, then the optimal values of the desired OOD solution are $(\epsilon_{\text{inv}}, \epsilon_{\text{ood}})^T = (\mathcal{L}_{\text{ERM}}(f_{\text{ood}}), \boldsymbol{L}_{\text{ood}}(f_{\text{ood}}))^T$, which corresponds to an ideal OOD preference for the objectives that is $\boldsymbol{p}_{\text{ood}} = (\frac{1}{\epsilon_{\text{inv}}}, \frac{1}{\epsilon_{\text{ood}}})^T$. Then the solution of Eq. 9 needs to maximally satisfy the OOD preference, i.e., maximize $\boldsymbol{L}(f)^T \boldsymbol{p}_{\text{ood}}$.

### D.1 DETAILED DESCRIPTION OF PAIR-o FOR OOD OPTIMIZATION

To find a Pareto optimal solution that satisfies the OOD preference $\boldsymbol{p}_{\text{ood}}$, we leverage the preference-aware MOO solver (Mahapatra & Rajan, 2020). Different from Mahapatra & Rajan (2020), we adopt an explicit 2-stage "descent" and "balance" scheme, following the common practice in OOD generalization (Gulrajani & Lopez-Paz, 2021).

Illustrated as in Fig. 15, in the "descent" phase, we train the model to minimize the ERM loss such that it approaches the Pareto front by merely minimizing $\mathcal{L}_{\text{ERM}}$ first. Then, in the "balance" phase, we adjust the solution to maximally satisfy the OOD preference $\boldsymbol{p}_{\text{ood}}$.

Meanwhile, to avoid divergence from the Pareto front, at each step, the descent direction $\boldsymbol{g}_{\text{des}}$ not only needs to maximize $\boldsymbol{L}(f)^T \boldsymbol{p}_{\text{ood}}$, but also needs to avoid ascending all the loss values. More formally, let $\boldsymbol{G}$ denote the gradient signals produced by $\boldsymbol{L}$, at step $t$ of the "balance" phase, it solves the following LP for the objective weights $\beta^*$,

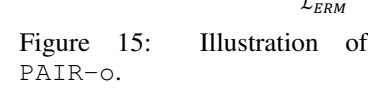

Figure 15: Illustration of PAIR-o.

$$\beta^* = \arg\max_{\beta \in \mathcal{S}^{m+1}} (\boldsymbol{G}\beta)^T \boldsymbol{g}_b,$$
$$\text{s.t.} \ (\boldsymbol{G}\beta)^T \boldsymbol{G}_j \geq \boldsymbol{g}_b^T \boldsymbol{G}_j, \ \forall j \in \bar{J} - J^*,$$
$$(\boldsymbol{G}\beta)^T \boldsymbol{G}_j \geq 0, \ \forall j \in J^*, \tag{17}$$

where $\mathcal{S}^{m+1} = \{\beta \in \mathbb{R}_+^{m+1} | \sum_{i=1}^{m+1} \beta_i = 1\}$, $\boldsymbol{g}_b$ is the adjustment direction that leads to the preferred Pareto optimal solution by $\boldsymbol{p}_{\text{ood}}$, $J = \{j | G_j^T \boldsymbol{g}_b > 0\}$ are the indices of objectives which donot conflict with $\boldsymbol{g}_b$ while $\bar{J} = \{j | G_j^T \boldsymbol{g}_b \leq 0\}$ are those have conflicts with $\boldsymbol{g}_b$, $J^* = \{j | L_j \boldsymbol{p}_{\text{ood}j} = \max_{j'} (L_{j'} \boldsymbol{p}_{\text{ood}j'})\}$ is the index of the objective which diverges from the preference most.

Specifically, Mahapatra & Rajan (2020) show that using the following $\boldsymbol{g}_b$ could provably lead the solution converge to the desired preferred Pareto optimal solution, which is defined as follows

$$\boldsymbol{g}_b = \boldsymbol{p} \odot (\log((m+1)\hat{\boldsymbol{L}}) - \mu(\boldsymbol{L})), \tag{18}$$

where $\odot$ is the element-wise product operator, $\mu(\boldsymbol{L})$ is the quantitative divergence of the current solution from the preferred direction, calculated through the losses at the current step, as follows

$$\mu(\boldsymbol{L}) = \text{KL}(\hat{\boldsymbol{L}} | \mathbf{1}/m) = \sum_i^{m+1} \hat{\boldsymbol{L}}_i \log(m\hat{\boldsymbol{L}}_i), \tag{19}$$

---

**Algorithm 1** Pseudo code for `PAIR-o`.

---

1: **Input:** Training data $\mathcal{D}_{\text{tr}} = \{X_i, Y_i\}_{i=1}^N$ with environment partitions $\mathcal{D}_{\text{tr}} = \{\mathcal{D}^e\}_{e \in \mathcal{E}_{\text{tr}}}$; learning rate $\eta$; batch size $b$; number of sampled environments $d$; OOD preference $\boldsymbol{p}_{\text{ood}}$ for ERM loss $\mathcal{L}_{\text{ERM}}$ and $m$ OOD losses $\boldsymbol{L}_{\text{ood}} = \{\mathcal{L}_{\text{ood}}^i\}_{i=1}^m$; pre-training epochs $e_p$; maximum training epochs for "balance" phase $e_b$; Trainable parameters at the "balance" phase $\theta$;
2: Randomly initialize parameters in the model $f = w \circ \varphi$;
3: **for** $i = 1$ to $e_p$ **do**
4:  Sample batches of data $\{X_j, Y_j\}_{j=1}^b$;
5:  Make predictions with $f$: $\{\widehat{Y}_j\}_{j=1}^b = f(\{X_j\}_{j=1}^b)$;
6:  Calculate the empirical loss $L_{\text{ERM}}$ with $\{\widehat{Y}_j\}_{j=1}^b$;
7:  Update parameters of $f$ with the empirical loss $\mathcal{L}_{\text{ERM}}$ using the learning rate $\eta$;
8: **end for**
9: **for** $i = 1$ to $e_b$ **do**
10:  **for** $D^e \in \text{permute}(\{\mathcal{D}^e\}_e \in \mathcal{E}_{\text{tr}})$ **do**
11:   Sample a batch of the data from $D^e$, $\{X_j^e, Y_j^e\}_{j=1}^b \sim D^e$;
12:   Make predictions with $f$: $\{\widehat{Y}_j^e\}_{j=1}^b = f(\{X_j^e\}_{j=1}^b)$;
13:  **end for**
14:  Calculate empirical and OOD losses $\mathcal{L}_{\text{ERM}}$ and $\mathcal{L}_{\text{ood}}$ and obtain the overall losses $\boldsymbol{L}$;
15:  Obtain gradients $\boldsymbol{G} = \partial \boldsymbol{L}/\partial\theta$;
16:  Calculate the OOD divergence $\mu(\boldsymbol{L})$ using Eq. 19;
17:  Obtain the adjustment direction $\boldsymbol{g}_b$ using Eq. 18;
18:  Obtain the index sets $J, J^*, \bar{J}$ required by Eq. 17;
19:  Solve Eq. 19 for the loss weights $\beta^*$;
20:  Update parameters $\theta^{i+1} = \theta^i - \eta \boldsymbol{G}\beta^*$;
21: **end for**

---

where $\hat{\boldsymbol{L}}$ is the normalized loss as

$$\hat{\boldsymbol{L}}_i = \boldsymbol{p}_{\text{ood}_i}\boldsymbol{L}_i / \sum_{j}^{m+1} \boldsymbol{p}_j \boldsymbol{L}_j.$$

Then, we elaborate the detailed algorithm of `PAIR-o` implemented via the EPO solver (Mahapatra & Rajan, 2020) as in Algorithm 1.

We now state a informal version of the convergence guarantee.

**Theorem D.1.** *(Informal) Given $L_{ERM}$ along with $m$ differentiable OOD losses $\boldsymbol{L}_{ood}$, at each step in the "balance" phase (line 9 to line 21 in Algorithm 1), there exists a step size $\eta_0$ such that, the set of new loss values $\boldsymbol{L}^{(i+1)} = (L_{ERM}, L_i, ..., L_m)^T$ with the updated parameters $\theta^{(t+1)}$ by any $\eta \in [0, \eta_0]$, denoted as $\mathcal{A}^t$ has the following properties:*

*(i). $\mathcal{A}^t$ contains the exact Pareto optimal solution satisfying the OOD preference vector, i.e., $\boldsymbol{L}^* \in \mathcal{A}^t$;*

*(ii). $\mathcal{A}^t$ grows monotonically smaller and smaller.*

From (i) and (ii) in Theorem D.1, it suffices to know that as the optimization continues, $\mathcal{A}^t$ converges to the losses of the exact Pareto optimal solution, hence for the parameters. The proof for Theorem D.1 simply follows the Theorem 1 to Corollary 1 in Mahapatra & Rajan (2020). Note that `PAIR-o` provides a general framework to find a better OOD solution that properly trades off ERM and OOD objectives. In experiments, we find that using the simply modified variant of EPO solver (Mahapatra & Rajan, 2020) in `PAIR-o` can effectively find a descent path under the gradient conflicts that leads to a better OOD solution. Nevertheless, a more sophisticated preference-aware MOO solver can be developed and integrated into the framework of `PAIR-o`, which we believe is a promising future direction (Zhao & Zhang, 2015; Zhou et al., 2018; 2020).

## D.2 DETAILED DESCRIPTION OF `PAIR-s` FOR OOD MODEL SELECTION

In this section, we provide a detailed description of `PAIR-s` for OOD model selection for Sec. 4.1. Before start, we also provide a detailed description of the critical reasons for designing `PAIR-s`

in Appendix G.1. From the IRM example, it is obvious that traditional model selection methods that merely use validation performance, i.e., ERM performance, are not suitable to select a desired solution for OOD generalization. Otherwise, the OOD performance would be easily compromised due to its conflicts with ERM objective. This issue is more serious when the validation set has a large gap between the test set (cf. Training-domain validation set selection for COLOREDMNIST in Table 3). Intuitively, models selected merely based on ERM performance tend to have a high preference or better performance on environments that have a similar distribution of the corresponding validation set, which will lead to higher variance of performances at different environments or a lower worst environment performance. Therefore, it is natural to jointly consider the ERM and OOD performances in model selection. Specifically, the selected model is expected to maximally satisfy the exact Pareto optimality.

Since our focus of PAIR-s is mainly to validate the existence of previous mode selection issues, we simply incorporate the PAIR score as an additional model selection criteria. More specifically, given a OOD preference $p_{\text{ood}}$, we can calculate the PAIR selection score as

$$s_{\text{PAIR}} = \boldsymbol{L}^T \hat{\boldsymbol{p}}_{\text{ood}}, \tag{20}$$

where $\hat{p}_{\text{ood}}$ is the normalized OOD preference as $p_{\text{ood}}/\sum_{i=1}^{m+1} p_{\text{ood}_i}$. With the PAIR score, we then can apply it into the DOMAINBED model selection algorithms (Gulrajani & Lopez-Paz, 2021). Specifically, the model selection in DOMAINBED aims to select models from several rigorous hyperparameter trials according to the validation accuracy. For the model selection in each run, one can obtain all training domain validation accuracies but only one test domain validation accuracy for fairness.

The algorithm is detailed as in Algorithm 2. The PAIR score is mainly used to select models among the logged steps within one run. To avoid trivial cases, we expect the models participated into the selection are converged. To this end, we heuristically use a threshold $c$ to filter out the first $c$ steps and find it empirically effective. To select models from different runs, we will first use the validation accuracy to filter out some unreliable cases, and then adopt the PAIR to finalize the model selection. The only exception is the test domain validation accuracy, where the test domain validation accuracy is more likely to be a reliable indicator than the PAIR score.

The main limitation of the PAIR estimation is about the estimation of the loss values. In stochastic gradient descent, one could only obtain a stochastic estimate of loss values based on a minibatch sample of $\mathcal{D}_{\text{tr}}$. When the stochastic estimates of the loss values are unbiased, the PAIR is unbiased, too. However, there can exist certain variances in the stochastic estimates, which can severely affect the precision of the score thus the comparison of different models. Although Theorem E.1 establishes certain theoretical guarantees that allows for some degree of uncertainties, the variances are usually unavoidable. A instant fix for the issue is that one could afford some additional evaluation time to obtain a better estimate of the loss values. Besides, one could also jointly consider the uncertainty of the estimation and derive a more accurate model selection (Wald et al., 2021), which we leave for future work.

### D.3 DISCUSSION ON THE PRACTICAL CHOICES OF OOD PREFERENCE

Essentially, the performances of both PAIR-o and PAIR-s have certain dependence on the quality of the OOD preference $p_{\text{ood}}$, however, it is often the case that the ideal OOD preference is usually unknown. It is desirable to analyze the performances of PAIR-o and PAIR-s under a imprecise OOD preference. Mahapatra & Rajan (2020) discussed a bit that when the exact Pareto optimal solution under the preference does not exist, the EPO solver can still find a Pareto optimal solution that is closest to the preferred direction. We discuss it in a more general way by developing a new MOO formulation of Eq. 16 under a approximated preference up to some approximation error of $\epsilon$. The theoretical discussion can be found in Sec. E.2. In this section, we focus on the practical side of the choice of $p_{\text{ood}}$.

We first discuss some heuristics that can be leveraged to obtain a proper OOD preference under two scenarios:

(i). one has little-to-no knowledge about the OOD loss values;

(ii). one has the access to some running histories that one has some empirical knowledge about the OOD loss values;

---

**Algorithm 2** Pseudo code for `PAIR-s`.

---

1: **Input:** Running history $\mathcal{H}$ from $R$ runs, where each running history is consist of loss history $\mathcal{L} = \{\mathcal{L}_1^t, \mathcal{L}_2^t, ..., \mathcal{L}_{(m+1)}^t\}_{t=1}^T$ of $(m+1)$ losses, i.e., $\mathcal{L}_{\text{ERM}}$ and $\boldsymbol{L}_{\text{ood}} = \{\mathcal{L}_{\text{ood}}^i\}_{i=1}^m$, and training and validation accuracy history $\mathcal{A} = \{A_{\text{tr}}^t, A_{\text{val}}^t\}_{t=1}^T$, from $T$ logging steps; OOD preference $\boldsymbol{p}_{\text{ood}}$; Convergence step $c$; Validation accuracy percentile $p$;
2: **for** $r = 1$ **to** $R$ **do**
3:     Calculate PAIR score using $\boldsymbol{p}_{\text{ood}}$ for all $T$ steps as $\mathcal{S} = \{s^t\}_{t=1}^T$ using Eq. 20;
4:     Filter out the first $c$ steps to avoid trivial cases and get $\widehat{\mathcal{S}} = \{s^t\}_{t=c}^T$;
5:     Store the step with maximum PAIR score as $s_* = \arg\max_t \widehat{\mathcal{S}}$;
6: **end for**
7: Obtain the selected steps from $R$ runs as $\mathcal{S} = \{s_*^r\}_{r=1}^R$;
8: Obtain the validation accuracies for all selected steps $\mathcal{A}_{\text{val}} = \{A_{\text{val}}^{s_*^r}\}_{r=1}^R$;
9: Calculate the validation selection bar as $\bar{A}_{\text{val}} = (\max \mathcal{A}_{\text{val}} - \min \mathcal{A}_{\text{val}}) * p + \min \mathcal{A}_{\text{val}}$;
10: Filter out all runs that have a validation accuracy lower than $\bar{A}_{\text{val}}$ and obtain $\bar{\mathcal{H}}$;
11: Find the run with highest PAIR score as $r_* = \arg\max_{r \in \bar{\mathcal{H}}} s_*^r$;
12: Return associated history of $r_*$;

---

In practice, i) mostly fits to `PAIR-o` while ii) mostly fits to `PAIR-s`.

When **i)** one has little-to-no knowledge about the OOD loss values, one can leverage certain theoretical inductive biases about the OOD losses. In fact, it is usual the case that the theoretical conditions for the optimality of OOD objectives do not hold in practice (Ganin et al., 2016; Sagawa* et al., 2020; Krueger et al., 2021; Shi et al., 2022; Rame et al., 2021). In this case, minimizing the OOD losses acts more like a necessary condition for a satisfactory OOD solution. Therefore, one could assign a sufficiently larger preference to OOD objectives than ERM objective. For example, throughout all experiments in the paper, we mostly assign $(1, 1e10, 1e12)$ to ERM, IRMv1, and VREx losses, which works under many scenarios.

Besides, among different OOD objectives, one could easily know which is more likely to be optimized than another. Therefore, to ensure all OOD losses are equally maximally optimized, we could assign the easily-optimizable OOD objectives higher preference. For example in IRMX, VREx tends to be easier to optimize than IRMv1 therefore we assign a higher preference to VREx. Moreover, if one could know the performances of different OOD objectives, it is natural to assign a higher preference to those which solely perform better.

When **ii)** one has the access to some running histories that one has some empirical knowledge about the OOD loss values, one could obtain a empirical estimate of the OOD loss values w.r.t. ERM loss values at convergence. Since the estimate is obtained under gradient conflicts, one could expect the ratios of OOD loss w.r.t. ERM loss should be higher when one could resolve the gradient conflicts properly. Therefore, one could assign a slightly higher preference to OOD losses than the empirically estimated ratios. In the model selection experiments, we directly increase the ratio by $1e2$ and find it works well as expected.

In fact, both **i)** and **ii)** are discussed under minimal assumption about the external knowledge of the optimization process, the task and the data. We expect a better estimate of the OOD preference could be obtained when more external inductive biases are incorporated. For instance, `PAIR-o` generalize to ParetoDA (Lv et al., 2021) when one could obtain a validation set that has similar distribution to the test data. Even under the case that such data is not available, one could also adopt some techniques such as Mixup (Zhang et al., 2018) to obtain an approximation. We believe that obtaining a better estimate of the ideal OOD preference would be a promising future development based on our work.

### D.4 DISCUSSION ON THE USE OF PAIR IN PRACTICE

### D.4.1 SCALABILITY

Similar to other MOO algorithms (Sener & Koltun, 2018; Lin et al., 2019; Mahapatra & Rajan, 2020), `PAIR-o` requires full gradients of the predictor to make an accurate derivation of the objective weights $\beta^*$, which could be a bottleneck when deployed to large-scale networks, as it usually involves

a prohibitively massive number of parameters. Sener & Koltun (2018) develops an approximation of the full gradients using the gradients w.r.t. the latent representation produced by the featurizer, i.e., $\partial \boldsymbol{L}/\partial\varphi(X)$. However, it requires a strong assumption on the structure of the data and the model. Moreover, when it involves complex network architectures such as DenseNet (Huang et al., 2017) or DistillBERT (Sanh et al., 2019) in WILDS, the approximation or even the full gradients can be even imprecise, as the gradients of the complex neural networks can not be directly concatenated as those of simple linear networks.

To this end, we develop another approximation that takes only the gradients of the classifier, which usually appears as a linear classification layer in the predictor. Interestingly, we empirically find $\partial \boldsymbol{L}/\partial w$ can even produce more useful signals for OOD generalization than the gradients w.r.t. classifier, shown as in Table 1.

When considering a more resource restricted scenarios, such as the iWildCam and RxRx1 in WILDS, we freeze the featurizer after the "descent" phase, which can further resolve the memory and computation overheads. It also aligns with some recent discoveries that the featurizer trained merely with ERM may already discovery all useful patterns (Rosenfeld et al., 2022). Zhang et al. (2022a) also find the technique useful in Camelyon17 dataset of WILDS.

### D.4.2 Loss value estimation

Similar to other MOO algorithms (Sener & Koltun, 2018; Lin et al., 2019; Mahapatra & Rajan, 2020), PAIR-o is described and analyzed in full batch setting, i.e., full gradient descent. However, in practice, stochastic setting tends to appear more often than vanilla gradient descent due to the scalability considerations. As also discussed in Sec. 4.1, variances are unavoidable no matter the estimated values are biased or unbiased. Fortunately, the robustness of PAIR-o to the preference can partially mitigate the issue.

The another potential limitation in PAIR-o could be the possibly negative estimate of some OOD losses, such as the stochastic estimates of IRMv1, since general MOO algorithms together with PAIR-o only accept non-negative loss values as the inputs. To this end, we will use IRMv1 as an example to explain how one could handle the potentially negative values in loss value estimation.

We will first introduce the unbiased empirical estimator of IRMv1, following Arjovsky et al. (2019); Ahuja et al. (2021b). More specifically, considering the IRMv1 objective,

$$\min_{\varphi} \sum_{e \in \mathcal{E}_{\text{tr}}} \mathcal{L}_e(\varphi) + \lambda |\nabla_{w|w=1} \mathcal{L}_e(w \cdot \varphi)|^2. \tag{21}$$

Observe that

$$\nabla_{w|w=1.0} \mathcal{L}_e(w \cdot \varphi) = \frac{\partial \mathbb{E}^e \big[\ell(w \cdot \varphi(X^e), Y^e)\big]}{\partial w}\Big|_{w=1.0} = \mathbb{E}^e \left[\frac{\partial \ell(w \cdot \varphi(X^e), Y^e)}{\partial w}\Big|_{w=1.0}\right]$$

and

$$\begin{aligned}
\|\nabla_{w|w=1.0} \mathcal{L}_e(w \cdot \varphi)\|^2 &= \left(\frac{\partial \mathbb{E}^e \big[\ell(w \cdot \varphi(X^e), Y^e)\big]}{\partial w}\Big|_{w=1.0}\right)^2 \\
&= \left(\mathbb{E}^e \left[\frac{\partial \ell(w \cdot \varphi(X^e), Y^e)}{\partial w}\Big|_{w=1.0}\right]\right)^2,
\end{aligned} \tag{22}$$

for which the simplification is derived by taking the derivative inside the expectation, using the Leibniz integral rule. Obviously, the stochastic estimate of Eq. 22 is biased.

To obtain an unbiased estimate of IRMv1 penalty, observe that

$$\mathbb{E}[X]^2 = \mathbb{E}[AB],$$

if $A$, $B$ and $X$ are i.i.d. random variables w.r.t. the same distribution $\mathcal{X}$. Equipped with this observation, we can further write Eq. 22 as

$$\begin{aligned}
\|\nabla_{w|w=1.0} \mathcal{L}_e(w \cdot \varphi)\|^2 &= \mathbb{E}^e \left[\left(\frac{\partial \ell(w \cdot \varphi(X^e), Y^e)}{\partial w}\Big|_{w=1.0}\right)\left(\frac{\partial \ell(w \cdot \varphi(\tilde{X}^e), \tilde{Y}^e)}{\partial w}\Big|_{w=1.0}\right)\right], \\
&= \left[\mathbb{E}^e \left(\frac{\partial \ell(w \cdot \varphi(X^e), Y^e)}{\partial w}\Big|_{w=1.0}\right)\mathbb{E}^e \left(\frac{\partial \ell(w \cdot \varphi(\tilde{X}^e), \tilde{Y}^e)}{\partial w}\Big|_{w=1.0}\right)\right],
\end{aligned} \tag{23}$$

where $(X^e, Y^e) \sim \mathbb{P}^e$ and $(\tilde{X}^e, \tilde{Y}^e) \sim \mathbb{P}^e$ are i.i.d. samples from $\mathbb{P}^e$ of the environment $e$. As $\mathbb{E}^e \left( \frac{\partial \ell(w \cdot \varphi(X^e), Y^e)}{\partial w} \Big|_{w=1.0} \right)$ and $\mathbb{E}^e \left( \frac{\partial \ell(w \cdot \varphi(\tilde{X}^e), \tilde{Y}^e)}{\partial w} \Big|_{w=1.0} \right)$ can separately be estimated in minibatches without bias, Eq. 23 essentially provides a practical unbiased estimator of IRMv1.

However, different from IRMv1, Eq. 23 does not have any guarantees for its non-negativity, though the expectation of Eq. 23 is non-negative. To this end, we propose two heuristics to mitigate the issue.

The first heuristic is to add all minibatch estimates $\mathbb{E}^e \left( \frac{\partial \ell(w \cdot \varphi(X^e), Y^e)}{\partial w} \Big|_{w=1.0} \right)$ by a sufficiently large constant $C$, such that the minimum value of $\mathbb{E}^e \left( \frac{\partial \ell(w \cdot \varphi(X^e), Y^e)}{\partial w} \Big|_{w=1.0} \right) + C$ is non-negative. Moreover, as the constant does not affect the calculation of the gradients, when IRMv1 is minimized to $0$, $\mathbb{E}^e \left( \frac{\partial \ell(w \cdot \varphi(X^e), Y^e)}{\partial w} \Big|_{w=1.0} \right)$ is also optimized to $C$.

The other heuristic is to multiply the negative minibatch estimates $\mathbb{E}^e \left( \frac{\partial \ell(w \cdot \varphi(X^e), Y^e)}{\partial w} \Big|_{w=1.0} \right)$ by a proper negative constant $-C$, which will make all estimations non-negative. On the other hand, however, it can dramatically affect the variances in the estimations. Essentially, this multiplication will enlarge the expectation of the estimated IRMv1, and may cause instability of the training, due to the unrobustness of IRMv1. Therefore, we can heuristically search the values $C$ from $1$ to $1e-4$ by observing the early training dynamics. If the training is unstable, then we heuristically tune $C$ to be smaller by $1e-2$.

Although both of the heuristics above can not rigorously recover a non-negative estimate of IRMv1 penalty (which is essentially impossible for the formulations like IRMv1), we empirically find them effective, for which we hypothesize is because of the robustness of `PAIR-o` to the preference in OOD generalization.

### D.4.3 GENERALIZING TO OTHER OOD METHODS

As shown in Fig. 1(b), the gradient conflicts between ERM and OOD objectives generally exist (Arjovsky et al., 2019; Krueger et al., 2021; Wald et al., 2021; Pezeshki et al., 2021; Rame et al., 2021). It implies that, on the one hand, the optimization dilemma generally exist for all OOD objectives. Meanwhile, both `PAIR-o` and `PAIR-s` are generically applicable to all OOD methods. In experiments (Sec. 5), we validate the generality of `PAIR-s` only for several OOD methods from the four main lines as discussed in related works (Sec. B.1) though, `PAIR-o` essentially has similar generality as `PAIR-s`, for whose performances at real world datasets, we will leave for future verification due to the limited computational resources. Nevertheless, we can theoretically discuss the implementation options about how `PAIR-o` can be applied to different OOD methods.

First, for Domain Generalization based methods (Ganin et al., 2016; Sun & Saenko, 2016; Li et al., 2018; Dou et al., 2019), such as DANN (Ganin et al., 2016), `PAIR-o` can directly take the domain classification loss and the label classification loss as the inputs.

Second, for Distributionally Robust Optimization methods (Namkoong & Duchi, 2016; Hu et al., 2018; Sagawa* et al., 2020), `PAIR-o` can take the worst group loss or some more sophisticated regularizations and the ERM loss as the inputs.

Third, for the causal invariance based methods (Peters et al., 2016; Rojas-Carulla et al., 2018; Arjovsky et al., 2019; Creager et al., 2021; Parascandolo et al., 2021; Wald et al., 2021; Ahuja et al., 2021a; Chen et al., 2022b) and agreement based methods (Koyama & Yamaguchi, 2020; Krueger et al., 2021; Shi et al., 2022; Rame et al., 2021), they can be handled by `PAIR-o` similarly as IRMX.

## E THEORETICAL DISCUSSIONS

### E.1 PROOF FOR PROPOSITION 1

We first restate the proposition with formally defined Setting A by Kamath et al. (2021).

**Setting A (identical to Kamath et al. (2021)):** Considering the task of linear classification/regression $\mathcal{X} \rightarrow \mathcal{Y}$ where the quality of predictors $f : \mathcal{X} \rightarrow \widehat{\mathcal{Y}}$ is measured by population

losses $l : \hat{\mathcal{Y}} \times \mathcal{Y} \to \mathbb{R}_{\geq 0}$, $\hat{\mathcal{Y}} = \mathbb{R}$, $\mathcal{Y} \subseteq \mathbb{R}$, $\ell$ is either the square loss $\ell_{\text{sq}}(\hat{y}, y) := \frac{1}{2}(\hat{y} - y)^2$, or the logistic loss $\ell_{\log}(\hat{y}, y) := \log(1 + \exp(-\hat{y}y))$ when $\mathcal{Y} = \{-1, 1\}$ (binary classification).

**Proposition 2.** *Under Setting A ([Kamath et al. (2021)](#)), for all $\alpha \in (0, 1)$, let $\mathcal{E} := \{(\alpha, \beta_e) : \beta_e \in (0, 1)\}$ be any instance of the two-bit environment (Eq. 13), $\mathcal{I}_X$ denote the invariant predictors produced by Eq. 7, it holds that $\mathcal{I}_{\mathcal{S} \cap X}(\mathcal{E}) = \mathcal{I}(\mathcal{E})$.*[4]

Our proof is proceeded by discussing the set of invariant predictors elicited by an ideal V-REx ([Krueger et al., 2021](#)) objective $\mathcal{I}_X(\mathcal{E})$ (in a more general way), and then incorporating $\mathcal{I}_X(\mathcal{E})$ into that elicited by IRM$_{\mathcal{S}}$ or IRMv1 ([Arjovsky et al., 2019](#)) $\mathcal{I}_{\mathcal{S}}(\mathcal{E})$ for the two-bit failure case (Eq. 13).

We now first discuss the invariant predictors produced by the invariance constraints ideally elicited by V-REx. Recall that V-REx ([Krueger et al., 2021](#)) aims to minimize the variances of ERM losses at different environments:

$$\mathcal{L}_{\text{VREx}} := \text{var}(\{\mathcal{L}_e\}_{e \in \mathcal{E}_{\text{tr}}}).$$

Therefore, when $\mathcal{L}_{\text{VREx}}$ is minimized, we have $\mathcal{L}_{e_1} = \mathcal{L}_{e_2}$, $\forall e_1, e_2 \in \mathcal{E}_{\text{tr}}$. Then, we can define the invariant predictors produced by V-REx, as the following.

**VREx$_0$:** Define $\mathcal{I}_X(\mathcal{E}) := \{f : \mathcal{X} \to \hat{\mathcal{Y}} \mid \mathcal{L}_{e_1}(f) = \mathcal{L}_{e_2}(f), \forall e_1, e_2 \in \mathcal{E}\}$. VREx$_0$ is the objective:

$$\min_{f \in \mathcal{I}_X(\mathcal{E}_{\text{tr}})} \sum_{e \in \mathcal{E}_{\text{tr}}} \mathcal{L}_e(f).$$

Then, we characterize the set of $\mathcal{I}_X$ through the following lemma.

**Lemma 1.** *Under Setting A, let $f = w \circ \varphi$ be the predictor elicited by $\mathcal{I}(\mathcal{E})$ and $(X_e, Y_e) \sim \mathcal{D}_e$.*

*If $\begin{cases} \ell = \ell_{\text{sq}}, \ \mathbb{E}_{\mathcal{D}_e}[Y_e^2] \text{ is identical}, \text{ the distribution of } \varphi(X_e) \text{ is identical (or } f \equiv 0) \\ \ell = \ell_{\log} \text{ and } H(Y_e | \varphi(X_e)) \text{ is identical} \end{cases}$ for all $e \in \mathcal{E}$,*

*then $\mathcal{I}(\mathcal{E}) \subseteq \mathcal{I}_X(\mathcal{E})$.*

*Proof.* For any $f = w \circ \varphi \in \mathcal{I}(\mathcal{E})$, using Observation 2 in (Kamath et al, 2021), we have that

$$\mathbb{E}_{\mathcal{D}_{e_1}}[Y \mid \varphi(X) = z] = \mathbb{E}_{\mathcal{D}_{e_2}}[Y \mid \varphi(X) = z], \tag{24}$$

for all $e_1, e_2 \in \mathcal{E}$ and for all $z \in \mathcal{Z}$.[5]

(i) For square loss $\ell_{\text{sq}}$,

$$\begin{aligned} \mathcal{L}_e(f) &= \frac{1}{2} \mathbb{E}_{\mathcal{D}_e}[(f(X) - Y)^2] \\ &= \frac{1}{2} \mathbb{E}_{\mathcal{D}_e}[f(X)^2 - 2f(X)Y + Y^2] \\ &= \frac{1}{2} \mathbb{E}_{\mathcal{D}_e}\left[\mathbb{E}_{\mathcal{D}_e}[w \circ \varphi(X)^2 - 2w \circ \varphi(X)Y \mid \varphi(X)]\right] + \frac{1}{2} \mathbb{E}_{\mathcal{D}_e}[Y^2], \end{aligned}$$

where $w$ is the simultaneously optimal classifier for all $e \in \mathcal{E}$.

Then, note that for all $z \in \mathcal{Z}$, it holds that

$$\mathbb{E}_{\mathcal{D}_e}[w(z)^2 - 2w(z)Y \mid \varphi(X) = z] = w(z)^2 - 2w(z)\mathbb{E}_{\mathcal{D}_e}[Y \mid \varphi(X) = z].$$

Using equation 24 and the assumptions that $\mathbb{E}_{\mathcal{D}_e}[Y^2]$ is identical and the distribution of $\varphi(X)$ is identical (or $f \equiv 0$) for all $e \in \mathcal{E}$, we can conclude that for all $e_1, e_2 \in \mathcal{E}$, $\mathcal{L}_{e_1}(f) = \mathcal{L}_{e_2}(f)$.

(ii) For logistic loss $\ell_{\log}$, note that the simultaneously optimal $w$ has the form

$$w(z) = \log\left(\frac{\text{Pr}_{\mathcal{D}_e}[Y = 1 \mid \varphi(X) = z]}{\text{Pr}_{\mathcal{D}_e}[Y = -1 \mid \varphi(X) = z]}\right) = \log\left(\frac{1 + \mathbb{E}_{\mathcal{D}_e}[Y \mid \varphi(X) = z]}{1 - \mathbb{E}_{\mathcal{D}_e}[Y \mid \varphi(X) = z]}\right),$$

for all $e \in \mathcal{E}$ and all $z \in \mathcal{Z}$. We can thus conclude that in this case, $\mathcal{L}_e(f) = \mathbb{E}_{\mathcal{D}_e}[H(Y | \varphi(X) = z)] = H(Y | \varphi(X))$, which completes the proof. $\qquad\square$

---

[4]Motivated readers might be interested in the necessities of keeping IRMv1 in the objectives, for which we provide details in Appendix C.2.

[5]We assume that the support of $\varphi(X)$ (denoted as $\mathcal{Z}$) is identical in each environment for simplicity.

**Remarks.** We formulate Lemma 1 in a general setting that covers Two-Bit-Env as a special case. It can be easily verified that the assumptions in this lemma are all satisfied in Two-Bit-Env (Eq. 13). Moreover, we can show that other environment settings (e.g., those in IB-IRM (Ahuja et al., 2021a)) also satisfy the assumptions.

**Proposition 3.** *Under Setting A, for all $\alpha \in (0,1)$, let $\mathcal{E} := \{(\alpha, \beta_e) : \beta_e \in (0,1)\}$ and $f$ be an odd (or linear) predictor. It holds that $\mathcal{I}_X(\mathcal{E}) \cap \mathcal{I}_S(\mathcal{E}) = \mathcal{I}(\mathcal{E})$.*

*Proof.* From the proof of Proposition 5 in Kamath et al. (2021), we know that there are only two predictors in $\mathcal{I}(\mathcal{E})$: The zero predictor $f_0 \equiv 0$ (for both $\ell_{\mathrm{sq}}$ and $\ell_{\mathrm{log}}$) and $f_{\mathrm{IRM}}(x_1, x_2) = (1 - 2\alpha) \cdot x_1$ (for $\ell = \ell_{\mathrm{sq}}$) or $f_{\mathrm{IRM}}(x_1, x_2) = \log \frac{1-\alpha}{\alpha} \cdot x_1$ (for $\ell = \ell_{\mathrm{log}}$).

(i) For square loss $\ell_{\mathrm{sq}}$, $\mathcal{L}_e(f) = \frac{1}{2}\mathbb{E}_{\mathcal{D}_e}[f(X)^2 - 2f(X)Y + Y^2]$. Note that in Two-Bit-Env, $Y^2 \equiv 1$. Thus, in this case, $f \in \mathcal{I}_X(\mathcal{E})$ implies that $\mathbb{E}_{\mathcal{D}_e}[f(X)^2 - 2f(X)Y]$ is identical for all $e \in \mathcal{E}$. Moreover,

$$f \in \mathcal{I}_S(\mathcal{E}) \Rightarrow \nabla_{w|w=1}\mathcal{L}_e(f) = 0 \text{ for all } e \in \mathcal{E}$$
$$\Rightarrow \mathbb{E}_{\mathcal{D}_e}[f(X)^2] = \mathbb{E}_{\mathcal{D}_e}[f(X)Y] \text{ for all } e \in \mathcal{E}.$$

We can conclude that for any $f \in \mathcal{I}_X(\mathcal{E}) \cap \mathcal{I}_S(\mathcal{E})$, it holds that

$$\mathbb{E}_{\mathcal{D}_e}[f(X)^2] \text{ and } \mathbb{E}_{\mathcal{D}_e}[f(X)Y] \text{ are identical for all } e \in \mathcal{E}, \tag{25}$$
$$\mathbb{E}_{\mathcal{D}_e}[f(X)^2] = \mathbb{E}_{\mathcal{D}_e}[f(X)Y] \text{ for all } e \in \mathcal{E}. \tag{26}$$

Denote $f_{(1,1)} := f(X_1 = 1, X_2 = 1)$, and $f_{(1,-1)}, f_{(-1,1)}, f_{(-1,-1)}$ are similarly defined. For condition equation 25,

$$
\begin{aligned}
\mathbb{E}_{\mathcal{D}_e}[f(X)^2] &= \frac{1-\alpha}{2}\left(f_{(1,1)}^2 + f_{(-1,-1)}^2\right) + \frac{\alpha}{2}\left(f_{(1,-1)}^2 + f_{(-1,1)}^2\right) \\
&\quad + \frac{\beta_e(1-2\alpha)}{2}\left(-f_{(1,1)}^2 - f_{(-1,-1)}^2 + f_{(1,-1)}^2 + f_{(-1,1)}^2\right), \\
\mathbb{E}_{\mathcal{D}_e}[f(X)Y] &= \frac{1-\alpha}{2}\left(f_{(1,1)} - f_{(-1,-1)}\right) + \frac{\alpha}{2}\left(f_{(-1,1)} - f_{(1,-1)}\right) \\
&\quad - \frac{\beta_e}{2}\left(f_{(1,1)} - f_{(-1,-1)} + f_{(-1,1)} - f_{(1,-1)}\right).
\end{aligned}
\tag{27}
$$

To enforce condition equation 25 for any $\alpha, \beta_e \in (0,1)$, it is required that

$$
\begin{cases}
f_{(1,1)} - f_{(-1,-1)} + f_{(-1,1)} - f_{(1,-1)} = 0, \\
-f_{(1,1)}^2 - f_{(-1,-1)}^2 + f_{(1,-1)}^2 + f_{(-1,1)}^2 = 0.
\end{cases}
\Rightarrow
\begin{cases}
f_{(1,1)} - f_{(-1,-1)} = -\left(f_{(-1,1)} - f_{(1,-1)}\right), \\
f_{(1,1)}^2 + f_{(-1,-1)}^2 = f_{(1,-1)}^2 + f_{(-1,1)}^2.
\end{cases}
$$

In this case, condition equation 26 implies that $f_{(1,1)}^2 + f_{(-1,-1)}^2 = (1 - 2\alpha)\left(f_{(1,1)} - f_{(-1,-1)}\right)$. Without restricting $f$ to be an odd predictor (or equivalently, linear predictor), this constraint is a circle passing through $f_0$ and $f_{\mathrm{IRM}}$. Requiring that $f$ is odd, i.e., $f_{(1,1)} = -f_{(-1,-1)}$ and $f_{(1,-1)} = -f_{(-1,1)}$, we can conclude that there are only two predictors left in $\mathcal{I}_X(\mathcal{E}) \cap \mathcal{I}_S(\mathcal{E})$, which are $f_{(1,1)} = f_{(-1,-1)} = f_{(1,-1)} = f_{(-1,1)} = 0$ and

$$
\begin{cases}
f_{(1,1)} = 1 - 2\alpha, \\
f_{(-1,-1)} = 2\alpha - 1, \\
f_{(1,-1)} = 1 - 2\alpha, \\
f_{(-1,1)} = 2\alpha - 1.
\end{cases}
\Rightarrow f(x_1, x_2) = (1 - 2\alpha) \cdot x_1.
$$

(ii) For logistic loss $\ell_{\mathrm{log}}$, $\mathcal{L}_e(f) = \mathbb{E}_{\mathcal{D}_e}\left[\log\left(1 + \exp\left(-f(X)Y\right)\right)\right]$. Similarly, $f \in \mathcal{I}_X(\mathcal{E}) \cap \mathcal{I}_S(\mathcal{E})$ implies that

$$\mathbb{E}_{\mathcal{D}_e}\left[\log\left(1 + \exp\left(-f(X)Y\right)\right)\right] \text{ is identical for all } e \in \mathcal{E}, \tag{28}$$
$$\mathbb{E}_{\mathcal{D}_e}\left[\frac{-f(X)Y}{1 + \exp\left(f(X)Y\right)}\right] = 0. \tag{29}$$

From condition equation 28 and that $f$ is an odd predictor ($f_{(1,1)} = -f_{(-1,-1)}$ and $f_{(1,-1)} = -f_{(-1,1)}$), we can conclude that

$$\frac{(1+e^{f_{(1,1)}})^{2\alpha}}{(1+e^{-f_{(1,1)}})^{2-2\alpha}} = \frac{(1+e^{f_{(1,-1)}})^{2\alpha}}{(1+e^{-f_{(1,-1)}})^{2-2\alpha}} \Rightarrow f_{(1,1)} = f_{(1,-1)},$$

which is due to that $\frac{(1+e^x)^{2\alpha}}{(1+e^{-x})^{2-2\alpha}}$ is a one-to-one function.

In this case, condition equation 29 can be simplified as

$$e^{f_{(1,1)}} f_{(1,1)}\alpha - f_{(1,1)}(1-\alpha) = 0 \Rightarrow f_{(1,1)} = 0 \text{ or } f_{(1,1)} = \log\frac{1-\alpha}{\alpha}.$$

Thus, the only predictors in $\mathcal{I}_X(\mathcal{E}) \cap \mathcal{I}_S(\mathcal{E})$ are $f_0$ and $f_{\text{IRM}}$. $\qquad\square$

**Corollary 1.** *Under Setting A, for all $\alpha \in (0,1)$ and $\mathcal{E}_{\text{tr}} = \{(\alpha, \beta_{e_1}), (\alpha, \beta_{e_2})\}$ for any two distinct $\beta_{e_1}, \beta_{e_2} \in (0,1)$, $\mathcal{I}_X(\mathcal{E}_{\text{tr}}) \cap \mathcal{I}_S(\mathcal{E}_{\text{tr}}) = \mathcal{I}_X(\mathcal{E}) \cap \mathcal{I}_S(\mathcal{E})$.*

*Proof.* This directly follows from the observation that in the proof of Proposition 3, enforcing condition equation 25 and equation 28 for two distinct $\beta_{e_1}, \beta_{e_2}$ impose the identical constraints on $f$. $\qquad\square$

### E.2 PROOF FOR THEOREM 4.1

We first restate the informal version of the theorem as the following, while the formal description of Theorem E.1 will be given in Theorem E.4 with more formal definitions.

**Theorem E.1.** *(Informal) For $\gamma \in (0,1)$ and any $\epsilon, \delta > 0$, if $\mathcal{F}$ is a finite hypothesis class, both ERM and OOD losses are bounded above, let $I_{\text{PAIR}}$ be the index of all losses, $p_{\max} := \max_{i \in I_{\text{PAIR}}} p_i$ and $L_{\max} := \max_{i \in I_{\text{PAIR}}} L_i$, if the number of training samples $|D| \geq \frac{32L_{\max}^2 p_{\max}^2}{\delta^2} \log\frac{2(m+1)|\mathcal{F}|}{\gamma}$, then with probability at least $1 - \gamma$, `PAIR-o` and `PAIR-s` yield an $\epsilon$-approximated solution of $f_{ood}$.*

The proof for Theorem 4.1 is also a theoretical discussion on the performances of `PAIR-o` and `PAIR-s` under an approximated OOD preference. Essentially, the performances of both `PAIR-o` and `PAIR-s` have a certain dependence on the quality of the OOD preference $\boldsymbol{p}_{\text{ood}}$, however, it is often the case that the ideal OOD preference is usually unknown. It is desirable to analyze the performances of `PAIR-o` and `PAIR-s` under an imprecise OOD preference. Mahapatra & Rajan (2020) discussed a bit that when the exact Pareto optimal solution under the preference does not exist, the EPO solver can still find a Pareto optimal solution that is closest to the preferred direction. We discuss it in a more general way by developing a new MOO formulation of Eq. 16 under an approximated preference up to some approximation error of $\epsilon$.

Without loss of generality, given a OOD preference $\boldsymbol{p}_{\text{ood}} = (p_{\text{ERM}}, p_1, ..., p_m)^T = (\frac{1}{\epsilon_{\text{inv}}}, \frac{1}{\epsilon}_{\text{ood}})^T$, the ERM loss $\mathcal{L}_{\text{ERM}}$ and $m$ OOD losses $\boldsymbol{L}_{\text{ood}} = (\mathcal{L}_{\text{ood}}^1, \mathcal{L}_{\text{ood}}^2, .., \mathcal{L}_{\text{ood}}^m)^T$, Eq. 16 can be reformulated as

$$\boldsymbol{f}_{\text{PAIR}} := \underset{f \in \mathcal{F}}{\arg\min} \quad \mathcal{L}_{\text{ERM}}(f) \tag{30}$$
$$\text{s.t.} \qquad p_{\text{ERM}}\mathcal{L}_{\text{ERM}}(f) = p_1\mathcal{L}_{\text{ood}}^1(f) = p_2\mathcal{L}_{\text{ood}}^2(f) = \cdots = p_m\mathcal{L}_{\text{ood}}^m(f).$$

We remark that under the ideal OOD preference, the optimal solution of Eq. 30, is also the optimal solution to Eq. 16 (i.e., the unconstrained version). In other words, $\boldsymbol{f}_{\text{PAIR}} = f_{\text{ood}}$. We will use $\boldsymbol{f}_{\text{PAIR}}$ to differentiate from the solution to the unconstrained version. We focus on Eq. 30 for the reason that it is more convenient to establish the discussion on the approximated OOD preference, from the perspective of optimization constraints.

Exactly enforcing the above preference constraint is too restrictive *both practically and theoretically*, instead we incorporate the approximation by relaxing the constraint of the loss values w.r.t. the OOD preference. The $\epsilon$-approximated problem of Eq. 30 is as the following

$$\boldsymbol{f}_{\text{PAIR}}^\epsilon := \underset{f \in \mathcal{F}}{\arg\min} \quad \mathcal{L}_{\text{ERM}}(f) \tag{31}$$
$$\text{s.t.} \qquad \forall i,j \in I_{\text{PAIR}}, i \neq j, |p_i\mathcal{L}_i(f) - p_j\mathcal{L}_j(f)| \leq \epsilon,$$

where $I_{\text{PAIR}} := \{\text{ERM}, \text{ood}_1, \text{ood}_2, \ldots, \text{ood}_m\}$ is the index set of overall losses. We denote the relaxed constraint set in Eq. 31 as $\boldsymbol{P}_{\text{PAIR}}^\epsilon := \{f \mid \forall i, j \in I_{\text{PAIR}}, i \neq j, |p_i \mathcal{L}_i(f) - p_j \mathcal{L}_j(f)| \leq \epsilon\}$. Clearly, it holds that the solution sets satisfy $\boldsymbol{f}_{\text{PAIR}}^0 = \boldsymbol{f}_{\text{PAIR}}$.

Then we define the empirical version of the $\epsilon$-approximated problem Eq. 31 with preference vector $\boldsymbol{p}_{\text{ood}}$ as follows.

$$
\begin{aligned}
\hat{\boldsymbol{f}}_{\text{PAIR}}^\epsilon &:= \underset{f \in \mathcal{F}}{\arg\min} \quad \widehat{\mathcal{L}}_{\text{ERM}}(f) \\
&\text{s.t.} \qquad \forall i, j \in I_{\text{PAIR}}, i \neq j, |p_i \widehat{\mathcal{L}}_i(f) - p_j \widehat{\mathcal{L}}_j(f)| \leq \epsilon.
\end{aligned}
\tag{32}
$$

Similarly, we denote the above constraint set as $\widehat{\boldsymbol{P}}_{\text{PAIR}}^\epsilon := \{f \mid \forall i, j \in I_{\text{PAIR}}, i \neq j, |p_i \widehat{\mathcal{L}}_i(f) - p_j \widehat{\mathcal{L}}_j(f)| \leq \epsilon\}$.

Assume a finite hypothesis class $\mathcal{F}$ and define

$$
\delta = \min_{f \in \mathcal{F}, \forall i, j \in I_{\text{PAIR}}, i \neq j} \big| |p_i \mathcal{L}_i(f) - p_j \mathcal{L}_j(f)| - \epsilon \big|.
$$

First, we recall the definition of $\nu$-representative sample from Shalev-Shwartz & Ben-David (2014).

**Definition E.2.** *(Shalev-Shwartz & Ben-David (2014)) A training set $S$ is called $\nu$-representative (w.r.t. domain $\mathcal{X}$, hypothesis $\mathcal{F}$, loss $\ell$ and distribution $\mathcal{D}$) if*

$$
\forall f \in \mathcal{F}, |\widehat{\mathcal{L}}(f) - \mathcal{L}(f)| \leq \nu,
$$

*where $\mathcal{L}(f) := \mathbb{E}_{(X,Y) \sim \mathcal{D}}[\ell(f(X), Y)]$ and $\widehat{\mathcal{L}}(f) := \frac{1}{|S|} \sum_{(X_i, Y_i) \in S} \ell(f(X_i), Y_i)$.*

Equipped with this definition, we can now characterize the condition under which the constraint sets in equation 31 and equation 32 contain exact the same predictors.

**Lemma 2.** *For any $\epsilon > 0$, assuming $\delta > 0$ and denoting $p_{\max} := \max_{i \in I_{PAIR}} p_i$, if the training set $\mathcal{D}_{\text{tr}}$ is $\frac{\delta}{4 p_{\max}}$-representative w.r.t. domain $\mathcal{X}$, hypothesis $\mathcal{F}$, distribution $\mathcal{D}$ and all the ERM and OOD losses $\{\mathcal{L}_{ERM}, \boldsymbol{L}_{ood}\}$, then $\boldsymbol{P}_{PAIR}^\epsilon = \widehat{\boldsymbol{P}}_{PAIR}^\epsilon$.*

*Proof.* We first show that $\boldsymbol{P}_{\text{PAIR}}^\epsilon \subseteq \widehat{\boldsymbol{P}}_{\text{PAIR}}^\epsilon$. By the definition of $\delta$, for all $f \in \mathcal{F}$, and $\forall i, j \in I_{\text{PAIR}}, i \neq j$ we have

$$
|p_i \mathcal{L}_i(f) - p_j \mathcal{L}_j(f)| \leq \epsilon - \delta \text{ or } |p_i \mathcal{L}_i(f) - p_j \mathcal{L}_j(f)| \geq \epsilon + \delta.
\tag{33}
$$

Using this property, for any $f \in \boldsymbol{P}_{\text{PAIR}}^\epsilon$, we can conclude that $\forall i, j \in I_{\text{PAIR}}, i \neq j$,

$$
|p_i \mathcal{L}_i(f) - p_j \mathcal{L}_j(f)| \leq \epsilon \Rightarrow |p_i \mathcal{L}_i(f) - p_j \mathcal{L}_j(f)| \leq \epsilon - \delta.
$$

This inequality further implies that

$$
\begin{aligned}
&|p_i \mathcal{L}_i(f) - p_i \widehat{\mathcal{L}}_i(f) + p_j \widehat{\mathcal{L}}_j(f) - p_j \mathcal{L}_j(f) + p_i \widehat{\mathcal{L}}_i(f) - p_j \widehat{\mathcal{L}}_j(f)| \leq \epsilon - \delta \\
\Rightarrow & \big| |p_i \widehat{\mathcal{L}}_i(f) - p_j \widehat{\mathcal{L}}_j(f)| - |p_i \mathcal{L}_i(f) - p_i \widehat{\mathcal{L}}_i(f) + p_j \widehat{\mathcal{L}}_j(f) - p_j \mathcal{L}_j(f)| \big| \leq \epsilon - \delta \\
\Rightarrow & |p_i \widehat{\mathcal{L}}_i(f) - p_j \widehat{\mathcal{L}}_j(f)| \leq \epsilon - \delta + |p_i \mathcal{L}_i(f) - p_i \widehat{\mathcal{L}}_i(f) + p_j \widehat{\mathcal{L}}_j(f) - p_j \mathcal{L}_j(f)| \\
\Rightarrow & |p_i \widehat{\mathcal{L}}_i(f) - p_j \widehat{\mathcal{L}}_j(f)| \leq \epsilon - \delta + p_i |\mathcal{L}_i(f) - \widehat{\mathcal{L}}_i(f)| + p_j |\widehat{\mathcal{L}}_j(f) - \mathcal{L}_j(f)|,
\end{aligned}
$$

which is based on the triangle inequality of the absolute value function.

From the definition of $\frac{\delta}{4 p_{\max}}$-representative, we have $|\mathcal{L}_i(f) - \widehat{\mathcal{L}}_i(f)| \leq \frac{\delta}{4 p_{\max}}, \forall i \in I_{\text{PAIR}}$. Substituting this in the above inequality, we obtain

$$
\begin{aligned}
|p_i \widehat{\mathcal{L}}_i(f) - p_j \widehat{\mathcal{L}}_j(f)| &\leq \epsilon - \delta + \frac{p_i \delta}{4 p_{\max}} + \frac{p_j \delta}{4 p_{\max}} \\
&\leq \epsilon - \frac{\delta}{2},
\end{aligned}
$$

which implied that $f \in \widehat{\boldsymbol{P}}_{\text{PAIR}}^\epsilon$.

Then, we prove that $\widehat{\boldsymbol{P}}^\epsilon_{\text{PAIR}} \subseteq \boldsymbol{P}^\epsilon_{\text{PAIR}}$.

For any $f \in \widehat{\boldsymbol{P}}^\epsilon_{\text{PAIR}}$, it holds that $\forall i, j \in I_{\text{PAIR}}, i \neq j$,

$$|p_i\widehat{\mathcal{L}}_i(f) - p_j\widehat{\mathcal{L}}_j(f)| \leq \epsilon$$
$$\Rightarrow |p_i\widehat{\mathcal{L}}_i(f) - p_i\mathcal{L}_i(f) + p_j\mathcal{L}_j(f) - p_j\widehat{\mathcal{L}}_j(f) + p_i\mathcal{L}_i(f) - p_j\mathcal{L}_j(f)| \leq \epsilon$$
$$\Rightarrow \left||p_i\mathcal{L}_i(f) - p_j\mathcal{L}_j(f)| - |p_i\widehat{\mathcal{L}}_i(f) - p_i\mathcal{L}_i(f) + p_j\mathcal{L}_j(f) - p_j\widehat{\mathcal{L}}_j(f)|\right| \leq \epsilon$$
$$\Rightarrow |p_i\mathcal{L}_i(f) - p_j\mathcal{L}_j(f)| \leq \epsilon + |p_i\widehat{\mathcal{L}}_i(f) - p_i\mathcal{L}_i(f) + p_j\mathcal{L}_j(f) - p_j\widehat{\mathcal{L}}_j(f)|$$
$$\Rightarrow |p_i\mathcal{L}_i(f) - p_j\mathcal{L}_j(f)| \leq \epsilon + p_i|\widehat{\mathcal{L}}_i(f) - \mathcal{L}_i(f)| + p_j|\mathcal{L}_j(f) - \widehat{\mathcal{L}}_j(f)|$$
$$\Rightarrow |p_i\mathcal{L}_i(f) - p_j\mathcal{L}_j(f)| \leq \epsilon + \frac{p_i\delta}{4p_{\max}} + \frac{p_j\delta}{4p_{\max}}$$
$$\leq \epsilon + \frac{\delta}{2},$$

which is again based on the triangle inequality of the absolute value function and the definition of $\frac{\delta}{4p_{\max}}$-representative. Together with equation 33, we conclude that $|p_i\mathcal{L}_i(f) - p_j\mathcal{L}_j(f)| \leq \epsilon - \delta \Rightarrow f \in \boldsymbol{P}^\epsilon_{\text{PAIR}}$, which implies $\widehat{\boldsymbol{P}}^\epsilon_{\text{PAIR}} \subseteq \boldsymbol{P}^\epsilon_{\text{PAIR}}$.

Based on the above discussion, we have proven that $\boldsymbol{P}^\epsilon_{\text{PAIR}} = \widehat{\boldsymbol{P}}^\epsilon_{\text{PAIR}}$. $\qquad\square$

**Assumption E.3.** *For all $f \in \mathcal{F}, X \in \mathcal{X}, Y \in \mathcal{Y}$, the ERM loss is bounded, i.e., $|\ell(f(X), Y)| \leq L_{ERM} < \infty$, and all the OOD objectives $\boldsymbol{L}_{ood}$ can be written as the expectation of some bounded loss functions, i.e., $\forall i \in [m], \mathcal{L}^i_{ood}(f) = \mathbb{E}_{(X,Y)\sim\mathcal{D}}[\ell^i_{ood}(f(X), Y)]$ and $|\ell^i_{ood}(f(X), Y)| \leq L^i_{ood} < \infty$.*

We remark that the assumption is natural and generally holds for many OOD objectives including IRMv1 (Arjovsky et al., 2019) and VREx (Krueger et al., 2021).

**Theorem E.4.** *For any $\epsilon > 0, \gamma \in (0, 1)$, if Assumption E.3 holds and $\delta > 0$, denoting $p_{\max} := \max_{i \in I_{PAIR}} p_i$ and $L_{\max} := \max_{i \in I_{PAIR}} L_i$, if the number of training samples $|\mathcal{D}_{\text{tr}}| \geq \frac{32L^2_{\max}p^2_{\max}}{\delta^2} \log \frac{2(m+1)|\mathcal{F}|}{\gamma}$, then with probability at least $1 - \gamma$, we have for any $f^\epsilon_{PAIR} \in \boldsymbol{f}^\epsilon_{PAIR}$ and $\hat{f}^\epsilon_{PAIR} \in \hat{\boldsymbol{f}}^\epsilon_{PAIR}$, $\mathcal{L}_{ERM}(f^\epsilon_{PAIR}) \leq \mathcal{L}_{ERM}(\hat{f}^\epsilon_{PAIR}) \leq \mathcal{L}_{ERM}(f^\epsilon_{PAIR}) + \frac{\delta}{2p_{\max}}$.*

*Proof.* We proceed by first assuming that the training set $D$ is $\frac{\delta}{4p_{\max}}$-representative w.r.t. domain $\mathcal{X}$, hypothesis $\mathcal{F}$, distribution $\mathcal{D}$ and all the ERM and OOD losses $\{\mathcal{L}_{\text{ERM}}, \boldsymbol{L}_{\text{ood}}\}$, and then we establish the sample complexity required for this condition. From Lemma 2, we know that given this condition and the assumptions in the theorem, $\boldsymbol{P}^\epsilon_{\text{PAIR}} = \widehat{\boldsymbol{P}}^\epsilon_{\text{PAIR}}$. Then, since the training set $\mathcal{D}_{\text{tr}}$ is $\frac{\delta}{4p_{\max}}$-representative w.r.t. the ERM loss $\mathcal{L}_{\text{ERM}}$, we have for any $f^\epsilon_{\text{PAIR}} \in \boldsymbol{f}^\epsilon_{\text{PAIR}}$ and $\hat{f}^\epsilon_{\text{PAIR}} \in \hat{\boldsymbol{f}}^\epsilon_{\text{PAIR}}$,

$$\left|\mathcal{L}_{\text{ERM}}(f^\epsilon_{\text{PAIR}}) - \widehat{\mathcal{L}}_{\text{ERM}}(f^\epsilon_{\text{PAIR}})\right| \leq \frac{\delta}{4p_{\max}},$$
$$\left|\mathcal{L}_{\text{ERM}}(\hat{f}^\epsilon_{\text{PAIR}}) - \widehat{\mathcal{L}}_{\text{ERM}}(\hat{f}^\epsilon_{\text{PAIR}})\right| \leq \frac{\delta}{4p_{\max}}.$$

Moreover, based on the optimality of problem equation 32, we can conclude that

$$\mathcal{L}_{\text{ERM}}(\hat{f}^\epsilon_{\text{PAIR}}) - \frac{\delta}{4p_{\max}} \leq \widehat{\mathcal{L}}_{\text{ERM}}(\hat{f}^\epsilon_{\text{PAIR}}) \leq \widehat{\mathcal{L}}_{\text{ERM}}(f^\epsilon_{\text{PAIR}}) \leq \mathcal{L}_{\text{ERM}}(f^\epsilon_{\text{PAIR}}) + \frac{\delta}{4p_{\max}}$$
$$\Rightarrow \mathcal{L}_{\text{ERM}}(\hat{f}^\epsilon_{\text{PAIR}}) \leq \mathcal{L}_{\text{ERM}}(f^\epsilon_{\text{PAIR}}) + \frac{\delta}{2p_{\max}}.$$

Then, using the optimality of problem equation 31, it holds that

$$\mathcal{L}_{\text{ERM}}(f^\epsilon_{\text{PAIR}}) \leq \mathcal{L}_{\text{ERM}}(\hat{f}^\epsilon_{\text{PAIR}}) \leq \mathcal{L}_{\text{ERM}}(f^\epsilon_{\text{PAIR}}) + \frac{\delta}{2p_{\max}}.$$

It remains to analyze the sample complexity of ensuring that the training set $\mathcal{D}_{\text{tr}}$ is $\frac{\delta}{4p_{\max}}$-representative w.r.t. $\mathcal{X}, \mathcal{F}, \mathcal{D}$ and all the ERM and OOD losses $\{\mathcal{L}_{\text{ERM}}, \boldsymbol{L}_{\text{ood}}\}$.

For any $i \in I_{\text{PAIR}}$, based on Assumption E.3, we can write $\mathcal{L}_i(f) = \mathbb{E}_{(X,Y)\sim\mathcal{D}}[\ell_i(f(X), Y)]$ and $\widehat{\mathcal{L}}_i(f) = \frac{1}{|D|}\sum_{(X_j,Y_j)\in D}\ell_i(f(X_j), Y_j)$ with $|\ell_i(f(X), Y)| \leq L_i \leq L_{\max}, \forall f, X, Y$. Using Hoeffding's inequality, we can conclude that for any $f \in \mathcal{F}$,

$$\Pr\left[\left|\widehat{\mathcal{L}}_i(f) - \mathcal{L}_i(f)\right| \geq \frac{\delta}{4p_{\max}}\right] \leq 2\exp\left(\frac{-|D|\delta^2}{32L_{\max}^2 p_{\max}^2}\right).$$

Thus, for any $\gamma \in (0, 1)$, if we require

$$|D| \geq \frac{32L_{\max}^2 p_{\max}^2}{\delta^2}\log\frac{2(m+1)|\mathcal{F}|}{\gamma},$$

it holds that

$$\Pr\left[\exists f \in \mathcal{F}, \left|\widehat{\mathcal{L}}_i(f) - \mathcal{L}_i(f)\right| \geq \frac{\delta}{4p_{\max}}\right] \leq \sum_{f\in\mathcal{F}}\Pr\left[\left|\widehat{\mathcal{L}}_i(f) - \mathcal{L}_i(f)\right| \geq \frac{\delta}{4p_{\max}}\right] \leq \frac{\gamma}{m+1}.$$

Thus,

$$\Pr\left[\exists i \in I_{\text{PAIR}}, \exists f \in \mathcal{F}, \left|\widehat{\mathcal{L}}_i(f) - \mathcal{L}_i(f)\right| \geq \frac{\delta}{4p_{\max}}\right]$$

$$\leq \sum_{i\in I_{\text{PAIR}}}\Pr\left[\exists f \in \mathcal{F}, \left|\widehat{\mathcal{L}}_i(f) - \mathcal{L}_i(f)\right| \geq \frac{\delta}{4p_{\max}}\right] \leq \gamma.$$

Finally, we can conclude that with probability at least $1 - \gamma$, $\forall i \in I_{\text{PAIR}}, \forall f \in \mathcal{F}$,

$$\left|\widehat{\mathcal{L}}_i(f) - \mathcal{L}_i(f)\right| \leq \frac{\delta}{4p_{\max}},$$

which completes the proof. □

**Remarks.** The $\epsilon$-approximated formulation has a close relationship to another relaxation as the following.

$$\boldsymbol{f}_{\text{PAIR}} \coloneqq \underset{f\in\mathcal{F}}{\arg\min} \quad \mathcal{L}_{\text{ERM}}(f)$$

$$\text{s.t.} \qquad \mathcal{L}_{\text{PAIR}}^i(f) \leq \epsilon_i, \forall i \in [m].$$

Essentially, both the $\epsilon$-approximated formulation and the above formulation are natural relaxation of the original problem (Eq. 30 or Eq. 16). As the $\epsilon_i \to \epsilon_{\text{ood}\,i}$, the above formulation also yields the optimal solution $f_{\text{ood}}$. In this work, since we focus on the approximations on the preference, $\epsilon$-approximated formulation essentially provides a convenient touch which could be of independent interests for future discussions.

## F   MORE DETAILS ON EXPERIMENTS

In this section, we provide more details about the experiments (Sec. 5) in the main paper.

### F.1   MORE DETAILS ON COLOREDMNIST EXPERIMENTS

In the proof-of-concept experiments with COLOREDMNIST, we follow the evaluation settings as IRM (Arjovsky et al., 2019) and the test-domain selection as DomainBed (Gulrajani & Lopez-Paz, 2021). Specifically, we use a 4-Layer MLP with a hidden dimension of 256. By default, we use Adam Kingma & Ba (2015) optimizer with a learning rate of $1e-3$ and a weight decay of $1e-3$ to train the model with 500 epochs and select the last epoch as the output model for each hyperparameter setting. We choose the final model from different hyperparameter setups as the one that maximizes the accuracy on the validation that share the same distribution as test domain. We then do grid search for the corresponding hyperparameters. For pretraining epochs, we search from $\{0, 50, 100, 150, 200, 250\}$. For OOD penalty, we search from $\{1e1, 1e2, 1e3, 1e4, 1e5\}$. We evaluate each configuration of hyperparameters 10 times and report the mean and standard deviation of the performances. Besides, for IRMv1, we will refresh the history in Adam optimizer when the pretraining finishes, following the practice in Gulrajani & Lopez-Paz (2021). We also empirically

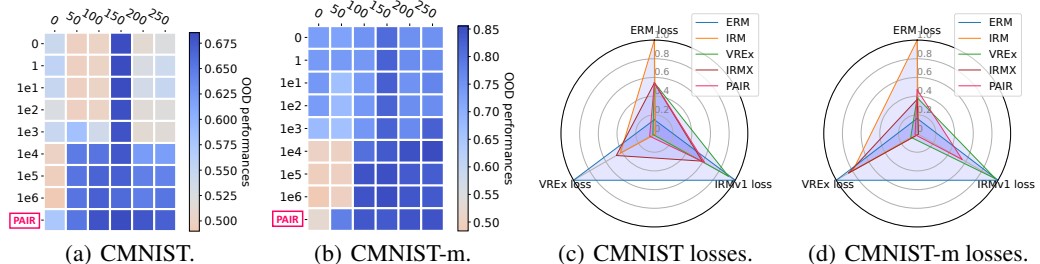

(a) CMNIST.    (b) CMNIST-m.    (c) CMNIST losses.    (d) CMNIST-m losses.

Figure 16: (a),(b) PAIR can effectively find a better solution than exhaustive tuning of penalty weights in IRMX. That is because PAIR can adaptively adjust the penalty weights during the optimization process, and leads to a Pareto optimal solution, as shown in (c),(d).

find that refreshing the optimizer after pretraining can bring a better performance of IRMv1 in COLOREDMNIST. While for VREx, we find the refreshing is not needed.

For the implementation of IRMX, we change the penalty to be the sum of IRMv1 and VREx losses and conduct the same hyperparameter search as for IRMv1 for fair comparison. As for the implementation of PAIR, we use SGD with a momentum of 0.9 (Sutskever et al., 2013) after pretraining, to avoid the interference of Adam to the gradient direction and convergence of EPO (Mahapatra & Rajan, 2020) solver. Moreover, we also empirically find that SGD requires larger learning rate (we search over two choices, i.e., 0.01 and 0.1) for approaching the direction. This is because of the design in EPO solver that it first fits to the preference direction then does the "pure" gradient descent, while the intrinsically conflicting directions pointed by the objectives can make the loss surface more steep. We will leave in-depth understanding of the above phenomenon and more sophisticated optimizer design in more complex tasks and network architectures to future works (Zhao & Zhang, 2015; Zhou et al., 2020).

## F.2 MORE DETAILS ABOUT ABLATION STUDIES

**Comparison between PAIR-o and the linear weighting scheme under exhaustive parameter search.** In the main paper, to investigate how PAIR-o can find a better OOD solution under objective conflicts, we first conduct a ablation study to compare the OOD performances of PAIR-o and the exhaustive tuned IRMX. Specifically, we tune both IRMv1 and VREx penalty weights from a substantially larger scope, i.e., $\{1, 1e1, 1e2, 1e3, 1e4, 1e5, 1e6\}$. As for pretraining epochs, we search from $\{0, 50, 100, 150, 200, 250\}$. The results of IRMX in COLOREDMNIST and the modified COLOREDMNIST are shown as in Fig. 16(a) and Fig. 16(b), respectively. Each point represents the best performed IRMX with the configuration of the corresponding pretraining epoch, the IRMv1 penalty weight and different VREx penalty weights from $\{1, 1e1, 1e2, 1e3, 1e4, 1e5, 1e6\}$.

We also present a full exhaustive hyperparameter tunning study based on linear weighting scheme for IRMX, shown in Fig. 17, where we further enlarge the search space of penalty weights from $1e6$ to $1e12$ to better compare with IRMX optimized via PAIR-o. Similar to Fig. 16(a) and Fig. 16(b), each point in Fig. 17 is selected from *best performed* models trained with the corresponding IRMv1 penalty weights, and pretraining epoch, and all possible VREx penalty weights from $\{1, 1e1, 1e2, 1e3, 1e4, 1e5, 1e6, 1e7, 1e8, 1e9, 1e10, 1e11, 1e12\}$.

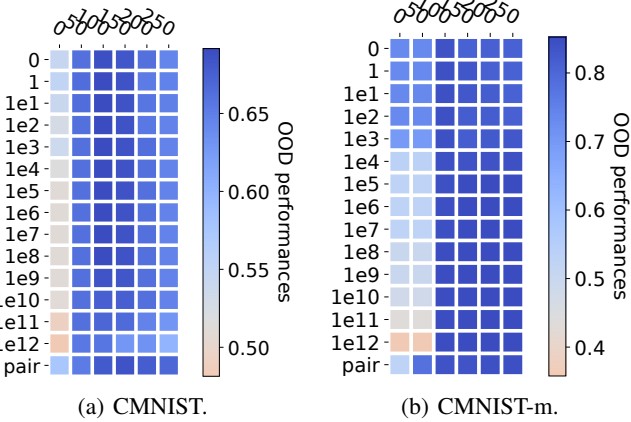

(a) CMNIST.    (b) CMNIST-m.

Figure 17: Full exhaustive hyperparameter tunning study

Compared to IRMv1 shown as in Fig. 8, IRMX can substantially improve the OOD performances in both COLOREDMNIST and the modified COLOREDMNIST, confirming our theoretical results. However, the OOD performances of IRMX turn out to be upper bounded by that optimized with PAIR-o at each pretraining epochs. In other words, PAIR-o requires substantially less parameter tuning efforts to achieve the top OOD performances, confirming the advances of PAIR-o. In more complex tasks where the exhaustive parameter tunning is prohibitively expensive, such as in the experiments with WILDS (Koh et al., 2021), IRMX performs worse than PAIR, which further validates the effectiveness of PAIR-o.

To better demonstrate the advantages of PAIR-o over linear weighting scheme, we replicate the previous study in two datasets from WILDS, i.e., CIVILCOMMENTS and FMoW. Due to the computational resource limits, we limit the search scope of IRMv1 and VREx to $\{1e-2, 1, 1e2\}$, respectively. It can be found that, even with a broader hyperparameter search space, IRMX optimized via linear weighting scheme remain under-performed than PAIR-o.

Table 5: Comparison between linear weighting scheme and PAIR-o in WILDS.

| CIVILCOMMENTS | IRMv1\VREx | $1e-2$ | $1$ | $1e2$ | FMoW | IRMv1\VREX | $1e-2$ | $1$ | $1e2$ |
|---|---|---|---|---|---|---|---|---|---|
| | $1e-2$ | 72.5($\pm2.00$) | 73.8($\pm1.40$) | 73.1($\pm0.67$) | | $1e-2$ | 33.64($\pm0.59$) | 34.20($\pm1.33$) | 34.43($\pm0.72$) |
| | 1 | 73.5($\pm1.47$) | 74.3($\pm0.83$) | 73.2($\pm0.67$) | | 1 | 30.25($\pm0.87$) | 33.75($\pm0.78$) | 33.7($\pm0.78$) |
| | $1e2$ | 72.1($\pm0.59$) | 70.1($\pm2.09$) | 74.3($\pm0.51$) | | $1e2$ | 21.33($\pm1.51$) | 21.00($\pm2.41$) | 13.14($\pm1.63$) |
| PAIR-o | | | **75.2**($\pm0.7$) | | | | | **35.5**($\pm1.13$) | |

**Loss values distribution at convergence.** As for the loss distribution experiments (Fig. 16(c), 16(d)), we plot the ERM,IRMv1 and VREx loss values at convergence of best performed algorithms. The plotted values are in log-scale and normalized to $[0, 1]$. It can be found that PAIR-o effectively find a better solution in terms of IRMv1 and VREx losses, while not generating the ERM performances too much, which confirms our motivations for the design of PAIR.

**Penalty weights trajectory.** To examine whether PAIR-o can effectively adjust the penalty weights of ERM and OOD objectives, especially when the model has not arrived at the Pareto front (i.e., the gradient conflicts are expected to be more intense), we plot the trajectories of penalty weights generated by PAIR-o in both CMNIST and CMNIST-m, shown as in Fig. 18.

It can be found that the whole training process can be divided into three phases: "Fitting" phase; "Adaption" phase; and "Generalization" phase. In the "Fitting" phase, the model is trained with only the ERM objectives and is expected to approach the Pareto front first (cf. Fig. 15). It also corresponds to the "descent" phase in the PAIR-o algorithm, hence the penalty weight for ERM objective is 1 while for OOD objective is 0.

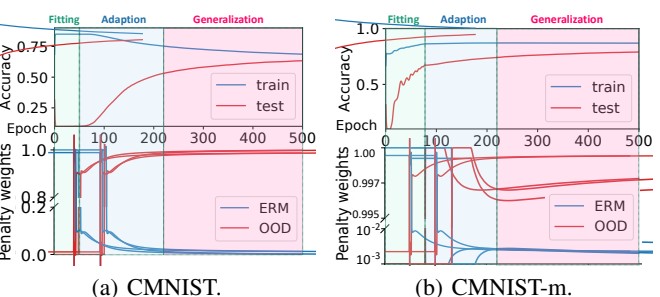

(a) CMNIST.  (b) CMNIST-m.

Figure 18: Penalty weights trajectory

Then, when PAIR-o enters into the "balance" phase, PAIR-o begins to yield high weights to OOD objectives, while not diminishing the weights to ERM objectives. That is the "Adaption" phase, where PAIR-o begins to adjust the solution towards the Pareto front as well as the preferred direction. When the solution is close to the Pareto front, then PAIR-o enters into the "Generalization" phase. That is to incorporate the invariance into the features by assigning high weights to the OOD objectives.

**Preference sensitivity analysis under strict hyperparameter configuration.** Another reason for the high performance of PAIR-o at both COLOREDMNIST and realistic datasets from WILDS is because of its robustness to different preference choices. In complementary to the theoretical discussion in Theorem E.1, we also conducted preference sensitivity analysis experiments under strict hyperparameter configurations. In other words, the hyperparameter search space is restricted to *single* point, i.e., a learning rate of 0.01, and a pretraining epoch of 150. The results are shown in Fig. 19 for both the original and the modified COLOREDMNIST dataset. It can be found that, PAIR-o maintains high performance and robustness to different preference choices.

It also aligns with our discussion on preference choice in practice (Sec. D.3), that we need to assign a higher preference to *robust, and more easy-to-optimize* objectives, i.e., VREx. When the relative preferences are given within a reasonable scale, PAIR-o easily yields top OOD performances.

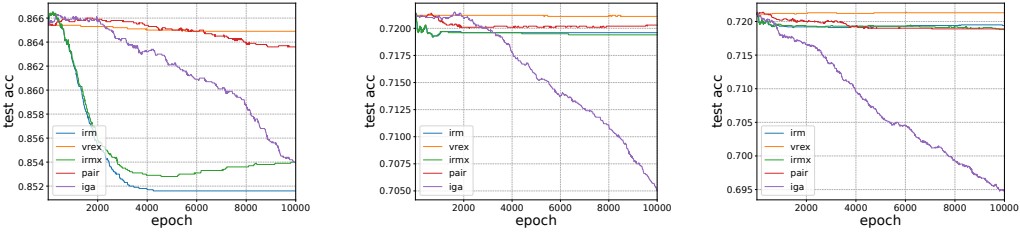

(a) CMNIST.  (b) CMNIST-m.

Figure 19: Preference sensitivity under *strict* hyperparameter configuration. $x$-axis is the preference for VREx while $y$-axis is the preference for IRMv1

**Additional ablation study on COLOREDMNIST with "perfect" initialization.** We also conduct experiments with "perfect" initializations for different methods, to check whether the OOD constraints can enforce the invariance, following Zhang et al. (2022a). Besides the OOD methods used in the paper, we also include another OOD method IGA (Koyama & Yamaguchi, 2020) to give a more comprehensive overview of their performances with "perfect" initialization. We also introduce another variant of ColoredMNIST, i.e., **CMNIST-11**: $\{(0.25, 0.10), (0.25, 0.20)\}$ to complement more details. All methods are initialized with a ERM model learned on gray-scale ColoredMNIST data which is expected to learn to use digit shapes in the image to make predictions. The learning rate is $1e-3$ and the penalty weight is $1e5$. Different from Zhang et al. (2022a), we use SGD to optimize the models, as Adam would generate larger step sizes when the gradients continue to be within a small range under the "perfect" initialization. Results are shown as in Fig. 20.

(a) "Perfect" init. on CMNIST-10.  (b) "Perfect" init. on CMNIST-11.  (c) "Perfect" init. on CMNIST-25.

Figure 20: OOD performances with "Perfect" initializations.

It can be found that, in CMNIST-10, IRM, IRMx and IGA cannot enforce the invariance while V-REx and PAIR maintain the invariance, which is consistent to our previous findings. Moreover, IGA fails to maintain the invariance in CMNIST-11 and CMNIST-25, demonstrating the relatively low robustness of IGA objective. Besides, V-REx consistently maintain the invariance even in CMNIST-11, for the reason that the gradient signals of variance in "perfect" initialization tend to vanish. In contrast, PAIR improve over both IRM and IRMx to maintain the invariance, confirming the effectiveness of PAIR.

**Additional ablation study on the performance of PAIR-o and PAIR-s with more OOD objectives and their composite with IRMv1.** Besides VREx, we conduct additional ablation studies of PAIR with IB (Ahuja et al., 2021a), Fishr (Rame et al., 2021), CLOvE (Wald et al., 2021), IGA (Koyama & Yamaguchi, 2020) and SD (Pezeshki et al., 2021), based on COLOREDMNIST and the modified COLOREDMNIST. We focus on the cases with no less than 2 OOD objectives, as one could simply obtain a low OOD loss for single OOD objective, where linear weighting scheme is likely to approach the desired OOD solution as the Pareto front is simpler. However, it is often the case that single OOD objective is not sufficiently robust to locate the desired OOD solution to the Pareto front.

In experiments, we follow the same evaluation protocol as previous experiments on COLOREDM-NIST. Due to the resource limits of NVIDIA RTX 3090Ti used for the original COLOREDMNIST experiments in previous sections, we switch the hardware and software platform to Linux servers with NVIDIA V100 graphics cards with CUDA 10.2, hence the results in Table 6 and Table 7 are not directly comparable with those in Table 1. Similar to previous experiments, for the stability

of MOO solver under heterogeneous objectives, we search learning rate for VREx and Fishr from $\{0.01, 0.02, 0.04, 0.1, 0.2\}$ at stage 2 while a larger scope $\{0.1, 0.2, 0.4, 0.8, 1\}$ for other objectives. Note that even considering the learning rate into the hyperparameter search space, PAIR still uses a smaller scope than that of linear weighting scheme. Besides, we follow our previous discussion in Appendix D.3 to set up the preference of different OOD objectives. Specifically, for Fishr, we use a larger preference of $1e12$ than that for IRMv1 ($1e8$), since the agreements based methods tend to have a smaller loss than IRMv1. While for the other objectives, we use a smaller preference of $1e8$ than that for IRMv1 ($1e12$). Note that this is only a heuristic setup and the performance of PAIR can be further improved if the preferences can be tuned.

Table 6: Generality study of PAIR for IRMv1 with other objectives in COLOREDMNIST.

| | IRMv1 | PAIR-o | PAIR-s | CMNIST | CMNIST-M | Avg. | ΔAvg. |
|---|---|---|---|---|---|---|---|
| ERM | | | | 17.14(±0.73) | 73.30(±0.85) | 45.22 | |
| IRMv1 | | | | 67.29(±0.99) | 76.89(±3.23) | 72.09 | +0.00 |
| IB | | | | 55.48(±3.67) | 76.01(±0.58) | 65.75 | |
| | ✓ | | | 56.09(±2.04) | 75.66(±10.6) | 65.88 | −6.21 |
| | ✓ | ✓ | | 61.12(±2.33) | 83.30(±3.00) | 72.21 | +0.12 |
| | ✓ | ✓ | ✓ | 60.69(±2.26) | 83.70(±1.79) | 72.20 | +0.11 |
| VREx | | | | 68.62(±0.73) | 83.52(±2.52) | 76.07 | |
| | ✓ | | | 66.19(±1.41) | 81.75(±1.68) | 73.97 | +1.88 |
| | ✓ | ✓ | | 68.89(±1.13) | 83.80(±1.60) | 76.35 | +4.26 |
| | ✓ | ✓ | ✓ | 69.16(±0.76) | **83.96**(±1.65) | **76.56** | +4.47 |
| Fishr | | | | **69.38**(±0.39) | 77.29(±1.61) | 73.34 | |
| | ✓ | | | 66.20(±2.31) | 81.07(±3.98) | 73.63 | +1.54 |
| | ✓ | ✓ | | 68.90(±0.56) | 82.70(±1.09) | 75.80 | +2.49 |
| | ✓ | ✓ | ✓ | 68.78(±0.78) | 84.02(±1.37) | 76.40 | +3.31 |
| CLOvE | | | | 55.55(±9.97) | 74.20(±2.45) | 64.88 | |
| | ✓ | | | 66.35(±1.51) | 77.70(±1.00) | 72.02 | −0.07 |
| | ✓ | ✓ | | 64.99(±2.29) | 75.70(±1.05) | 70.35 | −1.75 |
| | ✓ | ✓ | ✓ | 65.55(±2.17) | 77.29(±1.55) | 71.42 | −0.67 |
| IGA | | | | 58.67(±7.69) | 76.27(±1.01) | 68.97 | |
| | ✓ | | | 51.22(±3.67) | 74.20(±2.45) | 62.71 | −9.38 |
| | ✓ | ✓ | | 66.17(±2.34) | 81.84(±3.09) | 74.01 | +1.91 |
| | ✓ | ✓ | ✓ | 66.51(±0.78) | 82.12(±3.04) | 74.32 | +2.23 |
| SD | | | | 62.31(±1.54) | 76.73(±0.90) | 69.52 | |
| | ✓ | | | 62.48(±1.25) | 81.24(±0.69) | 71.86 | −0.23 |
| | ✓ | ✓ | | 59.52(±6.12) | 82.82(±0.64) | 71.17 | −0.92 |
| | ✓ | ✓ | ✓ | 65.54(±0.91) | 83.57(±0.81) | 74.56 | +2.47 |
| Oracle | | | | 72.08(±0.24) | 86.53(±0.14) | 79.31 | 79.31 |

The results are given in Table. 6. It can be found that, not all OOD objectives can improve IRMv1 performance. For the OOD objectives that can enhance the OOD robustness when incorporated into IRMv1, PAIR can further improve over the combined OOD objectives optimized via linear weighting scheme. While for unrobust combinations, intuitively it is hard to improve the OOD performance for the following reasons:

(i). When the new objective combination is unrobust, the desired solution may not lie in the new Pareto optimal front;

(ii). Eventhough the desired solution lies in the new Pareto optimal front, the weakened OOD robustness introduces more local minimals that have low OOD losses while worse OOD generalization performance;

(iii). As an extra objective is involved, the OOD preference used in PAIR tends to have a higher divergence from the ideal one;

Therefore, given unrobust OOD objective combinations, the performance gain of PAIR is not theoretically guaranteed. Nevertheless, PAIR-o can still improve some of the unrobust objective

combinations, demonstrating its robustness. Notably, `PAIR-s` can further improve the performance of `PAIR-o` at most cases, demonstrating the generality of `PAIR`.

To study what OOD objectives are suitable to be combined with IRMv1 and whether using more OOD objectives can bring more performance improvements, additionally, we conduct experiments with all possible composites of IRMv1 and IB (Ahuja et al., 2021a), Fishr (Rame et al., 2021) and VREx (Krueger et al., 2021). In experiments, similar as in previous study, `PAIR-o` adopts a slightly broader learning rate search scope of $\{0.01, 0.02, 0.04, 0.1, 0.2\}$ at stage 2, in order to prevent divergence. Note that even considering the learning rate into the hyperparameter search space, `PAIR` still uses a smaller search scope than that of linear weighting scheme. `PAIR-s` adopts the training domain validation accuracy to perform the model selection. Both `PAIR-o` and `PAIR-s` adopts a heuristic preference setup that uses a decreasing preference from $1e12$ to $1e8$ by a step size of $1e2$ for more objectives. For example, in the composite of IB, IRMv1 and VREx, we adopt the preference of $(1e8, 1e10, 1e12)$ for the OOD objectives. The choice of preference follows previous discussion in Appendix D.3.

Table 7: Generality study of `PAIR` for composite objectives in COLOREDMNIST.

| | IB | VREx | Fishr | CMNIST | CMNIST-M | Avg. | $\Delta$ Avg. |
|---|---|---|---|---|---|---|---|
| ERM | | | | $17.14(\pm 0.73)$ | $73.30(\pm 0.85)$ | 45.22 | |
| IRMv1 | | | | $67.29(\pm 0.99)$ | $76.89(\pm 3.23)$ | 72.09 | $+0.00$ |
| Linear | ✓ | | | $56.09(\pm 2.04)$ | $75.66(\pm 10.6)$ | 65.88 | $-6.21$ |
| +PAIR-o | ✓ | | | $61.12(\pm 2.33)$ | $83.30(\pm 3.00)$ | 72.21 | $+0.12$ |
| +PAIR-o +PAIR-s | ✓ | | | $60.69(\pm 2.26)$ | $83.70(\pm 1.79)$ | 72.20 | $+0.11$ |
| Linear | | ✓ | | $66.19(\pm 1.41)$ | $81.75(\pm 1.68)$ | 73.97 | $+1.88$ |
| +PAIR-o | | ✓ | | $68.89(\pm 1.13)$ | $83.80(\pm 1.60)$ | 76.35 | $+4.26$ |
| +PAIR-o +PAIR-s | | ✓ | | $\mathbf{69.16}(\pm 0.76)$ | $\underline{83.96}(\pm 1.65)$ | $\underline{76.56}$ | $\underline{+4.47}$ |
| Linear | | | ✓ | $66.20(\pm 2.31)$ | $81.07(\pm 3.98)$ | 73.63 | $+1.54$ |
| +PAIR-o | | | ✓ | $66.45(\pm 0.90)$ | $82.70(\pm 1.09)$ | 74.58 | $+2.49$ |
| +PAIR-o +PAIR-s | | | ✓ | $67.57(\pm 0.81)$ | $83.22(\pm 2.10)$ | 75.40 | $+3.31$ |
| Linear | ✓ | ✓ | | $52.61(\pm 1.56)$ | $63.84(\pm 1.08)$ | 58.23 | $-13.9$ |
| +PAIR-o | ✓ | ✓ | | $68.35(\pm 1.73)$ | $81.25(\pm 3.08)$ | 74.80 | $+2.71$ |
| +PAIR-o +PAIR-s | ✓ | ✓ | | $\underline{69.05}(\pm 0.76)$ | $83.11(\pm 1.46)$ | 76.08 | $+3.99$ |
| Linear | ✓ | | ✓ | $51.91(\pm 1.26)$ | $68.88(\pm 3.22)$ | 60.39 | $-11.7$ |
| +PAIR-o | ✓ | | ✓ | $59.70(\pm 12.7)$ | $74.59(\pm 1.11)$ | 67.15 | $-4.94$ |
| +PAIR-o +PAIR-s | ✓ | | ✓ | $66.98(\pm 2.66)$ | $75.91(\pm 3.50)$ | 71.45 | $-0.65$ |
| Linear | | ✓ | ✓ | $64.83(\pm 2.95)$ | $79.34(\pm 5.77)$ | 72.09 | $+0.00$ |
| +PAIR-o | | ✓ | ✓ | $67.96(\pm 1.60)$ | $81.44(\pm 2.24)$ | 74.70 | $+2.61$ |
| +PAIR-o +PAIR-s | | ✓ | ✓ | $68.19(\pm 1.58)$ | $81.89(\pm 3.01)$ | 75.04 | $+2.95$ |
| Linear | ✓ | ✓ | ✓ | $50.00(\pm 0.32)$ | $69.60(\pm 2.33)$ | 59.80 | $-12.3$ |
| +PAIR-o | ✓ | ✓ | ✓ | $66.89(\pm 1.80)$ | $83.46(\pm 3.10)$ | 75.18 | $+3.08$ |
| +PAIR-o +PAIR-s | ✓ | ✓ | ✓ | $68.59(\pm 1.29)$ | $\mathbf{85.30}(\pm 0.64)$ | $\mathbf{76.95}$ | $\mathbf{+4.85}$ |
| Oracle | | | | $72.08(\pm 0.24)$ | $86.53(\pm 0.14)$ | 79.31 | |

The results are shown in Table 7. The best and second best results are in bold and underline, respectively. It can be found that incorporating more OOD objectives does not necessarily bring more performance improvements into IRMv1. The linear weighting scheme can further exacerbate the performance drops of unrobust OOD objective combinations. For example, when incorporating IB objective into IRMv1, the OOD performance drops, since IB is proposed to mitigate a specific type of distribution shifts instead of directly improving learning the invariance in the original IRMv1 setting. In contrast, it can be found that incorporating Fishr can bring performance increases at most cases. The reason is that minimizing Fishr loss can approximately minimizes the VREx loss, as shown by Rame et al. (2021). Therefore, we suspect that the reason for the performance drop could be that more objectives will make the Pareto front more complicated, and also lead to higher divergence of the OOD preference (since we are less likely know the ideal preference given more objectives). Hence, the preferred composition of the objectives are preferred to those that have theoretical guarantees and are as concise as possible.

Interestingly, we also find that, although incorporating more objectives in `PAIR-o` does not necessarily bring performance increase, a combination of `PAIR-o` and `PAIR-s` can further improve the OOD performance, despite of the simple implementation of `PAIR-o`. It serves as strong evidence for the generality and significance of `PAIR`.

### F.3 MORE DETAILS ABOUT EXPERIMENTS ON WILDS

In this section, we provide more details about the WILDS datasets as well as the evaluation setups in the experiments.

#### F.3.1 DATASET DESCRIPTION.

We select 6 challenging datasets from WILDS (Koh et al., 2021) benchmark for evaluating `PAIR-o` performance in realistic distribution shifts. The datasets cover from domain distribution shifts, subpopulation shifts and the their mixed. A summary of the basic information and statistics of the WILDS datasets can be found in Table. 8, Table. 9, respectively. In the following, we will give a brief introduction to each of the datasets. More details can be found in the WILDS paper (Koh et al., 2021).

Table 8: A summary of datasets information from WILDS.

| Dataset | Data ($x$) | Class information | Domains | Metric | Architecture |
|---|---|---|---|---|---|
| CAMELYON17 | Tissue slides | Tumor (2 classes) | 5 hospitals | Avg. acc. | DenseNet-121 |
| CIVILCOMMENTS | Online comments | Toxicity (2 classes) | 8 demographic groups | Wr. group acc. | DistillBERT |
| FMoW | Satellite images | Land use (62 classes) | 16 years x 5 regions | Wr. group acc. | DenseNet-121 |
| iWILDCAM | Photos | Animal species (186 classes) | 324 locations | Macro F1 | ResNet-50 |
| POVERTY | Satellite images | Asset (real valued) | 23 countries | Wr. group Pearson (r) | Resnet-18 |
| RxRx1 | Cell images | Genetic treatments (1,139 classes) | 51 experimental batches | Avg. acc | ResNet-50 |

Table 9: A summary of datasets statistics from WILDS.

| Dataset | # Examples | | | # Domains | | |
|---|---|---|---|---|---|---|
| | train | val | test | train | val | test |
| CAMELYON17 | 302,436 | 34,904 | 85,054 | 3 | 1 | 1 |
| CIVILCOMMENTS | 269,038 | 45,180 | 133,782 | - | - | - |
| FMoW | 76,863 | 19,915 | 22,108 | 11 | 3 | 2 |
| iWILDCAM | 129,809 | 14,961 | 42,791 | 243 | 32 | 48 |
| POVERTY | 10,000 | 4,000 | 4,000 | 13-14 | 4-5 | 4-5 |
| RxRx1 | 40,612 | 9,854 | 34,432 | 33 | 4 | 14 |

**Camelyon17.** We follow the WILDS splits and data processing pipeline for the Camelyon17 dataset (Bándi et al., 2019). It provides $450,000$ lymph-node scans from 5 hospitals. The task in Camelyon17 is to take the input of $96 \times 96$ medical images to predict whether there exists a tumor tissue in the image. The domains $d$ refers to the index of the hospital where the image was taken. The training data are sampled from the first 3 hospitals where the OOD validation and test data are sampled from the 4-th and 5-th hospital, respectively. We will use the average accuracy as the evaluation metric and a DenseNet-121 (Huang et al., 2017) as the backbone for the featurizer.

**CivilComments.** We follow the WILDS splits and data processing pipeline for the CivilComments dataset (Borkan et al., 2019). It provides $450,000$ comments collected from online articles. The task is to classify whether an online comment text is toxic or non-toxic. The domains $d$ are defined according to the demographic features, including male, female, LGBTQ, Christian, Muslim, other religions, Black, and White. CivilComments is used to study the subpopulation shifts, here we will use the worst group/domain accuracy as the evaluation metric. As for the backbone of the featurizer, we will use a DistillBert (Sanh et al., 2019) following WILDS (Koh et al., 2021).

**FMoW.** We follow the WILDS splits and data processing pipeline for the FMoW dataset (Christie et al., 2018). It provides satellite images from 16 years and 5 regions. The task in FMoW is to classify the images into 62 classes of building or land use categories. The domain is split according to the year that the satellite image was collected, as well as the regions in the image which could be Africa, America, Asia, Europe or Oceania. Distribution shifts could happen across different years and regions. The training data contains data collected before 2013, while the validation data contains

images collected within 2013 to 2015, and the test data contains images collected after 2015. The evaluation metric for FMoW is the worst region accuracy and the backbone model for the featurizer is a DenseNet-121 (Huang et al., 2017).

**iWildCam.** We follow the WILDS splits and data processing pipeline for the iWildCam dataset (Beery et al., 2020). It is consist of $203, 029$ heat or motion-activated photos of animal specifies from $323$ different camera traps across different countries around the world. The task of iWildCam is to classify the corresponding animal specifies in the photos. The domains is split according to the locations of the camera traps which could introduce the distribution shifts. We will use the Macro F1 as the evaluation metric and a ResNet-50 (He et al., 2016) as the backbone for the featurizer.

**PovertyMap.** We follow the WILDS splits and data processing pipeline for the PovertyMap dataset (Yeh et al., 2020). It consists of satellite imagery and survey data at $19, 669$ villages from 23 African countries between 2009 and 2016. Different from other datasets, the task in PovertyMap is a regression task that asks the model to predict the real-valued asset wealth index computed from Demographic and Health Surveys (DHS) data. The domain is split according to the countries that the image was taken and whether the image is of an urban or rural area. The relative small size of PoverMap allows for using cross-fold evaluation, where each fold defines a different set of OOD countries (Koh et al., 2021). We will use the Pearson correlation of the worst urban/rural subpopulation as the evaluation metric and a ResNet-18 (He et al., 2016) as the backbone for the featurizer.

**RxRx1.** We follow the WILDS splits and data processing pipeline for the RxRx1 dataset (Taylor et al., 2019). The input is an image of cells taken by fluorescent microscopy. The cells can be genetically perturbed by siRNA and the task of RxRx1 is to predict the class of the corresponding siRNA that have treated the cells. There exists $1, 139$ genetic treatments and the domain shifts are introduced by the experimental batches. We will use the average accuracy of the OOD experimental batches as the evaluation metric and a ResNet-50 (He et al., 2016) as the backbone for the featurizer.

### F.3.2 TRAINING AND EVALUATION DETAILS.

We follow previous works to implement and evaluate our models (Koh et al., 2021; Shi et al., 2022; Yao et al., 2022). The information of the referred paper and code is listed as in Table. 10.

Table 10: The information of the referred paper and code.

| Paper | Commit | Code |
|---|---|---|
| WILDS (Koh et al., 2021) | v2.0.0 | https://wilds.stanford.edu/ |
| Fish (Shi et al., 2022) | 333efa24572d99da0a4107ab9cc4af93a915d2a9 | https://github.com/YugeTen/fish |
| LISA (Yao et al., 2022) | bc424c47df6f072986b63cd906c44975bd34d9ff | https://github.com/huaxiuyao/LISA |

The general hyperparemter setting inherit from the referred codes and papers, and are shown as in Table 11. We use the same backbone models to implement the featurizer (He et al., 2016; Huang et al., 2017; Sanh et al., 2019). By default, we repeat the experiments by 3 runs with the random seeds of $0, 1, 2$. While for Camelyon17, we follow the official guide to repeat 10 times with the random seeds from 0 to 9, and for PovertyMap, we repeat the experiments 5 times with the random seeds from 0 to 4. Note that the PovertyMap use cross-fold validations hence each run will use different training and evaluation splits, following the WILDS official guideline.

For the evaluation of baselines, we refer the previous results from the literature (Koh et al., 2021; Shi et al., 2022; Yao et al., 2022) by default, while we rerun Fish (Shi et al., 2022) and LISA (Yao et al., 2022) to validate the reported results. Since the original implementation of Fish does not support the evaluation of the updated PovertyMap dataset, we mildly adjust the hyperparameter settings to reproduce the corresponding results as shown in Table. 11. We also reduce the batch size on FMoW due to the memory limits and we find it does not affect the reproducibility of Fish and LISA. Besides, since the original implementation of LISA does not support PovertyMap, which differentiates as a regression task that could be not suitable with Mixup (Zhang et al., 2018), however we find the "group by label" strategy in LISA works particularly well and reaches to the state of the art. For IRMX, we implement it as the simple addition of IRMv1 and VREx penalties based on the Fish implementation (Shi et al., 2022), and search the penalty weights using the same space as for other objectives (Koh et al., 2021) to ensure the fairness. Besides, since previously reported results did not

cover the performance of VREx in iWildCam and PovertyMap, we implement VREx and report the results based on the Fish implementation (Shi et al., 2022).

Table 11: General hyperparameter settings for the experiments on WILDS.

| Dataset | CAMELYON17 | CIVILCOMMENTS | FMoW | iWILDCAM | POVERTYMAP | RxRx1 |
|---|---|---|---|---|---|---|
| Num. of seeds | 10 | 3 | 3 | 3 | 5 | 3 |
| Learning rate | 1e-4 | 2e-6 | 1e-4 | 1e-4 | 1e-4 | 1e-3 |
| Weight decay | 0 | 0.01 | 0 | 0 | 0 | 1e-5 |
| Scheduler | n/a | n/a | n/a | n/a | n/a | Cosine Warmup |
| Batch size | 32 | 16 | 32 | 16 | 64 | 72 |
| Architecture | DenseNet121 | DistilBert | DenseNet121 | ResNet50 | ResNet18 | ResNet50 |
| Optimizer | SGD | Adam | Adam | Adam | Adam | Adam |
| Pretraing Step | 10000 | 20000 | 24000 | 24000 | 5000 | 15000 |
| Maximum Epoch | 2 | 5 | 12 | 9 | 200 | 90 |

For PAIR-o, we implement it based on the Fish code (Shi et al., 2022). The detailed algorithm can be found in Algorithm. 1. We leverage the same number of pretraining steps as in Fish to fulfill the first "descent" phase in PAIR-o algorithm. Then, during the "balance" phase, at each training step, we sampled $k$ batches of data from different domains, calculate loss and conduct the back-propagation. By default, we use only the gradients of the classifier to solve for the objective weights during the "balance" phase. Except for iWildCam and RxRx1 datasets, due the memory limits, as discussed in Sec. D.4.1, we use the freeze technique to ensure the consistency of batch size and number of sampled domains as in Table. 11. Moreover, as discussed in Sec. D.4.2, the unbiased stochastic estimate of IRMv1 penalties can not guarantee the non-negativity of the estimated loss values, which are however not compatible with MOO theory (Kaisa, 1999) (thus the same for PAIR-o). Therefore, we will manually adjust the negative values to be positive, by multiplying it with a adjustment rate (short in Neg. IRMv1 adj. rate in Table. 12). The adjustment rate is tuned from 1 to $1e-4$ with a step size of $1e-2$ to avoid the training divergence and instability. Following the discussion as in Sec. D.3, we tune the OOD relative preference by merely varying the preference for IRMv1 objective from the default choice of $(1, 1e10, 1e12)$ by a step size of $1e2$. We find the performances of IRMv1 and VREx highly correlate to the corresponding relative preference weights. We list hyperparameters of PAIR-o in Table 12. Although we did not tune the hyperparameters heavily, we find that PAIR-o generically works well across different challenging datasets and realistic distribution shifts on WILDS. As discussed in Sec. D.3, there could be more sophisticated approaches to further improve the search and estimate of OOD preference, which we will leave for future developments based on PAIR.

Table 12: Hyperparameter settings of PAIR-o for the experiments on WILDS.

| Dataset | CAMELYON17 | CIVILCOMMENTS | FMoW | iWILDCAM | POVERTYMAP | RxRx1 |
|---|---|---|---|---|---|---|
| Gradients from | Classifier | Classifier | Classifier | Classifier | Classifier | Classifier |
| Freeze featurizer | No | No | No | Yes | No | Yes |
| Relative Preference | (1,1e12,1e12) | (1,1e8,1e12) | (1,1e12,1e12) | (1,1e10,1e12) | (1,1e8,1e12) | (1,1e8,1e12) |
| Neg. IRMv1 adj. rate | 1 | 1e-4 | 1 | 1e-2 | 1e-2 | 1 |
| Group by | Hospitals | Demographics× toxicity | Times × regions | Trap locations | Countries | Experimental batches |
| Sampled domains | 3 | 5 | 5 | 10 | 5 | 10 |

## F.4 SOFTWARE AND HARDWARE

We implement our methods with PyTorch (Paszke et al., 2019). For the software and hardware configurations, we ensure the consistent environments for each datasets. Specifically, we run COLOREDMNIST experiments on Linux Servers with NVIDIA RTX 3090Ti graphics cards with CUDA 11.3, 40 cores Intel(R) Xeon(R) Silver 4114 CPU @ 2.20GHz, 256 GB Memory, and Ubuntu 18.04 LTS installed. While for WILDS and DOMAINBED experiments, we run on Linux servers with NVIDIA V100 graphics cards with CUDA 10.2.

# G MORE DETAILS OF MODEL SELECTION RESULTS ON DOMAINBED

## G.1 INTRODUCTION OF DIFFICULT MODEL SELECTION IN DOMAINBED

DOMAINBED is proposed by Gulrajani & Lopez-Paz (2021) to highlight the importance of model selection in OOD generalization. Specifically, they empirically show that, under rigorous hyperpa-

rameter tunning, ERM (Vapnik, 1991) achieves the state-of-the-art performances. Although recently progress are made to outperform ERM under the rigorous DOMAINBED evaluation protocol (Rame et al., 2021), whether there exists a proper model selection for OOD algorithms remains elusive.

The difficulty of a proper model selection for OOD algorithms is mainly because of: We lack the access to a validation set that have a similar distribution with the test data. Therefore, Gulrajani & Lopez-Paz (2021) provide 3 options to choose and construct a validation set from: training domain data; leave-one-out validation data; test domain data. However, all three validation set construction approaches have their own limitations, as they essentially posit different assumptions on the test distribution (Gulrajani & Lopez-Paz, 2021; Teney et al., 2021; Rame et al., 2021).

PAIR-s tries to address the limitations caused by the difficulty of finding a proper validation set for model selection in domain generalization, by leveraging the *prior* assumed within the OOD algorithm. Essentially, different lines of OOD algorithms discussed in Sec. B.1 adopt different prior and assumptions on the causes of the distribution shifts. The main purpose of the OOD evaluation is to validate the correctness of the posed assumptions. To this end, the selected models should properly reflect the preferences implied by the assumptions, i.e., the OOD loss values. When considering the loss values during the model selection, it is natural to leverage the MOO perspective and explicitly consider the trade-offs between ERM and OOD performance. The detailed description, implementation options, and potential leverages of PAIR-s are provided in Appendix D.

## G.2 TRAINING AND EVALUATION DETAILS

Since our main purpose of the DOMAINBED experiments is to validate the existence of the problem and the effectiveness of PAIR-s, we apply PAIR-s to the representative methods of the four discussed OOD solutions in Sec. B.1. Specifically, we choose the following four methods out of all implemented algorithms in DOMAINBED (https://github.com/facebookresearch/DomainBed):

- ERM: Empirical Risk Minimization (Vapnik, 1991)
- IRM: Invariant Risk Minimization (Arjovsky et al., 2019)
- GroupDRO: Group Distributionally Robust Optimization (Sagawa* et al., 2020)
- DANN: Domain Adversarial Neural Network (Ganin et al., 2016)
- Fishr: Invariant Gradient Variances for OOD Generalization (Rame et al., 2021)

Due to the limits of computational resources, we select 3 out of 7 datasets from DOMAINBED. We refer Rame et al. (2021) to prescribe the detail, listed as follows:

1. Colored MNIST (Arjovsky et al., 2019) is a variant of the MNIST handwritten digit classification dataset (Lecun et al., 1998). Domain $d \in \{90\%, 80\%, 10\%\}$ contains a disjoint set of digits colored: the correlation strengths between color and label vary across domains. The dataset contains 70,000 examples of dimension $(2, 28, 28)$ and 2 classes. Most importantly, the network, the hyperparameters, the image shapes, etc. are **not** the same as in the IRM setup for COLOREDMNIST experiments.

2. PACS (Li et al., 2017) includes domains $d \in \{\text{art, cartoons, photos, sketches}\}$, with 9,991 examples of dimension $(3, 224, 224)$ and 7 classes.

3. TerraIncognita (Beery et al., 2018) contains photographs of wild animals taken by camera traps at locations $d \in \{\text{L100, L38, L43, L46}\}$, with 24,788 examples of dimension $(3, 224, 224)$ and 10 classes.

Note that CMNIST dataset in DOMAINBED use a convolutional neural network as the backbone for the featurizer, which is not the same MLP for COLOREDMNIST experiments. By default, all real datasets leverage a ResNet-50 (He et al., 2016) pretrained on ImageNet, with a dropout layer before the newly added dense layer and fine-tuned with frozen batch normalization layers.

During the training, we strictly follow the evaluation protocol in DOMAINBED. Note that the hyperparameter configurations of Fishr have some differences from the default configurations hence we refer the configuration tables by Rame et al. (2021) directly, shown as follows.

Table 13: **Hyperparameters**, their default values and distributions for random search (Gulrajani & Lopez-Paz, 2021; Rame et al., 2021).

| Condition | Parameter | Default value | Random distribution |
|---|---|---|---|
| PACS/ TERRAINCOGNITA | learning rate | 0.00005 | $10^{\text{Uniform}(-5,-3.5)}$ |
| | batch size | 32 | $2^{\text{Uniform}(3,5.5)}$ if not DomainNet else $2^{\text{Uniform}(3,5)}$ |
| | weight decay | 0 | $10^{\text{Uniform}(-6,-2)}$ |
| | dropout | 0 | RandomChoice $([0, 0.1, 0.5])$ |
| COLOREDMNIST | learning rate | 0.001 | $10^{\text{Uniform}(-4.5,-3.5)}$ |
| | batch size | 64 | $2^{\text{Uniform}(3,9)}$ |
| | weight decay | 0 | 0 |
| All | steps | 5000 | 5000 |
| Fishr | regularization strength $\lambda$ | 1000 | $10^{\text{Uniform}(1,4)}$ |
| | ema $\gamma$ | 0.95 | Uniform$(0.9, 0.99)$ |
| | warmup iterations | 1500 | Uniform$(0, 5000)$ |

As for the construction of the validation set, we test with training domain validation set and test domain validation set, as leave-one-out domain selection requires more runs and more computational resources that are out of our limits. Specifically, to construct the validation set, the data from each domain will be first splitted into $80\%$ (for training and evaluation) and $20\%$ (for validation and model selection). For training domain validation set, the validation data is consist of the $20\%$ split from each training domain. While for the test domain validation set, the validation data is consist of the $20\%$ split from each test domain.

The whole evaluation will be repeated 3 times where in each repeat, there will be 20 samplings of hyperparameters from the distribution shown in Table 13. Therefore, there will be 20 runs in each repeat and there will be 1 model selected from the 20 runs.

For the implementation of `PAIR-s`, we follow the algorithm as in Algorithm 2. Since training domain validation accuracy tends to be a more unreliable indicator than test domain validation accuracy, i.e., has a worse reflection of the OOD generalization performance due to the high similarity with the training data (Teney et al., 2021), during the selection within each run, we filter out the models before the last 5 steps in COLOREDMNIST and the last 10 steps in PACS and TERRAINCOGNITA. During the selection within one repeat (across different runs), we use a percent of $50\%$ for step 9 in Algorithm 2 and finalize the selection according the PAIR score. Except for GroupDRO and DANN of which the objective value tend to have higher variance and relatively low OOD robustness, we aggregate the models within each repeat by the validation accuracy. In contrast, for the test domain validation accuracy, we filter out the models before the last 5 steps for DANN while 10 steps for others according to the robustness of the objectives during the selection within each run. During the selection within one repeat (across different runs), we directly adopt the validation accuracy to finalize the model selected. Note that Gulrajani & Lopez-Paz (2021) argue that test domain validation is more likely to be a invalid benchmarking methodology, since it requires access to the test domain which is usually inaccessible in realistic applications.

For the selection of loss values $L$, we use the values reported solely at each logging step, which is evaluated every 100 steps with a minibtach of the training data, listed as follows:

- ERM: N/A.
- IRM: ERM and IRMv1 (`nll,penalty`).
- GroupDRO: Worst group ERM loss (`losses.min()`).
- DANN: Weighted ERM and domain discrimination loss (`gen_loss`).
- Fishr: ERM and Fishr penalty (`nll,penalty`).

### G.3 FULL DOMAINBED RESULTS

In this section, we provide full results of the DOMAINBED experiments. To begin with, we first present the overall results of the three datasets, including the averages and the improvements of the worst

domain accuracies, as in Table. 14 and Table. 15. From results we can seed that `PAIR-s` consistently improves the OOD performance across all datasets and validation set options. Remarkably, in the most challenging setting that uses train domain validation set on COLOREDMNIST, `PAIR-s` improves the worst domain performances of IRMv1 and Fishr by a large margin up tp 14.3%. In the realistic dataset PACS, `PAIR-s` improves the worst domain performances of IRMv1 by a large margin up to 7.3%. In TERRAINCOGNITA, `PAIR-s` improves the worst domain performances of DANN by a large margin up to 3.1%. Besides the worst domain performance, `PAIR-s` improves the average domain performances up to 1.0% and empower the OOD methods to reach new state-of-the-arts.

When using the test domain validation set, since the validation set itself could reflect the OOD generalization performance, therefore the improvements could be lower. When comes to OOD objectives that have a relatively low robustness, the worst domain performance could be lower.

We also report the detailed results at each domain with the variance in the next section.

### G.3.1 OVERALL RESULTS

Table 14: Overeall OOD generalization performances using training domain validation accuracy.

| | PAIR-s | COLOREDMNIST | | PACS | | TERRAINCOGNITA | | Overall |
|---|---|---|---|---|---|---|---|---|
| | | Avg. acc | $\triangle$ wr. acc | Avg. acc | $\triangle$ wr. acc | Avg. acc | $\triangle$ wr. acc | Avg. acc |
| ERM | | $51.4 \pm 1.0$ | | $84.8 \pm 0.3$ | | $44.6 \pm 1.1$ | | 60.2 |
| DANN | | $51.5 \pm 0.1$ | | $82.5 \pm 0.8$ | | $44.9 \pm 0.9$ | | 59.6 |
| DANN | ✓ | $51.9 \pm 0.1$ | +0.9 | $83.3 \pm 0.5$ | +0.7 | $44.5 \pm 1.5$ | +3.1 | 59.9 |
| GroupDRO | | $51.8 \pm 0.0$ | | $84.1 \pm 0.8$ | | $46.6 \pm 1.1$ | | 60.8 |
| GroupDRO | ✓ | $53.0 \pm 0.4$ | +3.1 | $84.4 \pm 0.7$ | +1.1 | $46.6 \pm 1.1$ | +0.0 | 61.3 |
| IRM | | $51.6 \pm 0.1$ | | $83.5 \pm 1.1$ | | $44.9 \pm 0.3$ | | 60.0 |
| IRM | ✓ | $52.2 \pm 0.5$ | +14.3 | $85.1 \pm 0.9$ | +7.3 | $41.1 \pm 3.8$ | +1.4 | 59.5 |
| Fishr | | $51.8 \pm 0.1$ | | $85.6 \pm 0.5$ | | $47.0 \pm 1.4$ | | 61.5 |
| Fishr | ✓ | $54.2 \pm 1.0$ | +12.7 | $85.6 \pm 0.1$ | +1.1 | $47.7 \pm 1.1$ | +0.3 | 62.5 |

Table 15: Overeall OOD generalization performances using test domain validation accuracy.

| | PAIR-s | COLOREDMNIST | | PACS | | TERRAINCOGNITA | | Overall |
|---|---|---|---|---|---|---|---|---|
| | | Avg. acc | $\triangle$ wr. acc | Avg. acc | $\triangle$ wr. acc | Avg. acc | $\triangle$ wr. acc | Avg. acc |
| ERM | | $57.8 \pm 0.2$ | | $87.0 \pm 0.1$ | | $52.9 \pm 0.9$ | | 65.9 |
| DANN | | $57.4 \pm 0.8$ | | $84.7 \pm 0.5$ | | $50.8 \pm 0.3$ | | 64.3 |
| DANN | ✓ | $56.2 \pm 1.1$ | -2.6 | $85.7 \pm 0.2$ | +2.2 | $50.7 \pm 0.5$ | +0.4 | 64.2 |
| GroupDRO | | $61.3 \pm 0.4$ | | $86.9 \pm 0.0$ | | $52.5 \pm 0.2$ | | 66.9 |
| GroupDRO | ✓ | $60.1 \pm 0.7$ | -4.3 | $87.3 \pm 0.2$ | +1.8 | $52.0 \pm 0.7$ | +0.6 | 66.4 |
| IRM | | $68.1 \pm 1.6$ | | $84.4 \pm 0.5$ | | $49.2 \pm 0.6$ | | 67.2 |
| IRM | ✓ | $69.0 \pm 1.1$ | +2.9 | $86.0 \pm 0.4$ | +0.8 | $50.7 \pm 0.9$ | +0.4 | 68.6 |
| Fishr | | $68.0 \pm 2.9$ | | $87.5 \pm 0.1$ | | $53.7 \pm 0.2$ | | 69.7 |
| Fishr | ✓ | $68.2 \pm 3.0$ | +0.6 | $87.4 \pm 0.1$ | +0.6 | $52.1 \pm 0.7$ | -0.5 | 69.2 |

### G.3.2 TRAINING DOMAIN VALIDATION SET

Table 16: OOD generalization performances with training domain validation set on COLOREDMNIST.

| Algorithm | PAIR-s | +90% | +80% | -90% | Avg | △ wr. acc |
|---|---|---|---|---|---|---|
| ERM | | $71.0 \pm 0.5$ | $73.4 \pm 0.1$ | $10.0 \pm 0.1$ | 51.5 | |
| DANN | | $71.0 \pm 0.3$ | $73.4 \pm 0.1$ | $10.0 \pm 0.1$ | 51.5 | |
| DANN | ✓ | $71.6 \pm 0.3$ | $73.3 \pm 0.2$ | $10.9 \pm 0.4$ | 51.9 | +0.9 |
| GroupDRO | | $72.6 \pm 0.2$ | $73.1 \pm 0.0$ | $9.9 \pm 0.1$ | 51.8 | |
| GroupDRO | ✓ | $72.7 \pm 0.2$ | $73.2 \pm 0.5$ | $13.0 \pm 1.5$ | 53.0 | +3.1 |
| IRM | | $72.3 \pm 0.3$ | $72.6 \pm 0.4$ | $9.9 \pm 0.1$ | 51.6 | |
| IRM | ✓ | $67.4 \pm 2.6$ | $64.8 \pm 1.4$ | $24.2 \pm 1.6$ | 52.2 | +14.3 |
| Fishr | | $72.2 \pm 0.6$ | $73.1 \pm 0.3$ | $9.9 \pm 0.2$ | 51.8 | |
| Fishr | ✓ | $69.1 \pm 2.9$ | $70.9 \pm 1.7$ | $22.6 \pm 1.4$ | 54.2 | +12.7 |

Table 17: OOD generalization performances with training domain validation set on PACS.

| Algorithm | PAIR-s | A | C | P | S | Avg | △ wr. acc |
|---|---|---|---|---|---|---|---|
| ERM | | $82.6 \pm 1.6$ | $79.2 \pm 1.0$ | $97.2 \pm 0.5$ | $74.9 \pm 2.6$ | 83.5 | |
| DANN | | $84.7 \pm 1.8$ | $75.8 \pm 0.9$ | $97.3 \pm 0.1$ | $72.3 \pm 1.0$ | 82.5 | |
| DANN | ✓ | $86.5 \pm 0.9$ | $77.0 \pm 1.8$ | $97.0 \pm 0.2$ | $73.0 \pm 0.5$ | 83.3 | +0.7 |
| GroupDRO | | $83.4 \pm 1.7$ | $77.1 \pm 0.3$ | $97.6 \pm 0.2$ | $78.2 \pm 1.3$ | 84.1 | |
| GroupDRO | ✓ | $83.4 \pm 1.7$ | $78.3 \pm 0.3$ | $97.6 \pm 0.2$ | $78.2 \pm 1.3$ | 84.4 | +1.1 |
| IRM | | $82.9 \pm 2.6$ | $81.4 \pm 0.1$ | $96.7 \pm 0.6$ | $73.1 \pm 3.1$ | 83.5 | |
| IRM | ✓ | $82.4 \pm 2.3$ | $80.5 \pm 0.8$ | $97.2 \pm 0.2$ | $80.4 \pm 1.3$ | 85.1 | +7.3 |
| Fishr | | $85.3 \pm 1.1$ | $80.3 \pm 1.1$ | $97.9 \pm 0.3$ | $79.1 \pm 1.7$ | 85.6 | |
| Fishr | ✓ | $85.4 \pm 1.4$ | $80.2 \pm 0.8$ | $96.2 \pm 0.7$ | $80.5 \pm 0.8$ | 85.6 | +1.1 |

Table 18: OOD generalization performances with training domain validation set on TERRAINCOGNITA.

| Algorithm | PAIR-s | L100 | L38 | L43 | L46 | Avg | △ wr. acc |
|---|---|---|---|---|---|---|---|
| ERM | | $46.7 \pm 3.5$ | $41.8 \pm 1.0$ | $57.4 \pm 1.0$ | $39.7 \pm 0.2$ | 46.4 | |
| DANN | | $46.1 \pm 3.5$ | $41.2 \pm 1.0$ | $56.7 \pm 0.9$ | $35.6 \pm 1.1$ | 44.9 | |
| DANN | ✓ | $43.1 \pm 3.8$ | $41.1 \pm 0.9$ | $55.2 \pm 2.1$ | $38.7 \pm 1.9$ | 44.5 | +3.1 |
| GroupDRO | | $48.4 \pm 2.9$ | $40.3 \pm 3.1$ | $57.9 \pm 2.2$ | $40.0 \pm 0.5$ | 46.6 | |
| GroupDRO | ✓ | $48.4 \pm 2.9$ | $40.3 \pm 3.1$ | $57.9 \pm 2.2$ | $40.0 \pm 0.5$ | 46.6 | +0.0 |
| IRM | | $48.4 \pm 3.8$ | $35.6 \pm 2.9$ | $55.4 \pm 0.9$ | $40.1 \pm 1.4$ | 44.9 | |
| IRM | ✓ | $40.4 \pm 7.3$ | $38.3 \pm 2.5$ | $48.8 \pm 6.3$ | $37.0 \pm 0.9$ | 41.1 | +1.4 |
| Fishr | | $49.2 \pm 4.4$ | $40.6 \pm 1.4$ | $57.9 \pm 1.1$ | $40.4 \pm 1.2$ | 47.0 | |
| Fishr | ✓ | $51.0 \pm 3.3$ | $40.7 \pm 1.3$ | $58.2 \pm 0.1$ | $40.8 \pm 1.2$ | 47.7 | +0.3 |

### G.3.3 TEST DOMAIN VALIDATION SET

Table 19: OOD generalization performances with test domain validation set on COLOREDMNIST.

| Algorithm | PAIR-s | +90% | +80% | -90% | Avg | △ wr. acc |
|---|---|---|---|---|---|---|
| ERM | | 71.7 ± 0.2 | 72.7 ± 0.2 | 28.8 ± 0.8 | 57.8 | |
| DANN | | 73.0 ± 1.2 | 73.3 ± 0.1 | 25.8 ± 1.7 | 57.4 | |
| DANN | ✓ | 72.1 ± 0.3 | 73.2 ± 0.3 | 23.2 ± 3.8 | 56.2 | -2.6 |
| GroupDRO | | 73.4 ± 0.4 | 72.4 ± 0.0 | 38.1 ± 0.8 | 61.3 | |
| GroupDRO | ✓ | 73.2 ± 0.2 | 73.3 ± 0.3 | 33.8 ± 2.3 | 60.1 | -4.3 |
| IRM | | 72.3 ± 0.3 | 72.5 ± 0.4 | 59.4 ± 5.3 | 68.1 | |
| IRM | ✓ | 71.7 ± 0.4 | 73.1 ± 0.1 | 62.3 ± 3.1 | 69.0 | +2.9 |
| Fishr | | 73.8 ± 0.5 | 73.6 ± 0.1 | 56.7 ± 8.6 | 68.0 | |
| Fishr | ✓ | 73.7 ± 0.6 | 73.5 ± 0.2 | 57.3 ± 8.4 | 68.2 | +0.6 |

Table 20: OOD generalization performances with test domain validation set on PACS.

| Algorithm | PAIR-s | A | C | P | S | Avg | △ wr. acc |
|---|---|---|---|---|---|---|---|
| ERM | | 86.6 ± 0.7 | 82.5 ± 0.8 | 97.3 ± 0.5 | 81.8 ± 0.7 | 87.0 | |
| DANN | | 86.5 ± 0.8 | 79.9 ± 0.4 | 97.1 ± 0.1 | 75.3 ± 1.1 | 84.7 | |
| DANN | ✓ | 87.0 ± 0.2 | 81.4 ± 0.7 | 96.8 ± 0.5 | 77.5 ± 1.3 | 85.7 | +2.2 |
| GroupDRO | | 87.7 ± 0.4 | 82.1 ± 0.7 | 98.0 ± 0.2 | 79.6 ± 0.7 | 86.9 | |
| GroupDRO | ✓ | 86.7 ± 0.3 | 83.2 ± 1.1 | 97.8 ± 0.1 | 81.4 ± 0.5 | 87.3 | +1.8 |
| IRM | | 82.3 ± 1.5 | 80.8 ± 0.7 | 95.8 ± 1.3 | 78.9 ± 1.4 | 84.4 | |
| IRM | ✓ | 85.3 ± 0.3 | 81.7 ± 0.9 | 97.4 ± 0.3 | 79.7 ± 1.8 | 86.0 | +0.8 |
| Fishr | | 88.4 ± 0.4 | 82.2 ± 0.7 | 97.7 ± 0.5 | 81.6 ± 0.4 | 87.5 | |
| Fishr | ✓ | 87.4 ± 0.8 | 82.6 ± 0.5 | 97.5 ± 0.6 | 82.2 ± 0.0 | 87.4 | +0.6 |

Table 21: OOD generalization performances with test domain validation set on TERRAINCOGNITA.

| Algorithm | PAIR-s | L100 | L38 | L43 | L46 | Avg | △ wr. acc |
|---|---|---|---|---|---|---|---|
| ERM | | 58.7 ± 1.7 | 51.3 ± 1.8 | 59.9 ± 0.6 | 41.7 ± 1.0 | 52.9 | |
| DANN | | 53.8 ± 0.5 | 47.4 ± 1.0 | 59.0 ± 0.5 | 42.9 ± 0.3 | 50.8 | |
| DANN | ✓ | 54.4 ± 1.3 | 46.9 ± 1.2 | 58.1 ± 0.2 | 43.3 ± 0.0 | 50.7 | +0.4 |
| GroupDRO | | 57.3 ± 0.4 | 50.4 ± 1.1 | 59.7 ± 0.7 | 42.8 ± 0.7 | 52.5 | |
| GroupDRO | ✓ | 55.9 ± 3.2 | 50.6 ± 0.7 | 57.9 ± 0.4 | 43.4 ± 0.4 | 52.0 | +0.6 |
| IRM | | 53.6 ± 0.5 | 47.9 ± 1.9 | 54.1 ± 0.9 | 41.3 ± 0.6 | 49.2 | |
| IRM | ✓ | 59.3 ± 1.8 | 45.5 ± 0.6 | 56.4 ± 1.7 | 41.7 ± 0.7 | 50.7 | +0.4 |
| Fishr | | 60.7 ± 0.8 | 49.4 ± 0.7 | 59.5 ± 0.5 | 45.0 ± 0.5 | 53.7 | |
| Fishr | ✓ | 58.9 ± 1.0 | 46.4 ± 1.8 | 58.6 ± 0.7 | 44.5 ± 0.8 | 52.1 | -0.5 |

