# OpenReview forum: "Pareto Invariant Risk Minimization: Towards Mitigating the Optimization Dilemma in Out-of-Distribution Generalization"
_ICLR.cc/2023/Conference — ICLR 2023 poster_

### Official Review · Reviewer_1jAA · 2022-10-23

**Confidence:** 3
**Correctness:** 4
**Technical Novelty And Significance:** 3
**Empirical Novelty And Significance:** 3
**Recommendation:** 8

**Clarity, Quality, Novelty And Reproducibility:**

- The paper is well-written and flows logically. See weaknesses above about novelty.


**Strength And Weaknesses:**

*Strengths*
- The paper motivates its work via two key problems with domain generalization methods, most of which can be seen as regularized versions of ERM training. Namely, that the surrogate objectives may be inappropriate and that the regularized solutions do not find Pareto optimal solutions.

- The examples and illustrations are well-constructed and instructive for understanding some of the key issues.

- The results demonstrate the promise of using the proposed PAIR-o and PAIR-s schemes for better domain generalization. Particularly, PAIR-s can be applied non-IRM DG methods.

*Weaknesses*
- (Minor weakness) It seems odd that this paper focuses on IRM, as IRM has been often shown to be suboptimal and not necessarily SOTA.  It seems that the overall framework of Pareto optimization might be better applied to more SOTA domain generalization methods.

- (Minor weakness) The novelty It seems that the problems with domain generalization and MOO methods are parts of prior work. Is the main novelty in synthesizing these two areas? I do think it's useful but it seemed the "key problems" (i.e., poor IRM approximations and pareto efficiency) and the MOO method were not novel by themselves. Perhaps this is just a problem of distinguishing the paper's contributions and prior work.


**Summary Of The Paper:**

This paper motivates the need for Pareto optimal solutions from the fact that most domain generalization methods can be seen as regularized ERM problems that struggle to outperform ERM.
The paper suggests that this is due to two key problems. One is the approximation of the original OOD objective (often encoded as a regularization). The other is due to using linear penalty methods for approximating a Pareto optimal solution, which is known to only work if the frontier is convex.
Thus, the paper proposes to formulate domain generalization methods as multi-objective optimization problems with ERM as one objective and the OOD objectives as additional objectives.
Then, multiple gradient descent method for multi-objective optimization is applied.
The empirical results demonstrate the benefits of the proposed approach.


**Summary Of The Review:**

Overall, the perspective of multi-objective optimization for domain generalization is a fresh take on domain generalization and provides practical performance benefits. The clarity of writing and illustrations are very well done and will aid in the works impact. The work could be improved by clarifying its contributions with respect to prior work in IRM variants and MOO. Also, it would be interesting to apply this more generally to other OOD methods.

---

> ### Author Response · Authors · 2022-11-11
> **Response to Reviewer 1jAA**
>
> Thank you for your positive feedbacks and thoughtful comments. Please see our detailed responses to your questions and suggestions below.
>
> **Q1.  It seems odd that this paper focuses on IRM. It seems that the overall framework of Pareto optimization might be better applied to more SOTA domain generalization methods.**
>
> **R1.** We choose IRMv1 to study the OOD optimization problem, as i) The framework is one of the most representative approaches that incorporates causality in representation learning for OOD generalization. Many of the recently SOTA algorithms have a close relationship with IRM; ii) There hasn’t been a OOD method that consistently and significantly outperforms IRMv1 across all benchmarks; iii) The clear relationship between relaxed IRMv1 and causal invariance enables us easy to examine the effectiveness of our solution; iv) The OOD literature has a plenty of studies around IRMv1 drawbacks [1,2,3], so our study could be recognized by the community more easily.
>
> Following your suggestion, we also conduct experiments with other state-of-the-art OOD objectives, such as IB[3], IGA[4], CLOvE[5], SD[6], and Fishr[7], as well as  with all possible combinations of IB, VREx[8] and Fishr. Details can be found in the previous reply([exp. A](https://openreview.net/forum?id=esFxSb_0pSL&noteId=mTZiMqEo--4), [exp. B](https://openreview.net/forum?id=esFxSb_0pSL&noteId=cKBDeD_feq)), as well as in Table 6 and Table 7 in Appendix F.2.
> The results show that, incorporating other state-of-the-art objectives can actually lead to better OOD performance when optimized by PAIR, only if the objective combinations are theoretically sound. The empirical trends further demonstrate the effectiveness and generality of PAIR-o and PAIR-s. Beyond the examined combinations, we are looking forward to applying PAIR to more promising objectives as the field grows.
>
> **Q2. The work could be improved by clarifying its contributions with respect to prior work in IRM variants and MOO.**
>
> **R2.** We revised the discussions with related works to better distinguish our work from IRM variants and MOO. In short, most of the IRM variants focus on developing alternative objectives to learn the causal invariance, while they neglect the OOD optimization problem. Although many MOO methods gained success in resolving objective conflicts for multi-task learning, whether and how we can adopt MOO to model and mitigate the ERM and OOD conflicts remain underexplored.
>
> We hope our responses could clarify your concerns. Please let us know if you have any further questions.
>
> **References**
>
> [1] Rosenfeld et al., The Risks of Invariant Risk Minimization, ICLR 2021.
>
> [2] Kamath et al., Does invariant risk minimization capture invariance? AISTATS 2021.
>
> [3] Ahuja et al., Invariance Principle Meets Information Bottleneck for Out-of-Distribution Generalization, NeurIPS 2021.
>
> [4] Koyama and Yamaguchi, Out-of-distribution generalization with maximal invariant predictor, arXiv 2020.
>
> [5] Wald et al., On Calibration and Out-of-domain Generalization, NeurIPS 2021.
>
> [6] Pezeshki et al., Gradient Starvation: A Learning Proclivity in Neural Networks, NeurIPS 2021.
>
> [7] Rame et al., Fishr: Invariant Gradient Variances for Out-of-Distribution Generalization, ICML 2022.
>
> [8] Kruger et al., Out-of-Distribution Generalization via Risk Extrapolation (REx), ICML 2021.

---

> > ### Comment · Reviewer_1jAA · 2022-11-16
> > **Thanks for the response, one disagreement**
> >
> > Hi authors,
> >
> > Thank you for the response. Overall, I appreciate the response particularly in adding comparisons with other OOD objectives as I do think it is helpful, and I continue to hold my accept recommendation.
> >
> > One minor comment: I disagree with the statement "there hasn’t been a OOD method that consistently and significantly outperforms IRMv1 across all benchmarks".  This is either a trivial statement (i.e., it could be said of any OOD method because no OOD method "significantly" outperforms any method on "all" benchmarks) or a misleading statement.  For example, if you look at the public WILDS leaderboard for DG without unlabeled data, IRM is not the best method for ANY dataset: https://wilds.stanford.edu/leaderboard/ .

---

> > > ### Author Response · Authors · 2022-11-16
> > > **Further response to Reviewer 1jAA**
> > >
> > > Hi Reviewer 1jAA,
> > >
> > > We thank you very much for your knowledgeable comments and consistent support of our work! We agree that IRMv1 is not the empirically best OOD solution. We also believe that, with your support, the results and understandings developed in PAIR, aligned with previous studies in the community, are taking solid steps towards resolving the OOD optimization dilemma!

---

### Official Review · Reviewer_hPqV · 2022-10-23

**Confidence:** 3
**Correctness:** 3
**Technical Novelty And Significance:** 3
**Empirical Novelty And Significance:** Not applicable
**Recommendation:** 6

**Clarity, Quality, Novelty And Reproducibility:**

The authors analyzed potential issues of previous invariant risk minimization (IRM) formulations and their variants and proposed a multi-objective optimization scheme, named as PAreto Invariant Risk minimization (PAIR) to help improve generalizability and OOD performances.

Probably due to the page limit and significant amount of theoretical and empirical results to report, the authors may want to consider better organizing the presentation to clearly describe the contributions. Math notations as well as many acronyms may need to be clearly defined throughout the submission too.

**Strength And Weaknesses:**


The elaborated discussion on the practical and theoretical failures of the IRM variants (with relaxations IRMs, IRMX, VREX) is clear and motivating the proposed development. Illustrations, proof-of-concept as well as empirical benchmark results have also demonstrated the potential of the PAIR framework.

Due to the limited review time, this reviewer may have misunderstood some points but here are some potential issues which the authors may consider addressing.

1. Mathematical notations may need to be revised to be consistent. For example, the operators changed from $\circ$ in equation (1) to $\cdot$ in equations (2) and (3). Some notations were not defined throughout the submission. Probably due to the page limit, however, Section 4.1 is the part where the details of the methods were described but since the algorithm is also in the Appendix, maybe better description for the methods can help improve readability of the submission.

2. PAIR considers a composite OOD loss in the MOO setting. What other OOD objective combinations can be made,  which will lead to better performance? Can we make a structured analysis of which objectives should be combined in MOO?

3. It appears that the theoretical analysis for classification was mostly on binary classification problems. While empirical experiments on benchmarks are with multi-class classification problems. The authors may want to state clearly about the generalizability of the theoretical results.

4. The discussion in Appendix C.3 may also be mentioned in the main paper to clarify where to use PAIR-o and PAIR-s. Can optimization and model selection ideas be combined? Will it be useful in any settings or provide superior performance?

5. The empirical results, in particularly in Tables 2 and 3, did not always show consistent improvements and the improvements are not with a huge margin, if any. More discussions on the observed trends may also help better understand possible gaps.




**Summary Of The Paper:**

This submission analyzes and extends previous invariant risk minimization (IRM) formulations to a multi-objective optimization scheme, named as PAreto Invariant Risk minimization (PAIR) for finding models generalizing well and with good OOD performance. The authors motivated the work by analyzing how IRM may fail to achieve good generalizability and OOD performances and providing practical and theoretical failures of the IRM variants (relaxations). They discussed the importance of the trade-offs between ERM and OOD objectives and choosing desired solutions in Pareto optimal frontier and  developed PAIR-o, as an optimizer for OOD generalization and PAIR-s for model selection by leveraging the previous multi-objective optimization (MOO) solver EPO from the literature. Experimental results on several OOD benchmarks demonstrated the efficacy of the proposed PAIR framework.




**Summary Of The Review:**

This submission analyzes and extends previous invariant risk minimization (IRM) formulations to a multi-objective optimization scheme, named as PAreto Invariant Risk minimization (PAIR) for finding models generalizing well and good OOD performance, with both solid theoretical and empirical results. The authors leveraged the existing MOO solver for PAIR and it may be more interesting if new solution algorithms can be derived. The presentation may need to be significantly improved as detailed in the above concerns.

---

> ### Author Response · Authors · 2022-11-11
> **Response to Reviewer hPqV [2/2]**
>
> Please let us know if you have any further questions. We’d be grateful if you could take the above responses into consideration when making the final evaluation of our work.
>
>
> **References**
>
> [1] Arjovsky et al., Invariant Risk Minimization, arXiv 2020.
>
> [2] Kamath et al., Does invariant risk minimization capture invariance? AISTATS 2021.
>
> [3] Ahuja et al., Invariance Principle Meets Information Bottleneck for Out-of-Distribution Generalization, NeurIPS 2021.
>
> [4] Koyama and Yamaguchi, Out-of-distribution generalization with maximal invariant predictor, arXiv 2020.
>
> [5] Wald et al., On Calibration and Out-of-domain Generalization, NeurIPS 2021.
>
> [6] Pezeshki et al., Gradient Starvation: A Learning Proclivity in Neural Networks, NeurIPS 2021.
>
> [7] Kruger et al., Out-of-Distribution Generalization via Risk Extrapolation (REx), ICML 2021.
>
> [8] Rame et al., Fishr: Invariant Gradient Variances for Out-of-Distribution Generalization, ICML 2022.
>
> [9] Liu and Vicente, The Stochastic Multi-gradient Algorithm for Multi-objective Optimization and its Application to Supervised Machine Learning. Annals of Operations Research, 2021.

---

> ### Author Response · Authors · 2022-11-11
> **Response to Reviewer hPqV [1/2]**
>
> Thank you for taking the time to review our paper and your detailed comments. Please see our detailed responses to your questions and suggestions below.
>
> **Q1. Mathematical notations may need to be revised to be consistent. maybe better description for the methods can help improve readability of the submission.**
>
> **R1.** We follow the notations as in previous works[1,2] when deriving the IRMv1 drawbacks. The operators changed from $\circ$ in Eq. 1 to $\cdot$ in Eq. 2 is because we impose the linearity constraint to $\varphi$ and $w$, as stated in the paper.
> Nevertheless, we understand that the definitions and notations in our paper may be a bit heavy for the general audience, hence we added a unified table to Appendix A that defines all the key concepts and mathematical notations. We also improved the description of our method in Sec. 4.1 to better present the idea of PAIR.
>
> **Q2. What other OOD objective combinations can be made, which will lead to better performance? Can we make a structured analysis of which objectives should be combined in MOO?**
>
> **R2.** We thank for the insightful question. Following the suggestion, we conduct experiments with PAIR for IRMv1 combined with IB[3], IGA[4], CLOvE[5], SD[6], VREx[7], and Fishr[8], as well as  with all possible combinations of IB, VREx and Fishr. Details can be found in the previous reply([exp. A](https://openreview.net/forum?id=esFxSb_0pSL&noteId=mTZiMqEo--4), [exp. B](https://openreview.net/forum?id=esFxSb_0pSL&noteId=cKBDeD_feq)), as well as in Table 6 and Table 7 in Appendix F.2.
> The main conclusion is that, when considering the suitable objectives for PAIR, the preferred combination of the objectives are those that have theoretical guarantees and are as concise as possible. Otherwise, the optimization can be more difficult which further leads to a performance decrease.
>
> **Q3. The authors may want to state clearly about the generalizability of the theoretical results.**
>
> **R3.** We intend to use the binary classification case to illustrate the idea (otherwise the figures can be too complicated to visualize). Our theoretical results such as Proposition 1 can be easily extended to multi-class cases following the traditional machine learning theory practice. Besides, Theorem 4.1 is general and is compatible with both binary and multi-class cases. We have made corresponding revisions following your suggestions.
>
> **Q4. Can optimization and model selection ideas be combined? Will it be useful in any settings or provide superior performance?**
>
> **R4.** Thank you for the insightful question. Following the suggestion, we conducted experiments based on ColoredMNIST that combined PAIR-o and PAIR-s (given in this  reply([exp. A](https://openreview.net/forum?id=esFxSb_0pSL&noteId=mTZiMqEo--4), [exp. B](https://openreview.net/forum?id=esFxSb_0pSL&noteId=cKBDeD_feq)). The empirical trends provide extensive support for the effectiveness and generality of PAIR-s. It can be found that, given proper optimization objectives for PAIR-o, a combination of PAIR-s can further boost the OOD performance.
>
> **Q5. More discussions on the observed trends may also help better understand possible gaps.**
>
> **R5.** First we’d like to highlight some large improvements brought by PAIR, such as in Camelyon and FMoW where IRMX using linear weight scheme can even decrease the performance, and in ColoredMNIST with a training domain validation model selection.
>
> The improvements of PAIR-o can decrease on some datasets. We hypothesize it’s because of  the inevitable stochastic gradient bias in all MGDA MOO solvers [9], and potentially large variance in estimating the IRMv1 penalties (e.g., RxRx1 where both IRMv1 and VREx are shown to perform poor ), as we discussed in Appendix D.4.2.
>
> While for PAIR-s, as we discussed in Sec. 4.1 that PAIR-s can mitigate the drawbacks of selecting models using a unreliable validation set (has a large gap from the test domain), the improvements will be a bit smaller when the gaps narrow down (e.g., PACS using test domain validation accuracy). Besides, the estimation of satisfaction to Pareto optimality in PAIR-s can also be affected by the variances in estimating loss values in the stochastic setting (e.g., TerraIncognita), as we discussed in Appendix D.2.
>
> The aforementioned limitations of PAIR leave a huge room for future investigations. We have added this discussion to the draft following your suggestions, and we believe it would shed more light on how to resolve the OOD optimization issue.

---

> ### Author Response · Authors · 2022-11-15
> **Welcome for more discussions**
>
> Dear reviewer hPqV,
>
> Thanks again for your time and efforts in reviewing our paper. Here is a summary of our previous response for your information:
> - We made many improvements to the clarity of the paper following your suggestions, such as the notations, the generality of theories, and an in-depth interpretation of the empirical trends observed in Tables 2 and 3;
> - We also conducted a systematic analysis of what other objectives can be incorporated in IRMv1 and be optimized with PAIR. The details are given in previous replies [exp. A](https://openreview.net/forum?id=esFxSb_0pSL&noteId=mTZiMqEo--4), [exp. B](https://openreview.net/forum?id=esFxSb_0pSL&noteId=cKBDeD_feq) to avoid tedious replicates.
>   - The experimental results show that the improvements of PAIR are maximized if the combined OOD objectives are sufficiently robust (i.e., with theoretical guarantees);
>   - Nevertheless, PAIR can also improve the performance of unrobust OOD objective combinations;
>   - Incorporating PAIR-s can further improve the performance of PAIR-o in almost all cases.
>
> We humbly expect you could check it and confirm whether our response has addressed your concerns. We’d be glad to answer any outstanding questions or have further discussions.

---

### Official Review · Reviewer_eVsN · 2022-10-29

**Confidence:** 2
**Correctness:** 3
**Technical Novelty And Significance:** 3
**Empirical Novelty And Significance:** 3
**Recommendation:** 5

**Clarity, Quality, Novelty And Reproducibility:**

Clarity: A bit lacking in the clarify for the theoretical justifications for why invariance is an important principle and why their optimization finds the "best" point out of all Pareto fronts.

Quality/Novelty: The novelty was generally good and the plots are well-done and captions do a good job of explaining some new concepts. The quality of the paper was limited since the main contribution of the paper was to demonstrate the superiority of considering MOO in ERM vs OOD objectives. Throughout the paper, it was not made clear why more objectives should not be considered.

**Strength And Weaknesses:**

The paper provides theoretical instances of tradeoffs between optimizing for ERM vs OOD objectives from a Pareto optimality lens and illustrates their empirical techniques with some theoretical justifications. They also provide better alternative optimization algorithms to perform this optimization than the main baselines.

The main gains are derived from two observations: the first being that using OOD vs ERM objectives "can even eliminate the desired invariant solution from the Pareto front". This is very interesting and they provide a theoretical example in Figure 1, although the model is not explained very clearly. More importantly, it is unclear why the invariance property is even a desired property given that selecting for such solutions will provide an inferior solution in terms of OOD/ERM. If the invariance property is indeed necessary, it is then unclear why it should not be incorporated as an alternative objective. A clear explanation of the inadequacies of the current approach should be elucidated further.

The second gain is from "using the traditional linear weighting scheme that linearly reweights the ERM
and OOD objectives, cannot reach the solution if it lies in the non-convex part of the front" and they propose an alternative method of multiobjective optimization that finds a "best" point on the Pareto frontier. It would be good to explore if optimizing with  non-linear scalarizations (https://arxiv.org/abs/2006.04655) that can provably explore the Pareto frontier could help. Furthermore, the proposed method seems to only find 1 "best" point in the Pareto frontier, which almost degrades to single-objective optimization since there is no utilization of multiple Pareto optimal points (for ensembling for example). It becomes unclear why a MOO approach is needed, other than to perhaps inspire the right "weight" tuning?

In the experiments section, there seems to be no clear "tradeoff" between algorithm objectives that were considered theoretically. Rather it seemed like the different objective were actually the quality of optimization on different environments. Why are these not used as objectives instead? Furthermore, it seems like PAIR-O calls a packaged optimizer (EPO solver), so it does not seem the algorithm was novel.

**Summary Of The Paper:**

The authors provides many instances of optimization dilemma in ERM vs OOD from
the MOO perspective, and attributed the failures of OOD optimization to the compromised robustness
of relaxed OOD objectives and the unreliable optimization scheme.

**Summary Of The Review:**

The paper provided an interesting theoretical understanding of OOD losses and the invariance principle, as well as an advocation for using MOO to balance conflicting objectives. However, the lack of novelty in MOO and clarity in the necessity of their MO framework make this paper a weak reject.

---

> ### Author Response · Authors · 2022-11-11
> **Response to Reviewer eVsN  [6/6]**
>
> It can be found that, as more objectives are involved, the linear weighting scheme can further exacerbate the performance drops of unrobust OOD objective combinations due to the increased complexity of the Pareto front. Although PAIR-o can always improve over the linear weighting scheme, the overall performance may not be better than the original IRMv1. The reason could be a potentially higher divergence of the OOD preference used for PAIR-o from the ideal preference. Notably, PAIR-s can further improve the OOD performance of PAIR-o, demonstrating its significance and generality. Although the best overall performance is obtained by combining PAIR-o and PAIR-s with all the OOD objectives, the performance gain on individual datasets such as CMNIST is obtained by IRMX. Furthermore, considering the growing complexity of the Pareto front that asks for a more precise estimation of the preference, and additional computational cost of the extra objectives, in deep neural networks, a condensed objective combination tends to be more preferred than roughly combining all the objectives.
>
> In summary, when considering the suitable objectives for PAIR, the preferred combination of the objectives are those that have theoretical guarantees and are as concise as possible. We’ve revised our draft to incorporate this discussion.
>
> We’d be grateful if you could take the above responses into consideration when making the final evaluation of our work. Please let us know if there are any outstanding questions.
>
> **References**
>
> [1] Schölkopf et al., Towards Causal Representation Learning, Special Issue of Proceedings of the IEEE - Advances in Machine Learning and Deep Neural Networks, 2021.
>
> [2] Zhao et al., On learning invariant representations for domain adaptation, ICML 2019.
>
> [3] Gulrajani and Lopez-Paz, In search of lost domain generalization, ICLR 2021.
>
> [4] Peters et al., Causal inference using invariant prediction: identification and confidence intervals, Journal of the Royal Statistical Society, 2016.
>
> [5] Arjovsky et al., Invariant Risk Minimization, arXiv 2020.
>
> [6] Kamath et al., Does invariant risk minimization capture invariance? AISTATS 2021.
>
> [7] Golovin and Zhang, Random Hypervolume Scalarizations for Provable Multi-Objective Black Box Optimization, ICML 2020.
>
> [8] Stephen P. Boyd and Lieven Vandenberghe, Convex Optimization, Cambridge University Press, 2014.
>
> [9] Lin et al., ​​Pareto Multi-Task Learning, NeurIPS 2019.
>
> [10] Ahuja et al., Invariance Principle Meets Information Bottleneck for Out-of-Distribution Generalization, NeurIPS 2021.
>
> [11] Rame et al., Fishr: Invariant Gradient Variances for Out-of-Distribution Generalization, ICML 2022.
>
> [12] Wald et al., On Calibration and Out-of-domain Generalization, NeurIPS 2021.
>
> [13] Koyama and Yamaguchi, Out-of-distribution generalization with maximal invariant predictor, arXiv 2020.
>
> [14] Pezeshki et al., Gradient Starvation: A Learning Proclivity in Neural Networks, NeurIPS 2021.

---

> ### Author Response · Authors · 2022-11-11
> **Response to Reviewer eVsN [5/6]**
>
> |                                 | IB           | VREx         | Fishr        | CMNIST                        | CMNIST-M                      | Avg.                | $\Delta$ Avg.       |
> |---------------------------------|--------------|--------------|--------------|-------------------------------|-------------------------------|---------------------|---------------------|
> | ERM                             |              |              |              | $17.14$$\pm${0.73}             | $73.30$$\pm${0.85}             | $45.22$             |                     |
> | IRMv1                           |              |              |              | $67.29$$\pm${0.99}             | $76.89$$\pm${3.23}             | $72.09$             | $+0.00$             |
> | Linear | [x]|              |              | $56.09$$\pm${2.04}             | $75.66$$\pm${10.6}             | $65.88$             | $-6.21$             |
> | PAIR-o                         | [x]|              |              | $61.12$$\pm${2.33}             | $83.30$$\pm${3.00}             | $72.21$             | $+0.12$             |
> | PAIR-o+PAIR-s                  | [x]|              |              | $60.69$$\pm${2.26}             | $83.70$$\pm${1.79}             | $72.20$             | $+0.11$             |
> | Linear |              | [x]|              | $66.19$$\pm${1.41}             | $81.75$$\pm${1.68}             | $73.97$             | $+1.88$             |
> | PAIR-o                         |              | [x]|              | $68.89$$\pm${1.13}             | $83.80$$\pm${1.60}             | $76.35$             | $+4.26$             |
> | PAIR-o+PAIR-s                  |              | [x]|              | $\mathbf{69.16}$$\pm${0.76}    | $\underline{83.96}$$\pm${1.65} | $\underline{76.56}$ | $\underline{+4.47}$ |
> | Linear |              |              | [x]| $66.20$$\pm${2.31}             | $81.07$$\pm${3.98}             | $73.63$             | $+1.54$             |
> | PAIR-o                         |              |              | [x]| $66.45$$\pm${0.90}             | $82.70$$\pm${1.09}             | $74.58$             | $+2.49$             |
> | PAIR-o+PAIR-s                  |              |              | [x]| $67.57$$\pm${0.81}             | $83.22$$\pm${2.10}             | $75.40$             | $+3.31$             |
> | Linear | [x]| [x]|              | $52.61$$\pm${1.56}             | $63.84$$\pm${1.08}             | $58.23$             | $-13.9$             |
> | PAIR-o                         | [x]| [x]|              | $68.35$$\pm${1.73}             | $81.25$$\pm${3.08}             | $74.80$             | $+2.71$             |
> | PAIR-o+PAIR-s                  | [x]| [x]|              | $\underline{69.05}$$\pm${0.76} | $83.11$$\pm${1.46}             | $76.08$             | $+3.99$             |
> | Linear | [x]|              | [x]| $51.91$$\pm${1.26}             | $68.88$$\pm${3.22}             | $60.39$             | $-11.7$             |
> | PAIR-o                         | [x]|              | [x]| $59.70$$\pm${12.7}             | $74.59$$\pm${1.11}             | $67.15$             | $-4.94$             |
> | PAIR-o+PAIR-s                  | [x]|              | [x]| $66.98$$\pm${2.66}             | $75.91$$\pm${3.50}             | $71.45$             | $-0.65$             |
> | Linear |              | [x]| [x]| $64.83$$\pm${2.95}             | $79.34$$\pm${5.77}             | $72.09$             | $+0.00$             |
> | PAIR-o                         |              | [x]| [x]| $67.96$$\pm${1.60}             | $81.44$$\pm${2.24}             | $74.70$             | $+2.61$             |
> | PAIR-o+PAIR-s                  |              | [x]| [x]| $68.19$$\pm${1.58}             | $81.89$$\pm${3.01}             | $75.04$             | $+2.95$             |
> | Linear | [x]| [x]| [x]| $50.00$$\pm${0.32}             | $69.60$$\pm${2.33}             | $59.80$             | $-12.3$             |
> | PAIR-o                         | [x]| [x]| [x]| $66.89$$\pm${1.80}             | $83.46$$\pm${3.10}             | $75.18$             | $+3.08$             |
> | PAIR-o+PAIR-s                  | [x]| [x]| [x]| $68.59$$\pm${1.29}             | $\mathbf{85.30}$$\pm${0.64}    | $\mathbf{76.95}$    | $\mathbf{+4.85}$    |
> | Oracle |              |              |              | $72.08$$\pm${0.24}             | $86.53$$\pm${0.14}             | $79.31$             |                     |

---

> ### Author Response · Authors · 2022-11-11
> **Response to Reviewer eVsN [4/6]**
>
> It can be found that not all OOD objectives can lead the performance gain when incorporated into IRMv1. Sometimes the extra objective can lead to a performance decrease for both the linear weighting scheme and PAIR due to the weakened robustness of the objective combination. It is hard to improve the OOD performance for the following reasons:
> - As more objectives are involved, for the linear weighting scheme, the new Pareto front can be more complicated such that the desired solution is more likely to lie in the non-convex part; While for PAIR, the OOD preference used in \ours tends to have a higher divergence from the ideal one;
> - When the new objective combination is unrobust, the desired solution may not lie in the new Pareto optimal front;
> - Eventhough desired solution lies in the new Pareto optimal front, the weakened OOD robustness introduces more local minimals that have a low OOD losses while worse OOD generalization performance;
>
> ***Therefore, the theoretical soundness is important for choosing proper objectives.*** For example, when incorporating IB objective into IRMv1, the OOD performance drops, since IB is proposed to mitigate a specific type of distribution shifts instead of directly improving learning the invariance in the original IRMv1 setting. In contrast, it can be found that incorporating Fishr can bring performance increases in most cases. The reason is that minimizing Fishr loss can approximately minimize the vrex[11].
>
>
> In addition, we conduct another experiment to study whether more OOD objectives can bring more performance increases. Specifically, we use IB, VREx, Fishr to compose different groups of objectives, and compare the OOD performance optimized via linear weighting scheme and PAIR. More details and results are given at Table 7 in Appendix F.2.

---

> ### Author Response · Authors · 2022-11-11
> **Response to Reviewer eVsN [3/6]**
>
> |                        | IRMv1 | PAIR-o | PAIR-s | CMNIST              | CMNIST-M            | Avg.                | $\Delta$Avg. |
> |------------------------|---------------------------|----------------------------|----------------------------|---------------------|---------------------|---------------------|--------------|
> | ERM                    |                           |                            |                            | $17.14$             | $73.30$             | $45.22$             |              |
> | IRMv1                  |                           |                            |                            | $67.29$             | $76.89$             | $72.09$             | $+0.00$      |
> | IB     |                           |                            |                            | $55.48$             | $76.01$             | $65.75$             |              |
> |                        | [x]              |                            |                            | $56.09$             | $75.66$             | $65.88$             | $-6.21$      |
> |                        | [x]              | [x]               |                            | $61.12$             | $83.30$             | $72.21$             | $+0.12$      |
> |                        | [x]              | [x]               | [x]               | $60.69$             | $83.70$             | $72.20$             | $+0.11$      |
> | VREx   |                           |                            |                            | $68.62$             | $83.52$             | $76.07$             |              |
> |                        | [x]              |                            |                            | $66.19$             | $81.75$             | $73.97$             | $+1.88$      |
> |                        | [x]              | [x]               |                            | $68.89$             | $\underline{83.80}$ | $\underline{76.35}$ | $+4.26$      |
> |                        | [x]              | [x]               | [x]               | $\underline{69.16}$ | $\mathbf{83.96}$    | $\mathbf{76.56}$    | $+4.47$      |
> | Fishr  |                           |                            |                            | $\mathbf{69.38}$    | $77.29$             | $73.34$             |              |
> |                        | [x]              |                            |                            | $66.20$             | $81.07$             | $73.63$             | $+1.54$      |
> |                        | [x]              | [x]               |                            | $68.90$             | $82.70$             | $75.80$             | $+2.49$      |
> |                        | [x]              | [x]               | [x]               | $68.78$             | $84.02$             | $76.40$             | $+3.31$      |
> | CLOvE  |                           |                            |                            | $55.55$             | $74.20$             | $64.88$             |              |
> |                        | [x]              |                            |                            | $66.35$             | $77.70$             | $72.02$             | $-0.07$      |
> |                        | [x]              | [x]               |                            | $64.99$             | $75.70$             | $70.35$             | $-1.75$      |
> |                        | [x]              | [x]               | [x]               | $65.55$             | $77.29$             | $71.42$             | $-0.67$      |
> | IGA    |                           |                            |                            | $58.67$             | $76.27$             | $68.97$             |              |
> |                        | [x]              |                            |                            | $51.22$             | $74.20$             | $62.71$             | $-9.38$      |
> |                        | [x]              | [x]               |                            | $66.17$             | $81.84$             | $74.01$             | $+1.91$      |
> |                        | [x]              | [x]               | [x]               | $66.51$             | $82.12$             | $74.32$             | $+2.23$      |
> | SD     |                           |                            |                            | $62.31$             | $76.73$             | $69.52$             |              |
> |                        | [x]              |                            |                            | $62.48$             | $81.24$             | $71.86$             | $-0.23$      |
> |                        | [x]              | [x]               |                            | $59.52$             | $82.82$             | $71.17$             | $-0.92$      |
> |                        | [x]              | [x]               | [x]               | $65.54$             | $83.57$             | $74.56$             | $+2.47$      |

---

> ### Author Response · Authors · 2022-11-11
> **Response to Reviewer eVsN [2/6]**
>
> **Q4. Rather it seemed like the different objective were actually the quality of optimization on different environments. Why are these not used as objectives instead?**
>
>  **R4.** Throughout the experiments, we strictly follow the previous evaluation protocol that tries best to obtain a high optimization quality of the various OOD objectives. However, due to different priors that are assumed by different OOD objectives, and different relaxations made to realize the assumed priors, different OOD objectives can behave very differently. The reasons could be the violation of the underlying priors of the methods or the compromises in the relaxations. Consequently, desired solution may not lie in the Pareto front of the corresponding objective, hence the solution may not be obtainable even with best quality of optimization. We further provide more empirical evidence to give details on what objectives should be used for optimization in the next reply.
>
> **Q5. Throughout the paper, it was not made clear why more objectives should not be considered.**
>
> **R5.** To study what objectives should be incorporated for optimization and whether more objectives should be considered or not, we conduct two empirical studies to complement our discussion in the paper.
>
> First, we conduct a study of PAIR for IRMv1 combined with other OOD objectives such as IB[10], Fishr[11], CLOvE[12], IGA[13] and SD[14], on ColoredMNIST. More details and resaults are given at Table 6 in Appendix F.2.

---

> ### Author Response · Authors · 2022-11-11
> **Response to Reviewer eVsN [1/6]**
>
> Thank you for taking time to review our paper. Please see our responses to your questions and suggestions below. We reorganize the weakness and questions a bit to better clarify the concerns.
>
>
> **Q1. Why the invariance property is even a desired property given that selecting for such solutions will provide an inferior solution in terms of OOD/ERM. Why it should not be incorporated as an alternative objective.**
>
> **R1.** The necessity of the invariance property originates from the necessity of incorporating causality into representation learning for mitigating the OOD generalization barrier [1]. As discussed in our related work, after the community was aware of the limitations of previous approaches such as DANN [2,3], researchers start to put more and more focus in introducing causal invariance into the learned representations (cf. recently added new algorithms in DomainBed[3]).
>
> Although incorporating the causal invariance property has made some success[4], the framework of Invariant Risk Minimization[5] is arguably the first to formalize the notion of causal invariance in feature representation learning and derive formulations that are compatible with deep neural nets. Yet as discussed in [6] and our work, the original formulation of IRM is too difficult to optimize, and several relaxations have to be made. However, these relaxations inevitably compromise the invariance. Hence we propose PAIR that aims to resolve the optimization and compromise issues together, i.e., recover the required causal invariance in an easy-to-optimize framework. To summarize,
> - The invariance is necessary, but hard to be directly incorporated into the optimization;
> - The relaxations made the optimization of learning invariance possible, but introduces compromises;
>
> The framework of PAIR aims to resolve both the optimization and compromise issues. Besides, the other alternative implementations of the invariance, as we shown in Figure 1 (b), have similar gradient conflicts as IRMv1. Therefore, there also exists  an optimization dilemma in other OOD objectives that prevents the direct incorporation. We have revised our related work discussion to present the inadequacies of the current approach more clearly.
>
> **Q2. Can we explore the Pareto frontier, find multiple Pareto optimal solutions and ensemble them for a better solution?  PAIR seems to only find 1 "best" point in the Pareto frontier, which almost degrades to single-objective optimization since there is no utilization of multiple Pareto optimal points. It becomes unclear why a MOO approach is needed, other than to perhaps inspire the right "weight" tuning?**
>
> **R2.** We first thank you for your thoughtful advice on ensembling multiple Pareto optimal solutions for one that has better OOD performance. We agree that it is a promising future direction to develop and incorporate more sophisticated MOO solvers into the PAIR framework (e.g., non-linear scalarizations [7]). Nevertheless, it might be out of the scope for this work, since our focus is more on pointing out the OOD optimization problem that has been long neglected by the community, and providing both theoretical and empirical supports for the feasibility of resolving the OOD optimization problem from a MOO perspective.
> Besides, we’d like to note that PAIR-o is essentially an adaptive weight tuning approach, and can provably approach the Pareto optimal solutions that single objective approaches (transferred from any fixed linear weights) can fail to, i.e., the Pareto optimal solutions at the non-convex part of Pareto frontier [8, 9]. [7] seem to adopt Bayesian optimization based on some stochastic linear weights to explore the Pareto front, which could also be seen as a kind of adaptive weight tuning. We have revised our draft to incorporate your suggestion and the above discussion to present the necessity of a MOO approach more clearly.
>
> **Q3. In the experiments section, there seems to be no clear "tradeoff" between algorithm objectives that were considered theoretically.**
>
>  **R3.** We’d like to clarify that the tradeoff phenomenon can be observed in Figure 5 (c) and Figure 11 (c, d). Considering the ERM and IRMv1 losses for example, it can be found that, merely optimizing the ERM losses will induce a lower ERM loss but higher IRMv1 loss. Besides, using a linear weighting scheme to optimize IRMv1 and ERM will result in a lower IRMv1 loss but higher ERM loss. In contrast, PAIR finds a better tradeoff that has both moderate ERM and IRMv1 losses.

---

> ### Author Response · Authors · 2022-11-15
> **Welcome for more discussions**
>
> Dear reviewer eVsN,
>
> Thanks again for your time and efforts in reviewing our paper.  Our previous response might be a bit long, hence we provide a summary of our previous response:
> - We provided a clearer explanation of the inadequacies of the current approaches: Invariance property is necessary. The current approaches mostly focus on better objectives hence can all encounter the optimization dilemma;
> - Regarding the novelty, we summarized our key novelty and contributions in this [reply](https://openreview.net/forum?id=esFxSb_0pSL&noteId=eXlvuClLOE0). For the MOO part, our focus is not on inventing a new MOO solver, but on proving the correctness of our understanding of the OOD optimization dilemma, developed from the MOO perspective;
> - We clarified that PAIR-o is essentially a kind of adaptive weight tuning methods which are necessary for obtaining the desired OOD solution. Further improvements on the MOO solver such as exploiting multiple Pareto optimal solutions can be a promising future direction;
> - We clarified the existence of the trade-off in Figure 5 (c);
> - We conducted a systematic analysis of whether and why other objectives can be incorporated in IRMv1 and be optimized with PAIR or not. The experimental results show that PAIR prefers objective combinations with theoretical soundness such as IRMX. They also demonstrate the generality of PAIR in improving the performance of both robust and unrobust objective combinations.
>
> We’d appreciate your patience and we’d be glad to answer any outstanding questions or further discussions.

---

> > ### Comment · Reviewer_eVsN · 2022-11-17
> > **Still unclear why MOO is needed.**
> >
> > It seems like a smart version of scalarization should be tried and considered in your setting, as the main complaint with linear scalarization is that it cannot explore concave regions of the Pareto front. It would be good to explore if optimizing with non-adaptive non-linear scalarizations that can provably explore the Pareto frontier could help (see (https://arxiv.org/abs/2006.04655)). Furthermore, I'm still unsure if the multiobjective approach is still necessary since in the experiments, there does not seem to be a comparison of multiple Pareto optimal points and a clear reason one is preferred over the other.

---

> > > ### Author Response · Authors · 2022-11-17
> > > **Thanks for engagement! Further clarification to address your left concern.**
> > >
> > > Dear Reviewer eVsN,
> > >
> > > Thank you very much for the engagement! We believe our continued discussion could address your left concern well!
> > >
> > > First, we’d like to clarify that **the compared scalarization in our work is actually the vanilla linear scalarization that is commonly used in previous optimization of OOD objectives**. We believe that you also agree with our discussion on the drawbacks of previously widely adopted linear scalarization in finding a desired OOD solution.
> > >
> > > Since we are on the same line, we’d like to further clarify that throughout the paper, our main argument is that we should adopt a MOO solver that can provably approach any solutions in the front, **including [1] which aligns with our conclusions**. In other words, we focus on the **feasibility** instead of the **necessity of a specific solver** used in the paper. More concretely, any solver that satisfies our requirement is a possible solution, but it exceeds the scope of the work to iterate all possible solutions.
> > >
> > > Last but not least, we’d like to clarify that **[1] actually is targeting at a different problem from ours**. Specifically,
> > > - [1] proposes a smart random scalarization to minimize the hypervolume **in order to approach all possible Pareto optimal solutions**. In contrast, PAIR is solving for **one Pareto optimal solution that maximally satisfies the exact Pareto optimality**.  The exact Pareto optimality essentially specifies an expected ratio of the objective loss values, which leads to a solution that properly trades-off ERM and OOD loss values (shown as in Figure 5c).
> > > - Although [1] can obtain multiple Pareto optimal points, **it remains unclear on how to properly combine these solutions to one model** that has better OOD generalization ability. Nevertheless, we believe it’s a promising direction and we’d like to leave it as future work.
> > >
> > > We hope our response could address your concerns and we’re glad to answer any outstanding concerns.
> > >
> > >
> > > **References**:
> > >
> > > [1] Golovin and Zhang, Random Hypervolume Scalarizations for Provable Multi-Objective Black Box Optimization, ICML 2020.

---

> > > > ### Comment · Reviewer_eVsN · 2022-11-21
> > > > **Response**
> > > >
> > > > From what you said, I believe that you are targeting a specific point in the Pareto frontier, which is not novel and can be done with smart scalarizations. Therefore, I will keep my score but am not opposed to see the paper accepted.

---

> > > > > ### Author Response · Authors · 2022-11-22
> > > > > **Thanks for your continued engagement! Perhaps there is a misunderstanding?**
> > > > >
> > > > > Dear Reviewer eVsN,
> > > > >
> > > > > We sincerely thank you for your time and continued engagement! Following your follow-up comment, we surveyed the literature about “smart” scalarizations (non-linear scalarization; with provable guarantees), based on the related work discussions in [1].
> > > > >
> > > > > **It seems there isn’t a proper “smart” scalarization method that can approach a specific Pareto front point given a preference**:
> > > > > First, for traditional non-linear scalarizations such as Chebyshev scalarization and its variants, although their solutions can be preference-speciﬁc for some weights, they can not explore the Pareto Front for all trade-off combinations [2];
> > > > > Besides, for “smarter” scalarizations including [1,3,4,5,6], they are all targeting at multiple Pareto optimal solutions or the entire frontier, instead of a specific point imposed by a preference. The adopted Bayesian optimization in [1,3,4,5,6] is also known to have poor scalability when generalizing to high-dimensional input space [6].
> > > > >
> > > > > Could you provide more specific pointers to the scalarizations that can find a specific Pareto front point given a preference? Or perhaps there was a misunderstanding caused by the related work discussions? We will add the above discussion to our paper once we are able to revise it.
> > > > >
> > > > >
> > > > > **References**
> > > > >
> > > > > [1] Golovin and Zhang, Random Hypervolume Scalarizations for Provable Multi-Objective Black Box Optimization, ICML 2020.
> > > > >
> > > > > [2] Kaisa Miettinen, Nonlinear Multiobjective Optimization, Springer, 1998.
> > > > >
> > > > > [3] Paria et al., A Flexible Framework for Multi-Objective Bayesian Optimization using Random Scalarizations, UAI 2020.
> > > > >
> > > > > [4] Daulton et al., Differentiable Expected Hypervolume Improvement for Parallel Multi-Objective Bayesian Optimization, NeurIPS 2020.
> > > > >
> > > > > [5] Daulton et al., Parallel Bayesian Optimization of Multiple Noisy Objectives with Expected Hypervolume Improvement, NeurIPS 2021.
> > > > >
> > > > > [6] Daulton et al., Multi-Objective Bayesian Optimization over High-Dimensional Search Spaces, UAI 2022.

---

### Official Review · Reviewer_pR4m · 2022-11-01

**Confidence:** 3
**Correctness:** 3
**Technical Novelty And Significance:** 2
**Empirical Novelty And Significance:** 3
**Recommendation:** 5

**Clarity, Quality, Novelty And Reproducibility:**

Clarity and quality wise is good. This paper does a great job on explaining ideas. Plenty empirical evaluations and context discussions are given.
Originality wise can be improved. While I appreciate the engineering dedication, most of the techniques are not novel, from objective, to training, to evaluation, to model selection, as described in Summary Of The Paper. The biggest contribution seems to be Eq (7) with well rounded reason.

**Strength And Weaknesses:**

Strength:
1. The motivation of studying the IRMX objective is well discussed. Firstly, this paper has state-of-art literature on OOD generalization well discussed. Early examples (e.g. Figure 2 and Figure 4) help to understand the drawbacks of IRM variants and the effectiveness of PAIR. Discussions in Appendix B also validates the necessity of keeping IRMv1 in the objective.
2. The proposed model selection criterion PAIR-s achieves consistent worst environment improvement therefore it seems to be a good baseline for follow-up research. I also appreciate the detailed experimental setup.

Weaknesses:
1. I think the experimental setup between IRMX and PAIR-o is not fair enough. Most importantly, I think the one to one correspondence between $p_{ood}$ and linear weighting scheme can be established due to same number of objectives. If this is true, why they have different tuning range? IRMX is from 1 to 1e6 while PAIR is from 1e8 to 1e12 as specified in Fig 5 (d).  Furthermore, IRMX seems can not set different weights between IRMv1 and VREx while PAIR-o can. Is this intended? If IRMX is allowed to tuned as PAIR, I wonder if loss landscapes between these two will match each other in Fig 5(c). Right now they are almost opposite.
2. I am not quite sure about two context reasons provided at Page 5 on why PAIR-o is preferable than IRMX. For (i), how exactly linear weighting scheme can not reach non-convex Pareto front? For (ii), why is $p_{ood}$ considered as a preference between ERM and OOD but linear weighting scheme is not considered a preference?
3. I think the proposed advantage "PAIR-o can avoid exhaustive tuning" can be over claimed. I do not see clear reason the tuning over preference is less heavy than other algorithms such as linear weighting scheme.
4. I think Theorem 4.1/D.4 is misleading. Firstly, the formal Theorem D.4 actually describes the ERM generalization between empirical $\hat{f}^\epsilon$ and population $f^\epsilon$ specified by gap $\delta$. It shouldn't be claimed as "PAIR yield an $\epsilon$-approximated solution of $f_{ood}$" In Theorem 4.1. Secondly, the context before this theorem states "ideal preference is usually unknown" but I doubt the theorem can provide any guidance on choosing p. In particular, it tells us that p should have an upper bound depending on $\delta$ - more accurate solution (i.e. small $\delta$) needs smaller $p$. Is this contradicting the choice $p_{max} = 1e12$ in the experiment?
5. The scalability seems to be a honest issue. While I do not want to emphasize this one too much, I encourage the author to look for alternatives. For example, stochastic algorithms seem an option for IRMX with linear weight scheme.

**Summary Of The Paper:**

This paper illustrates the desired IRM solution $f_{\text{IRM}}$ does not belong to the Pareto front of the IRMv1/IRMs loss included MOO problem in the environment (Kamath et al., 2021) described. This paper includes the VERx objective by (Krueger et al., 2021) into the MOO problem (IRMX) and shows in Proposition 1 that IRMX enjoy the same set of the invariant predictors as the original IRM in the ideal case. This paper illustrates PAIR recovers the causal invariance (Arjovsky et al., 2019) better than IRMv2 and VREx in Figure 4. This paper proposes to solve the IRMX problem by a MGDA-based optimizer with preference-awareness (Mahapatra & Rajan, 2020) and a 2-stage descent and balance scheme (Gulrajani & Lopez-Paz, 2021). Specifically, in the descent step, the PAIR-o algorithm minimizes the ERM loss. In the balance step, the algorithm follows the EPO search algorithm by (Mahapatra & Rajan, 2020). This paper also proposes a model selection method based on the highest PAIR score on the validation set. To relax the computational burden, this paper proposes only using the gradients of the classifier $w$ and choose needs an empirically large enough preference to the OOD objective, $p_{ood} = (1,1e10, 1e12)$. This paper also consider solving IRMX using linear weighting scheme but it achieves suboptimal generalization vs PAIR. The ablation study shows IRMX and PAIR are picking opposite trade-off between VREx and IRMv1 (Fig 5 (c)). Experiments using DOMAINBED evaluation protocol demonstrate PAIR-s can help worst case performance.

**Summary Of The Review:**

While I appreciate the engineering dedication and detailed discussion, my biggest concern is misleading/over-claimed statement, unfair comparison and originality.

---

> ### Author Response · Authors · 2022-11-11
> **Response to Reviewer pR4m [3/3]**
>
> Please let us know if you have any further questions. We’d be grateful if you could take the above responses into consideration when making the final evaluation of our work.
>
> **References**
>
> [1] Stephen P. Boyd and Lieven Vandenberghe, Convex Optimization, Cambridge University Press, 2014.
>
> [2] Lin et al., ​​Pareto Multi-Task Learning, NeurIPS 2019.

---

> ### Author Response · Authors · 2022-11-11
> **Response to Reviewer pR4m [2/3]**
>
> **Q2. About the context reasons provided at Page 5 on why PAIR-o is preferable than IRMX.**
>
> **Q2.1. How exactly linear weighting scheme can not reach non-convex Pareto front?**
>
> **R2.1.** Boyd and Vandenberghe [1] provide an illustrative example in Figure 4.9 of their book. Geometrically, given two objectives, optimizing a MOO problem with different penalty weights can be seen as moving a line in the feasible region towards left and down sides until where there is only one feasible solution in the line, i.e., the one in the Pareto frontier that minimizes the weighted objective values given the corresponding penalty weights. Consequently, given a group of fixed penalty weights, each solution in the non-convex part of the Pareto front is dominated by another solution in the convex part. Therefore, using the linear weighting scheme can hardly approach the solutions at the non-convex part of the Pareto front. [2] also provides numerical examples for illustrating the drawbacks of linear weighting scheme.
>
> When it involves the neural nets and more objectives, the Pareto front could be even more complicated that the desired solution is less likely lying in the convex part of the front. Hence PAIR-o that adopts a MDGA fashion is more preferred, as they are able to approach Pareto optimal solutions at both convex and non-convex parts of the frontier.
>
> **Q2.2. Why is  considered as a preference between ERM and OOD but linear weighting scheme is not considered a preference?**
>
> **R2.2.** As previously explained, there is not a one-to-one correspondence. In other words, even setting IRMv1 and VREx with the same penalty weights, they can have very different loss values at convergence. To better clarify your concern, our experiments in R1 also show that even giving a larger hyperparameter search space for the IRMv1 and VREx penalty weights respectively in IRMX, IRMX remains no better than IRMX optimized with PAIR-o, in both ColoredMNIST and realistic datasets CivilComments and FMoW.
>
> **Q3.  "PAIR-o can avoid exhaustive tuning" can be over claimed.**
>
> **R3.** We respectfully disagree with the “overclaim” statement. As we clarify in **R1**, in ColoredMNIST, PAIR-o uses ***36 times less*** tuning efforts to achieve the top (or better) performances of IRMX optimized with linear weighting scheme. While in real world datasets, we use the same penalty weights for both IRMv1 and VREx following the common practice (e.g., in DomainBed, the penalty weights of IRMv1 and VREx are drawn from the same distribution, and the optimal weights are often the close to each other), and keep the parameter search scope of PAIR-o and IRMX similar (as that for other objectives, following previous evaluation protocols) to ensure a fair comparison. The extra experimental results with broader hyperparameter search scope in **R1**, and the performance gains of PAIR-o in realistic datasets also serve as strong evidence for its weak requirement on parameter tuning.
>
> **Q4.  Theorem 4.1/D.4 is misleading.**
>
> **Q4.1.Theorem D.4 actually describes the ERM generalization.**
>
> **R4.1.** We’d like to clarify that both the ***$f^\epsilon$ and $\hat{f}^\epsilon$ are derived under the constraints of $\epsilon$-approximated Pareto optimality of OOD losses*** (cf. Eq. 31 and Eq. 32). Therefore, when minimizing the ERM losses under the constraints, the OOD losses are also minimized, which guarantees the OOD generalization performance.
>
> **Q4.2.I doubt the theorem can provide any guidance on choosing p**
>
> **R4.2.** We’d like to clarify that, instead of providing guidance on choosing OOD preference, Theorem 4.1/D.4 are derived to discuss the OOD performance of PAIR in terms of sample complexity, given empirical losses and an imprecise OOD preference, since our previous discussions are based on population loss and ideal OOD preference. We have made corresponding changes to make the purposes of Theorem 4.1/D.4 clearer in our updated version.
>
> The discussions on choosing proper OOD preference are given afterward Theorem 4.1 in short and in Appendix D.3 with more details, considering the limited pages.
>
> **Q5.  The scalability seems to be a honest issue. I encourage the author to look for alternatives, e.g., stochastic algorithms seem an option for IRMX with linear weight scheme.**
>
> **R5.** We feel that the use of “full gradients” in the last paragraph of practical considerations in Sec. 4.2 might cause some misunderstandings. We’d like to clarify that “full gradients” means the gradients of the whole predictors, including the featurizer and the classifier, instead of the gradients calculated based on full training data. In fact, the experiments in WILDS and DomainBed are all conducted with stochastic algorithms such as SGD and Adam. We have revised our draft to clarify this potentially confusing point. Could you give us more details on this concern if we misunderstood your point?

---

> ### Author Response · Authors · 2022-11-11
> **Response to Reviewer pR4m [1/3]**
>
> Thank you for your time and detailed review comments. Please see our detailed responses to your comments and suggestions below. We reorganize the weakness and questions a bit to better clarify your concerns.
>
> **Q1. The experimental setup between IRMX and PAIR-o is not fair enough. Is there a one-to-one correspondence between the penalty weights and OOD preference? Do they have matched loss landscapes? Can different objectives in IRMX under linear weighting scheme be set with different weights?**
>
> **R1.** Thanks for pointing out the important aspect between linear weighting scheme and MDGA algorithm (i.e., PAIR-o) used to solve IRMX in our paper. We respectfully disagree with the “unfair comparison” statement.
>
> First, we’d like to clarify that ***there isn’t a one-to-one correspondence neither the matched loss landscapes***  between the penalty weights and OOD preference. The penalty weights are the actual parameters for different objectives, and are fixed. In contrast, the OOD preference serves as the ***ratio*** between the loss values of the expected Pareto optimal solution. PAIR-o, like every MGDA algorithm, will adaptively tune the actual penalty weights at each step in order to approach  the desired solution. Given different penalty weights, the loss landscapes will be different.
>
> Second, we’d like to clarify that IRMX in ***Figure 5 (a) actually uses different weights for IRMv1 and VREx objectives***, and Figure 5 (d) is used to demonstrate the robustness of PAIR-o to different preference choices instead of the preference tuning scope. Each point of IRMX at Figure 5 (a) is ***selected from the best results across all penalty choices from 1 to 1e6 for VREx*** and the corresponding penalty weights for IRMv1, while PAIR-o keeps using the (1,1e10,1e12) preference in most experiments in our paper. In other words, PAIR-o uses ***36 times less*** tuning efforts to achieve the top (or better) performances of  IRMX optimized with linear weighting scheme. The reason we choose the scope from 1 to 1e6 is that the performance of the linear weighting scheme is already saturated or even starts to decrease as we enlarge the penalty weights. While in real world datasets, we keep the scope of hyperparameter search range of PAIR-o and IRMX similar to ensure a fair comparison.
>
> To better demonstrate the advance of PAIR-o over linear weighting scheme, we further conduct experiments on ColoredMNIST with doubled search scope, where both penalty weights for IRMv1 and VREx are tuned from 1 to 1e12, which results 169 hyperparameter choices. The new results are given in Figure. 17 in the revised version, where we can derive the same conclusions. Besides, we also conduct extra experiments in realistic datasets, CivilComments and FMoW from WILDS. The hyperparameter tunning scope is {1e-2, 1, 1e2} for IRMv1 and VREx, respectively.
> | Civil | IRMv1\VREX |   1.00E-02  |       1       |   1.00E+02  | FMoW | IRMv1\VREX |   1.00E-02  |        1       |   1.00E+02  |
> |-------|------------|:-----------:|:-------------:|:-----------:|------|------------|:-----------:|:--------------:|:-----------:|
> |       | 1.00E-02   |  72.45(2.0) |  73.79(1.40)  | 73.13(0.67) |      | 1.00E-02   | 33.64(0.59) |   34.20(1.33)  | 34.43(0.72) |
> |       | 1          | 73.53(1.47) |  74.31(0.83)  | 73.22(0.67) |      | 1          | 30.25(0.87) |   33.75(0.78)  | 33.14(0.56) |
> |       | 1.00E+02   | 72.05(0.59) |  70.05(2.09)  |  74.3(0.51) |      | 1.00E+02   | 21.33(1.51) |   21.00(2.41)  | 13.14(1.63) |
> | PAIR  |            |             | **75.2(0.7)** |             |      |            |             | **35.5(1.13)** |             |
>
>
>
> It can be seen that, even with the penalty weights that have the same relative “preference” to IRMv1 and VREx in linear weighting scheme, it still underperforms PAIR-o. Especially, the best result in CivilComment is achieved with equal penalty weights.

---

> ### Author Response · Authors · 2022-11-15
> **Welcome for more discussions**
>
> Dear reviewer pR4m,
>
> Thanks again for your time and efforts in reviewing our paper. Here is a summary of our previous response as our reply might be a bit long:
> - Regarding the fairness of the comparison between the linear weighting scheme and PAIR-o, we clarified that the IRMX essentially uses different weights for IRMv1 and VREx in Figure 5. We also provided more experimental results to support our claim;
> - Regarding the novelty, we summarized our key novelty and contributions in this [reply](https://openreview.net/forum?id=esFxSb_0pSL&noteId=eXlvuClLOE0);
> - We also provided more clarifications on the necessity of PAIR, the correctness of our theorem, and other mentioned concepts.
>
> We humbly expect you could check our responses below and confirm whether our response has addressed your concerns. Please let us know if there is any outstanding concern.

---

### Author Response · Authors · 2022-11-11
**We have uploaded a revised version [Updated on 11 Nov.]**

Dear reviewers,


We thank all reviewers for their efforts and many helpful comments/suggestions. About the general questions on the novelty of PAIR, we’d like to note that,
- As the previous OOD works mostly focus on developing better objectives to learn the causal invariance, to our best knowledge, this is the ***first*** work that systematically identifies the existence of the OOD optimization dilemma, which has been long neglected in the literature;
- Although MOO is widely adopted to mitigate gradient conflicts in multi-task learning, whether and how we can adopt MOO to analyze and mitigate the OOD optimization dilemma remains ***underexplored***.
- Through the lens of MOO, our work develop ***unique*** understandings about the OOD optimization dilemma, which explains many failure phenomenons in the OOD generalization literature;
- Built upon the understandings, we provide the ***first*** direct solution – PAIR, that proves the feasibility of mitigating the dilemma from the MOO perspective, with plenty of theoretical and empirical supports.

We believe our work could lay the foundations for the development of next-generation optimizers and model selection criteria for OOD-generalizable deep learning models.


Besides, we have revised our paper following the comments/suggestions from all the reviewers. The revision is in purple color in the revised version.

Specifically, we have revised our paper to improve its clarity and readability:
- Improving the discussion with related work (Sec. 2 and Appendix B.1) to better distinguish our contributions (eVsN, hPqV, 1jAA);
- Improving the explanations of why PAIR implemented with MGDA is prefered over linear weighting scheme in Sec. 3.2 (pR4m, hPqV);
- Improving the generalizability of our theory in Sec. 3.2 and method description in Sec. 4.1 (hPqV);
- Adding a table of mathematical notations for the key concepts (hPqV);
- Adding a new section in Appendix B.2 to discuss the observed empirical trends, limitations of PAIR and future directions (hPqV);

We also conducted extra experiments that trained up to ***72, 000*** more models to help readers better understand the significance and generality of PAIR:
- Adding more results on the comparison between linear weighting scheme and PAIR in Appendix F.2, with doubled penalty search scope and extended studies on realistic datasets from FMoW (eVsN). The results provide more evidence for the advantages and necessity of PAIR-o over linear weighting scheme;
- Adding additional studies([exp. A](https://openreview.net/forum?id=esFxSb_0pSL&noteId=mTZiMqEo--4), [exp. B](https://openreview.net/forum?id=esFxSb_0pSL&noteId=cKBDeD_feq)) of PAIR with other OOD objectives, as well as their combinations with IRMv1 (eVsN, hPqV, 1jAA). The results show that the improvements brought by PAIR-o are affected by the quality of optimization objectives. Improperly combined OOD objectives (i.e., those without theoretical guarantees) do not necessarily improve the performance when optimized via PAIR-o, and even decrease the performance when optimized with linear weighting scheme;
- The previous study also examined the performance when incorporating PAIR-s into PAIR-o. The results show that PAIR-s can generally improve the OOD performance of PAIR-o for many OOD objectives and their combinations (hPqV).

In addition, we provide a link of our codes for reproducing the results in our paper: https://anonymous.4open.science/r/PAIR-F789/README.md .

Please let us know if you have any further questions. Thanks.

---

### Decision · Program_Chairs · 2023-01-20

**Decision:**

Accept: poster

**Justification For Why Not Higher Score:**

The paper is definitely interesting and novel. However, it has very limited applicability. I believe it is important for the specific sub community to hear about but not for larger ICLR community. Hence, I see no reason for spotlight or oral designation.

**Justification For Why Not Lower Score:**

The paper is addressing an important problem which has not been considered before. Moreover, it is written clearly and results are promising. I believe it will be impactful to the generalization/causality community and will connect them with multi-objective optimization community. It clearly has enough merit to be published and shared.

**Metareview: Summary, Strengths And Weaknesses:**

The paper is proposing a multi-objective optimization perspective on invariant risk minimization (IRM). The paper starts with demonstration of a problem with optimization of IRMs and later describes this is due to the fact that converting the Bilevel problem into a single objective via Lagrangian results in optimization difficulties. Moreover, authors provide a multi-objective approach to this difficulty and suggest utilizing MOO solvers. The results of this approach is promising and authors provide further theoretical justifications. The paper was reviewed by 4 experts and the summary of the points are as follows:

- Positive Aspects:
  - The paper is very well written and toy problems motivate it well.

- Negative Aspects:
  - Empirical results are not strong enough
  - Mathematical formalism can be improved with more consistent notation and better explanation of the theoretical results
  - MOO approach is not novel

Since the reviews were diverging, I also read the paper in detail and checked all technical details. First of all, I believe MOO approach being not novel is irrelevant as the paper is not about MOO it is about applying MOO to IRM. Hence, I ignore this comment. I agree with the remaining aspects. Moreover, it is also important to note that this paper connects two communities MOO and Domain Generalization/Causality which are somewhat related but never had a strong connection. I believe conceptual novelty of this connection could be very impactful and overwhelm the somewhat limited empirical improvements. I see this as a beginning of such a connection and it can very well be impactful. Overall, I recommend acceptance.

**Note From Pc:**

if the above contains the word "oral" or "spotlight" please see: "oral" presentation means -> notable-top-5% and "spotlight" means -> notable-top-25%. As stated in our emails, we are disassociating presentation type from AC recommendations